# Effectively Modeling Time Series with Simple Discrete State Spaces

**Michael Zhang**[*]**, Khaled Saab**[*]**, Michael Poli, Tri Dao, Karan Goel & Christopher Ré**
Stanford University
`mzhang@cs.stanford.edu, ksaab@stanford.edu`

## Abstract

Time series modeling is a well-established problem, which often requires that methods (1) expressively represent complicated dependencies, (2) forecast long horizons, and (3) efficiently train over long sequences. State-space models (SSMs) are classical models for time series, and prior works combine SSMs with deep learning layers for efficient *sequence modeling*. However, we find fundamental limitations with these prior approaches, proving their SSM representations cannot express autoregressive time series processes. We thus introduce SPACETIME, a new state-space time series architecture that improves all three criteria. For expressivity, we propose a new SSM parameterization based on the *companion matrix*—a canonical representation for discrete-time processes—which enables SPACETIME's SSM layers to learn desirable autoregressive processes. For long horizon forecasting, we introduce a "closed-loop" variation of the companion SSM, which enables SPACETIME to predict many future time-steps by generating its own layer-wise inputs. For efficient training and inference, we introduce an algorithm that reduces the memory and compute of a forward pass with the companion matrix. With sequence length $\ell$ and state-space size $d$, we go from $\tilde{O}(d\ell)$ naïvely to $\tilde{O}(d + \ell)$. In experiments, our contributions lead to state-of-the-art results on extensive and diverse benchmarks, with best or second-best AUROC on 6 / 7 ECG and speech time series classification, and best MSE on 14 / 16 Informer forecasting tasks. Furthermore, we find SPACETIME (1) fits AR($p$) processes that prior deep SSMs fail on, (2) forecasts notably more accurately on longer horizons than prior state-of-the-art, and (3) speeds up training on real-world ETTh1 data by 73% and 80% relative wall-clock time over Transformers and LSTMs.

## 1 Introduction

Time series modeling is a well-established problem, with tasks such as forecasting and classification motivated by many domains such as healthcare, finance, and engineering (Shumway et al., 2000). However, effective time series modeling presents several challenges:

- First, methods should be **expressive** enough to capture complex, long-range, and *autoregressive* dependencies. Time series data often reflects higher order dependencies, seasonality, and trends, which govern how past samples determine future samples (Chatfield, 2000). This motivates many classical approaches that model these properties (Box et al., 1970; Winters, 1960), alongside expressive deep learning mechanisms such as attention (Vaswani et al., 2017) and fully connected layers that model interactions between *every* sample in an input sequence (Zeng et al., 2022).

- Second, methods should be able to forecast a wide range of **long horizons** over various data domains. Reflecting real world demands, popular forecasting benchmarks evaluate methods on 34 different tasks (Godahewa et al., 2021) and 24−960 time-step horizons Zhou et al. (2021). Furthermore, as testament to accurately learning time series processes, forecasting methods should ideally also be able to predict future time-steps on horizons they were not explicitly trained on.

- Finally, methods should be **efficient** with training and inference. Many time series applications require processing very long sequences, *e.g.*, classifying audio data with sampling rates up to 16,000 Hz (Warden, 2018). To handle such settings—where we still need large enough models that

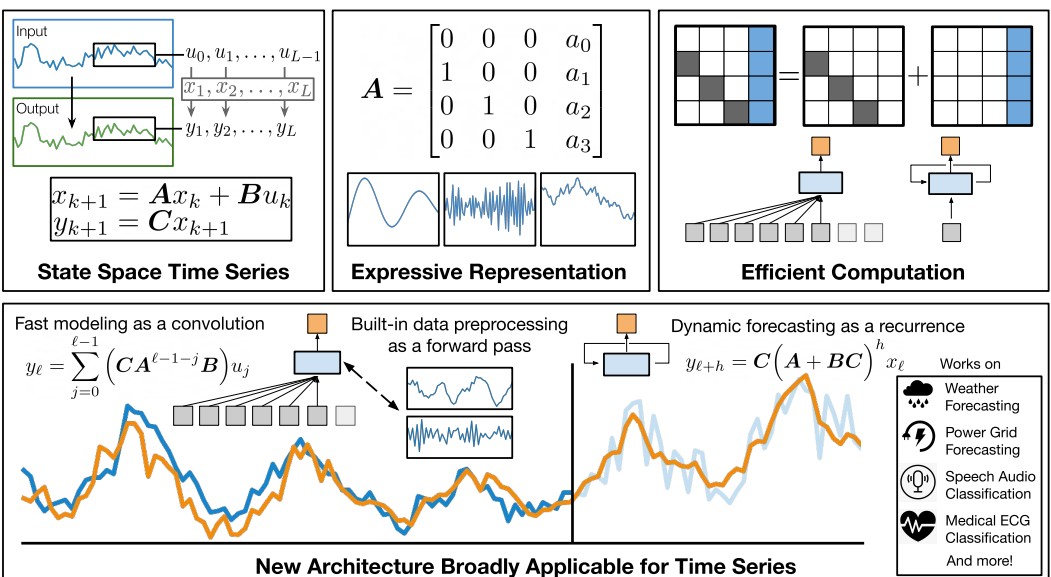

Figure 1: We learn time series processes as state-space models (SSMs) (**top left**). We represent SSMs with the *companion matrix*, which is highly expressive for discrete time series (**top middle**), and compute such SSMs efficiently as convolutions or recurrences via a shift + low-rank decomposition (**top right**). We use these SSMs to build SPACETIME, a new time series architecture broadly effective across tasks and domains (**bottom**).

can expressively model this data—training and inference should ideally scale *subquadratically* with sequence length and model size in time and space complexity.

Unfortunately, existing time series methods struggle to achieve all three criteria. Classical methods (*c.f.*, ARIMA (Box et al., 1970), exponential smoothing (ETS) (Winters, 1960)) often require manual data preprocessing and model selection to identify expressive-enough models. Deep learning methods commonly train to predict specific horizon lengths, *i.e.*, as *direct multi-step forecasting* (Chevillon, 2007), and we find this hurts their ability to forecast longer horizons (Sec. 4.2.2). They also face limitations achieving high expressivity *and* efficiency. Fully connected networks (FCNs) in Zeng et al. (2022) scale quadratically in $\mathcal{O}(\ell h)$ space complexity (with input length $\ell$ and forecast length $h$). Recent Transformer-based models reduce this complexity to $\mathcal{O}(\ell + h)$, but do not always outperform the above FCNs on forecasting benchmarks (Liu et al., 2022; Zhou et al., 2021).

We thus propose **SPACETIME**, a deep state-**space** architecture for effective **time** series modeling. To achieve this, we focus on improving each criteria via three core contributions:

1. For expressivity, our key idea and building block is a linear layer that models time series processes as *state-space models* (SSMs) via the *companion matrix* (Fig. 1). We start with SSMs due to their connections to both classical time series analysis (Kalman, 1960; Hamilton, 1994) and recent deep learning advances (Gu et al., 2021a). Classically, many time series models such as ARIMA and exponential smoothing (ETS) can be expressed as SSMs (Box et al., 1970; Winters, 1960). Meanwhile, recent state-of-the-art deep sequence models (Gu et al., 2021a) have used SSMs to outperform Transformers and LSTMs on challenging long-range benchmarks (Tay et al., 2020). Their primary innovations show how to formulate SSMs as neural network parameters that are practical to train. However, we find limitations with these deep SSMs for time series data. While we build on their advances, we prove that these prior SSM representations (Gu et al., 2021b;a; Gupta, 2022) cannot capture autoregressive processes fundamental for time series. We thus specifically propose the companion matrix representation for its expressive and memory-efficient properties. We prove that the companion matrix SSM recovers fundamental autoregressive (AR) and smoothing processes modeled in classical techniques such as ARIMA and ETS, while only requiring $\mathcal{O}(d)$ memory to represent an $\mathcal{O}(d^2)$ matrix. Thus, SPACETIME inherits the benefits of prior SSM-based sequence models, but introduces improved expressivity to recover fundamental time series processes simply through its layer weights.

2. For forecasting long horizons, we introduce a new "closed-loop" view of SSMs. Prior deep SSM architectures either apply the SSM as an "open-loop" (Gu et al., 2021a), where fixed-length inputs necessarily generate same-length outputs, or use closed-loop autoregression where final layer outputs are fed through the *entire* network as next-time-step inputs (Goel et al., 2022). We

describe issues with both approaches in Sec. 3.2, and instead achieve autogressive forecasting in a deep network with only a single SSM layer. We do so by explicitly training the SSM layer to predict its next time-step *inputs*, alongside its usual outputs. This allows the SSM to recurrently generate its own future inputs that lead to desired outputs—*i.e.*, those that match an observed time series—so we can forecast over many future time-steps without explicit data inputs.

3. For efficiency, we introduce an algorithm for efficient training and inference with the companion matrix SSM. We exploit the companion matrix's structure as a "shift plus low-rank" matrix, which allows us to reduce the time and space complexity for computing SSM hidden states and outputs from $\tilde{\mathcal{O}}(d\ell)$ to $\tilde{\mathcal{O}}(d + \ell)$ in SSM state size $d$ and input sequence length $\ell$.

In experiments, we find SPACETIME consistently obtains state-of-the-art or near-state-of-the-art results, achieving best or second-best AUROC on 6 out of 7 ECG and audio speech time series classification tasks, and best mean-squared error (MSE) on 14 out of 16 Informer benchmark forecasting tasks (Zhou et al., 2021). SPACETIME also sets a new best average ranking across 34 tasks on the Monash benchmark (Godahewa et al., 2021). We connect these gains with improvements on our three effective time series modeling criteria. For expressivity, on synthetic ARIMA processes SPACETIME learns AR processes that prior deep SSMs cannot. For long horizon forecasting, SPACETIME consistently outperforms prior state-of-the-art on the longest horizons by large margins. SPACETIME also generalizes better to *new* horizons not used for training. For efficiency, on speed benchmarks SPACETIME obtains 73% and 80% relative wall-clock speedups over parameter-matched Transformers and LSTMs respectively, when training on real-world ETTh1 data.

## 2 PRELIMINARIES

**Problem setting.** We evaluate effective time series modeling with classification and forecasting tasks. For both tasks, we are given input sequences of $\ell$ "look-back" or "lag" time series samples $\boldsymbol{u}_{t-\ell:t-1} = (u_{t-\ell}, \ldots, u_{t-1}) \in \mathbb{R}^{\ell \times m}$ for sample feature size $m$. For classification, we aim to classify the sequence as the true class $y$ out of possible classes $\mathcal{Y}$. For forecasting, we aim to correctly predict $H$ future time-steps over a "horizon" $\boldsymbol{y}_{t,t+H-1} = (u_t, \ldots, u_{t+H-1}) \in \mathbb{R}^{H \times m}$.

**State-space models for time series.** We build on the discrete-time state-space model (SSM), which maps observed inputs $u_k$ to hidden states $x_k$, before projecting back to observed outputs $y_k$ via

$$x_{k+1} = \boldsymbol{A}x_k + \boldsymbol{B}u_k \tag{1}$$
$$y_k = \boldsymbol{C}x_k + \boldsymbol{D}u_k \tag{2}$$

where $\boldsymbol{A} \in \mathbb{R}^{d \times d}$, $\boldsymbol{B} \in \mathbb{R}^{d \times m}$, $\boldsymbol{C} \in \mathbb{R}^{m' \times d}$, and $\boldsymbol{D} \in \mathbb{R}^{m' \times m}$. For now, we stick to *single-input single-output* conventions where $m, m' = 1$, and let $\boldsymbol{D} = 0$. To model time series in the single SSM setting, we treat $\boldsymbol{u}$ and $\boldsymbol{y}$ as copies of the same process, such that

$$y_{k+1} = u_{k+1} = \boldsymbol{C}(\boldsymbol{A}x_k + \boldsymbol{B}u_k) \tag{3}$$

We can thus learn a time series SSM by treating $\boldsymbol{A}, \boldsymbol{B}, \boldsymbol{C}$ as black-box parameters in a neural net layer, *i.e.*, by updating $\boldsymbol{A}, \boldsymbol{B}, \boldsymbol{C}$ via gradient descent *s.t.* with input $u_k$ and state $x_k$ at time-step $k$, following (3) predicts $\hat{y}_{k+1}$ that matches the next time-step sample $y_{k+1} = u_{k+1}$. This SSM framework and modeling setup is similar to prior works (Gu et al., 2021b;a), which adopt a similar interpretation of inputs and outputs being derived from the "same" process, *e.g.*, for language modeling. Here we study and improve this framework for time series modeling. As extensions, in Sec. 3.1.1 we show how (1) and (2) express univariate time series with the right $\boldsymbol{A}$ representation. In Sec. 3.1.2 we discuss the multi-layer setting, where layer-specific $\boldsymbol{u}$ and $\boldsymbol{y}$ now differ, and we only model first layer inputs and last layer outputs as copies of the same time series process.

## 3 METHOD: SPACETIME

We now present SPACETIME, a deep architecture that uses structured state-spaces for more effective time-series modeling. SPACETIME is a standard multi-layer encoder-decoder sequence model, built as a stack of repeated layers that each parametrize multiple SSMs. We designate the last layer as the "decoder", and prior layers as "encoder" layers. Each encoder layer processes an input time series sample as a sequence-to-sequence map. The decoder layer then takes the encoded sequence representation as input and outputs a prediction (for classification) or output sequence (for forecasting).

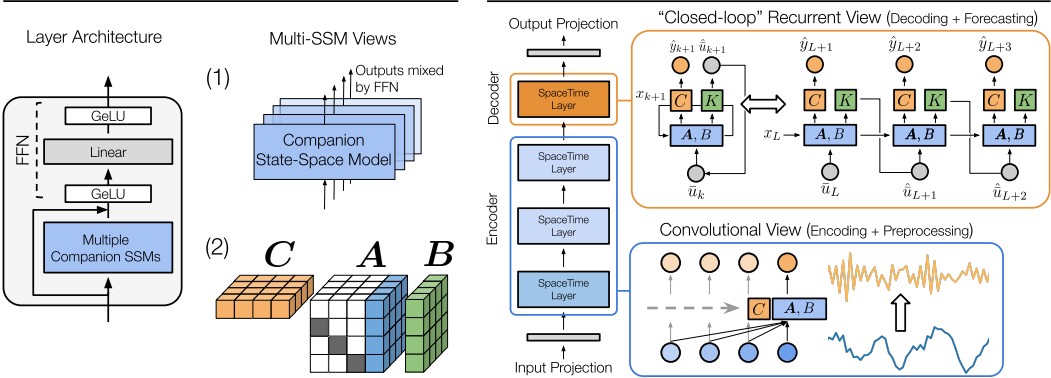

Figure 2: **SPACETIME architecture and components**. **(Left):** Each SPACETIME layer carries weights that model multiple companion SSMs, followed optionally by a nonlinear FFN. The SSMs are learned in parallel (1) and computed as a single matrix multiplication (2). **(Right):** We stack these layers into a SPACETIME network, where earlier layers compute SSMs as convolutions for fast sequence-to-sequence modeling and data preprocessing, while a decoder layer computes SSMs as recurrences for dynamic forecasting.

Below we expand on our contributions that allow SPACETIME to improve expressivity, long-horizon forecasting, and efficiency of time series modeling. In Sec. 3.1, we present our key building block, a layer that parametrizes the *companion matrix* SSM (companion SSM) for expressive and autoregressive modeling. In Sec. 3.2, we introduce a specific instantiation of the companion SSM to flexibly forecast over long horizons. In Sec. 3.3, we provide an efficient inference algorithm that allows SPACETIME to train and predict over long sequences in sub-quadratic time and space complexity.

## 3.1 THE MULTI-SSM SPACETIME LAYER

We discuss our first core contribution and key building block of our model, the SPACETIME layer, which captures the *companion SSM*'s expressive properties, and prove that the SSM represents multiple fundamental processes. To scale up this expressiveness in a neural architecture, we then go over how we represent and compute multiple SSMs in each SPACETIME layer. We finally show how the companion SSM's expressiveness allows us to build in various time series data preprocessing operations in a SPACETIME layer via different weight initializations of the same layer architecture.

### 3.1.1 EXPRESSIVE STATE-SPACE MODELS WITH THE COMPANION MATRIX

For expressive time series modeling, our SSM parametrization represents the state matrix $\boldsymbol{A}$ as a companion matrix. Our key motivation is that $\boldsymbol{A}$ should allow us to capture autoregressive relationships between a sample $u_k$ and various past samples $u_{k-1}, u_{k-2}, \ldots, u_{k-n}$. Such dependencies are a basic yet essential premise for time series modeling; they underlie many fundamental time series processes, *e.g.*, those captured by standard ARIMA models. For example, consider the simplest version of this, where $u_k$ is a linear combination of $p$ prior samples (with coefficients $\phi_1, \ldots, \phi_p$)

$$u_k = \phi_1 u_{k-1} + \phi_2 u_{k-2} + \ldots \phi_p u_{k-p} \tag{4}$$

*i.e.*, a noiseless, unbiased AR($p$) process in standard ARIMA time series analysis (Box et al., 1970).

To allow (3) to express (4), we need the hidden state $x_k$ to carry information about past samples. However, while setting the state-space matrices as trainable neural net weights may suggest we can learn arbitrary task-desirable $\boldsymbol{A}$ and $\boldsymbol{B}$ via supervised learning, prior work showed this could not be done without restricting $\boldsymbol{A}$ to specific classes of matrices (Gu et al., 2021b; Gupta, 2022).

Fortunately, we find that a class of relatively simple $\boldsymbol{A}$ matrices suffices. We propose to set $\boldsymbol{A} \in \mathbb{R}^{d \times d}$ as the $d \times d$ *companion matrix*, a square matrix of the form:

$$(\textbf{Companion Matrix}) \quad \boldsymbol{A} = \begin{bmatrix} 0 & 0 & \ldots & 0 & a_0 \\ 1 & 0 & \ldots & 0 & a_1 \\ 0 & 1 & \ldots & 0 & a_2 \\ \vdots & & \ddots & \vdots & \vdots \\ 0 & 0 & \ldots & 1 & a_{d-1} \end{bmatrix} \quad i.e., \quad \boldsymbol{A}_{i,j} = \begin{cases} 1 & \text{for } i-1 = j \\ a_i & \text{for } j = d-1 \\ 0 & \text{otherwise} \end{cases} \tag{5}$$

Then simply letting state dimension $d = p$, assuming initial hidden state $x_0 = 0$, and setting

$$a := \begin{bmatrix} a_0 & a_1 & \ldots & a_{d-1} \end{bmatrix}^T = \boldsymbol{0}, \quad \boldsymbol{B} = \begin{bmatrix} 1 & 0 & \ldots & 0 \end{bmatrix}^T, \quad \boldsymbol{C} = \begin{bmatrix} \phi_1 & \ldots & \phi_p \end{bmatrix}$$

allows the discrete SSM in (1, 2) to recover the AR($p$) process in (4). We next extend this result in Proposition 1, proving in App. B that setting $\boldsymbol{A}$ as the companion matrix allows the SSM to recover a wide range of fundamental time series and dynamical system processes beyond the AR($p$) process.

**Proposition 1.** *A companion state matrix SSM can represent ARIMA (Box et al., 1970), exponential smoothing (Winters, 1960; Holt, 2004), controllable linear time–invariant systems (Chen, 1984).*

As a result, by training neural network layers that parameterize the companion SSM, we provably enable these layers to learn the ground-truth parameters for multiple time series processes. In addition, as we only update $a \in \mathbb{R}^d$ (5), we can efficiently scale the hidden-state size to capture more expressive processes with only $O(d)$ parameters. Finally, by learning multiple such SSMs in a single layer, and stacking multiple such layers, we can further scale up expressivity in a deep architecture.

**Prior SSMs are insufficient.** We further support the companion SSM by proving that existing related SSM representations used in Gu et al. (2021a); Gupta (2022); Smith et al. (2022); Alcaraz and Strodthoff (2022) *cannot* capture the simple yet fundamental AR($p$) process. Such works, including S4 and S4D, build on the *Linear State-Space Layer* (LSSL) (Gu et al., 2021b), and cannot represent AR processes due to their continuous-time or diagonal parametrizations of $\boldsymbol{A}$.

**Proposition 2.** *No class of continuous-time LSSL SSMs can represent the noiseless AR($p$) process.*

We defer the proof to App. B.1. In Sec. 4.2.1, we support this analysis with empirical results, showing that these prior SSMs fit synthetic AR processes less accurately than the companion SSM. This suggests the companion matrix resolves a fundamental limitation in related work for time series.

### 3.1.2 LAYER ARCHITECTURE AND MULTI-SSM COMPUTATION

**Architecture.** To capture and scale up the companion SSM's expressive and autoregressive modeling capabilities, we model multiple companion SSMs in each SPACETIME layer's weights. SPACE-TIME layers are similar to prior work such as LSSLs, with $\boldsymbol{A}$, $\boldsymbol{B}$, $\boldsymbol{C}$ as trainable weights, and $\boldsymbol{D}$ added back as a skip connection. To model multiple SSMs, we add a dimension to each matrix. For $s$ SSMs per SPACETIME layer, we specify weights $\boldsymbol{A} \in \mathbb{R}^{s \times d \times d}$, $\boldsymbol{B} \in \mathbb{R}^{d \times s}$, and $\boldsymbol{C} \in \mathbb{R}^{s \times d}$. Each slice in the $s$ dimension represents an individual SSM. We thus compute $s$ outputs and hidden states in parallel by following (1) and (2) via simple matrix multiplications on standard GPUs.

To model dependencies across individual SSM outputs, we optionally follow each SPACETIME layer with a one-layer nonlinear feedforward network (FFN). The FFN thus mixes the $m$ outputs across a SPACETIME layer's SSMs, allowing subsequent layers to model dependencies across SSMs.

**Computation.** To compute the companion SSM, we could use the recurrence in (1). However, this sequential operation is slow on modern GPUs, which parallelize matrix multiplications. Luckily, as described in Gu et al. (2021a) we can also compute the SSM as a **1-D convolution**. This enables parallelizable inference and training. To see how, note that given a sequence with at least $k$ inputs and hidden state $x_0 = 0$, the hidden state and output at time-step $k$ by induction are:

$$x_k = \sum_{j=0}^{k-1} \boldsymbol{A}^{k-1-j} \boldsymbol{B} u_j \quad \text{and} \quad y_k = \sum_{j=0}^{k-1} \boldsymbol{C} \boldsymbol{A}^{k-1-j} \boldsymbol{B} u_j \tag{6}$$

We can thus compute hidden state $x_k$ and output $y_k$ as 1-D convolutions with "filters" as

$$\boldsymbol{F}^x = (\boldsymbol{B}, \boldsymbol{A}\boldsymbol{B}, \boldsymbol{A}^2\boldsymbol{B}, \dots, \boldsymbol{A}^{\ell-1}\boldsymbol{B}) \qquad \text{(Hidden State Filter)} \tag{7}$$

$$\boldsymbol{F}^y = (\boldsymbol{C}\boldsymbol{B}, \boldsymbol{C}\boldsymbol{A}\boldsymbol{B}, \boldsymbol{C}\boldsymbol{A}^2\boldsymbol{B}, \dots, \boldsymbol{C}\boldsymbol{A}^{\ell-1}\boldsymbol{B}) \qquad \text{(Output Filter)} \tag{8}$$

$$x_k = (\boldsymbol{F}^x * \boldsymbol{u})[k] \quad \text{and} \quad y_k = (\boldsymbol{F}^y * \boldsymbol{u})[k] \tag{9}$$

So when we have inputs available for each output (*i.e.*, equal-sized input and output sequences) we can obtain outputs by first computing output filters $\boldsymbol{F}^y$ (8), and then computing outputs efficiently with the Fast Fourier Transform (FFT). We thus compute each encoder SSM as a convolution.

For now we note two caveats. Having inputs for each output is not always true, *e.g.*, with long horizon forecasting. Efficient inference also importantly requires that $\boldsymbol{F}^y$ can be computed efficiently, but this is not necessarily trivial for time series: we may have long input sequences with large $k$.

Fortunately we later provide solutions for both. In Sec. 3.2, we show how to predict output samples many time-steps ahead of our last input sample via a "closed-loop" forecasting SSM. In Sec. 3.3 we show how to compute both hidden state and output filters efficiently over long sequences via an efficient inference algorithm that handles the repeated powering of $\boldsymbol{A}^k$.

### 3.1.3 Built-in Data Preprocessing with Companion SSMs

We now show how beyond autoregressive modeling, the companion SSM also enables SPACETIME layers to do standard data preprocessing techniques used to handle nonstationarities. Consider differencing and smoothing, two classical techniques to handle nonstationarity and noise:

$$u_k' = u_k - u_{k-1} \text{ (1st-order differencing)} \ \Big| \ u_k' = \frac{1}{n} \sum_{i=0}^{n-1} u_{k-i} \text{ ($n$-order moving average smoothing)}$$

We explicitly build these preprocessing operations into a SPACETIME layer by simply initializing companion SSM weights. Furthermore, by specifying weights for multiple SSMs, we simultaneously perform preprocessing with various orders in one forward pass. We do so by setting $\boldsymbol{a} = \boldsymbol{0}$ and $\boldsymbol{B} = [1, 0, \ldots, 0]^T$, such that SSM outputs via the convolution view (6) are simple sliding windows / 1-D convolutions with filter determined by $\boldsymbol{C}$. We can then recover arbitrary $n$-order differencing or average smoothing via $\boldsymbol{C}$ weight initializations, *e.g.*, (see App. D.7.1 for more examples),

$$\boldsymbol{C} = \begin{bmatrix} 1 & -2 & 1 & 0 & 0 & \ldots & 0 \\ 1/n & \ldots & 1/n & 0 & 0 & \ldots & 0 \end{bmatrix} \qquad \begin{array}{c} \text{(2nd-order differencing)} \\ \text{($n$-order moving average smoothing)} \end{array} \qquad (10)$$

### 3.2 Long Horizon Forecasting with Closed-loop SSMs

We now discuss our second core contribution, which enables long horizon forcasting. Using a slight variation of the companion SSM, we allow the same constant size SPACETIME model to forecast over many horizons. This *forecasting SSM* recovers the flexible and stateful inference of RNNs, while retaining the faster parallelizable training of computing SSMs as convolutions.

**Challenges and limitations.** For forecasting, a model must process an input lag sequence of length $\ell$ and output a forecast sequence of length $h$, where $h \neq \ell$ necessarily. Many state-of-the-art neural nets thus train by specifically predicting $h$-long targets given $\ell$-long inputs. However, in Sec. 4.2.2 we find this hurts transfer to new horizons in other models, as they only train to predict specific horizons. Alternatively, we could output horizons autoregressively through the network similar to stacked RNNs as in SASHIMI (Goel et al., 2022) or DeepAR (Salinas et al., 2020). However, we find this can still be relatively inefficient, as it requires passing states to each layer of a deep network.

**Closed-loop SSM solution.** Our approach is similar to autoregression, but *only* applied at a single SPACETIME layer. We treat the inputs and outputs as *distinct* processes in a multi-layer network, and add another matrix $\boldsymbol{K}$ to each decoder SSM to model future *input* time-steps explicitly. $\bar{\boldsymbol{u}} = (\bar{u}_0, \ldots, \bar{u}_{\ell-1})$ be the input sequence to a decoder SSM and $\boldsymbol{u} = (u_0, \ldots, u_{\ell-1})$ be the original input sequence, we jointly train $\boldsymbol{A}, \boldsymbol{B}, \boldsymbol{C}, \boldsymbol{K}$ such that $x_{k+1} = \boldsymbol{A}x_k + \boldsymbol{B}\bar{u}_k$, and

$$\hat{y}_{k+1} = \boldsymbol{C}x_{k+1} \qquad \text{(where $\hat{y}_{k+1} = y_{k+1} = u_{k+1}$)} \qquad (11)$$

$$\hat{\bar{u}}_{k+1} = \boldsymbol{K}x_{k+1} \qquad \text{(where $\hat{\bar{u}}_{k+1} = \bar{u}_{k+1}$)} \qquad (12)$$

We thus train the decoder SPACETIME layer to explicitly model its own next time-step inputs with $\boldsymbol{A}, \boldsymbol{B}, \boldsymbol{K}$, and model its next time-step outputs (*i.e.*, future time series samples) with $\boldsymbol{A}, \boldsymbol{B}, \boldsymbol{C}$. For forecasting, we first process the lag terms via (11) and (12) as convolutions

$$x_k = \sum_{j=0}^{k-1} \boldsymbol{A}^{k-1-j} \boldsymbol{B} u_j \quad \text{and} \quad \hat{\bar{u}}_k = \boldsymbol{K} \sum_{j=0}^{k-1} \boldsymbol{A}^{k-1-j} \boldsymbol{B} \bar{u}_j \qquad (13)$$

for $k \in [0, \ell-1]$. To forecast $h$ future time-steps, with last hidden state $x_\ell$ we first predict future input $\hat{\bar{u}}_\ell$ via (12). Plugging this back into the SSM and iterating for $h - 1$ future time-steps leads to

$$x_{\ell+i} = (\boldsymbol{A} + \boldsymbol{B}\boldsymbol{K})^i x_\ell \quad \text{for} \quad i = 1, \ldots, h-1 \qquad (14)$$

$$\Rightarrow (y_\ell, \ldots, y_{\ell+h-1}) = \left( \boldsymbol{C}(\boldsymbol{A} + \boldsymbol{B}\boldsymbol{K})^i x_\ell \right)_{i \in [h-1]} \qquad (15)$$

We can thus use Eq. 15 to get future outputs without sequential recurrence, using the same FFT operation as for Eq. 8, 9. This flexibly recovers $\mathcal{O}(\ell + h)$ time complexity for forecasting $h$ future time-steps, assuming that powers $(\boldsymbol{A} + \boldsymbol{B}\boldsymbol{K})^h$ are taken care of. Next, we derive an efficient matrix powering algorithm to take care of this powering and enable fast training and inference in practice.

### 3.3 Efficient Inference with the Companion SSM

We finally discuss our third contribution, where we derive an algorithm for efficient training and inference with the companion SSM. To motivate this section, we note that prior efficient algorithms

---

**Algorithm 1** Efficient Output Filter $\boldsymbol{F}^y$ Computation

---

**Require:** $\boldsymbol{A}$ is a companion matrix parameterized by the last column $a \in \mathbb{R}^d$, $\boldsymbol{B} \in \mathbb{R}^d$, $\tilde{\boldsymbol{C}} = \boldsymbol{C}(\boldsymbol{I} - \boldsymbol{A}^\ell) \in \mathbb{R}^d$, sequence length $\ell$.

1: Define $\mathrm{quad}(u, v) \in \mathbb{R}^\ell$ for vectors $u, v \in \mathbb{R}^d$: compute $q = u * v$ (linear convolution), zero-pad to length $\ell\lceil d/\ell\rceil$, split into $\lceil d/\ell\rceil$ chunks of size $\ell$ of the form $[q^{(1)}, \ldots, q^{(\lceil d/\ell\rceil)}]$ and return the length-$\ell$ Fourier transform of the sum $\mathcal{F}_\ell(q^{(1)} + \cdots + q^{(\lceil d/\ell\rceil)})$.

2: Compute the roots of unity $z = [\bar{\omega}^0, \ldots, \bar{\omega}^{\ell-1}]$ where $\omega = \exp(-2\pi i/\ell)$.

3: Compute $\tilde{\boldsymbol{F}}^y = \mathrm{quad}(\tilde{\boldsymbol{C}}, \boldsymbol{B}) + \mathrm{quad}(\tilde{\boldsymbol{C}}, a) * \mathrm{quad}(e_d, \boldsymbol{B})/(z - \mathrm{quad}(e_d, a)) \in \mathbb{R}^\ell$, where $e_d = [0, \ldots, 0, 1]$ is the $d$-th basis vector.

4: Return the inverse Fourier transform $\boldsymbol{F}^y = \mathcal{F}_\ell^{-1}(\tilde{\boldsymbol{F}}^y)$.

---

to compute powers of the state matrix $\boldsymbol{A}$ were only proposed to handle specific classes of $\boldsymbol{A}$, and do not apply to the companion matrix (Gu et al., 2021a; Goel et al., 2022; Gu et al., 2022).

Recall from Sec. 3.1.2 that for a sequence of length $\ell$, we want to construct the output filter $\boldsymbol{F}^y = (\boldsymbol{C}\boldsymbol{B}, \ldots, \boldsymbol{C}\boldsymbol{A}^{\ell-1}\boldsymbol{B})$, where $\boldsymbol{A}$ is a $d \times d$ companion matrix and $\boldsymbol{B}, \boldsymbol{C}$ are $d \times 1$ and $1 \times d$ matrices. Naïvely, we could use sparse matrix multiplications to compute powers $\boldsymbol{C}\boldsymbol{A}^j\boldsymbol{B}$ for $j = 0, \ldots, \ell-1$ sequentially. As $\boldsymbol{A}$ has $O(d)$ nonzeros, this would take $O(\ell d)$ time. We instead derive an algorithm that constructs this filter in $O(\ell \log \ell + d \log d)$ time. The main idea is that rather than computing the filter directly, we can compute its spectrum (its discrete Fourier transform) more easily, *i.e.*,

$$\tilde{\boldsymbol{F}}^y[m] := \mathcal{F}(\boldsymbol{F}^y) = \sum_{j=0}^{\ell-1} \boldsymbol{C}\boldsymbol{A}^j \omega^{mj}\boldsymbol{B} = \boldsymbol{C}(\boldsymbol{I} - \boldsymbol{A}^\ell)(\boldsymbol{I} - \boldsymbol{A}\omega^m)^{-1}\boldsymbol{B}, \quad m = 0, 1, \ldots, \ell-1.$$

where $\omega = \exp(-2\pi i/\ell)$ is the $\ell$-th root of unity. This reduces to computing the quadratic form of the resolvent $(\boldsymbol{I} - \boldsymbol{A}\omega^m)^{-1}$ on the roots of unity (the powers of $\omega$). Since $\boldsymbol{A}$ is a companion matrix, we can write $\boldsymbol{A}$ as a shift matrix plus a rank-1 matrix, $\boldsymbol{A} = \boldsymbol{S} + ae_d^T$. Thus Woodbury's formula reduces this computation to the resolvent of a shift matrix $(\boldsymbol{I} - \boldsymbol{S}\omega^m)^{-1}$, with a rank-1 correction. This resolvent can be shown analytically to be a lower-triangular matrix consisting of roots of unity, and its quadratic form can be computed by the Fourier transform of a linear convolution of size $d$. Thus one can construct $\boldsymbol{F}_k^y$ by linear convolution and the FFT, resulting in $O(\ell \log \ell + d \log d)$ time.

We validate in Sec. 4.2.3 that Algorithm 1 leads to a wall-clock time speedup of $2\times$ compared to computing the output filter naïvely by powering $\boldsymbol{A}$. In App. B.2, we prove the time complexity $O(\ell \log \ell + d \log d)$ and correctness of Algorithm 1. We also provide an extension to the closed-loop SSM, which can also be computed in subquadratic time as $\boldsymbol{A} + \boldsymbol{B}\boldsymbol{K}$ is a shift plus rank-2 matrix.

## 4 EXPERIMENTS

We test SPACETIME on a broad range of time series forecasting and classification tasks. In Sec. 4.1, we evaluate whether SPACETIME's contributions lead to state-of-the-art results on standard benchmarks. To help explain SPACETIME's performance and validate our contributions, in Sec. 4.2 we then evaluate whether these gains coincide with empirical improvements in expressiveness (Sec. 4.2.1), forecasting flexibility (Sec. 4.2.2), and training efficiency (Sec. 4.2.3).

### 4.1 MAIN RESULTS: TIME SERIES FORECASTING AND CLASSIFICATION

For forecasting, we evaluate SPACETIME on 40 forecasting tasks from the popular Informer (Zhou et al., 2021) and Monash (Godahewa et al., 2021) benchmarks, testing on horizons 8 to 960 time-steps long. For classification, we evaluate SPACETIME on seven medical ECG or speech audio classification tasks, which test on sequences up to 16,000 time-steps long. For all results, we report mean evaluation metrics over three seeds. ✗ denotes the method was computationally infeasible on allocated GPUs, *e.g.*, due to memory constraints (same resources for all methods; see App. C for details). App. C also contains additional dataset, implementation, and hyperparameter details.

**Informer (forecasting).** We report univariate time series forecasting results in Table 1, comparing against recent state-of-the-art methods (Zeng et al., 2022; Zhou et al., 2022a), related state-space models (Gu et al., 2021a), and other competitive deep architectures. We include extended results on additional horizons and multivariate forecasting in App. D.2. We find SPACETIME obtains lowest MSE and MAE on 14 and 11 forecasting settings respectively, $3\times$ more than prior state-of-the-art. SPACETIME also outperforms S4 on 15 / 16 settings, supporting the companion SSM representation.

**Monash (forecasting).** We also evaluate on 32 datasets in the Monash forecasting benchmark (Godahewa et al., 2021), spanning domains including finance, weather, and traffic. For space, we report

Table 1: **Univariate forecasting** results on Informer ETT datasets. **Best** results in **bold**. SPACETIME results reported as means over three seeds. Additional datasets, horizons, and method comparisons in App. D.2

| Methods | | SpaceTime | | NLinear | | FILM | | S4 | | FedFormer | | Autoformer | | Informer | | ARIMA | |
|---|---|---|---|---|---|---|---|---|---|---|---|---|---|---|---|---|---|
| Metric | | MSE | MAE | MSE | MAE | MSE | MAE | MSE | MAE | MSE | MAE | MSE | MAE | MSE | MAE | MSE | MAE |
| ETTh1 | 96 | 0.054 | 0.181 | **0.053** | **0.177** | 0.055 | 0.178 | 0.316 | 0.490 | 0.079 | 0.215 | 0.071 | 0.206 | 0.193 | 0.377 | 0.058 | 0.184 |
| | 192 | **0.066** | 0.207 | 0.069 | **0.204** | 0.072 | 0.207 | 0.345 | 0.516 | 0.104 | 0.245 | 0.114 | 0.262 | 0.217 | 0.395 | 0.073 | 0.209 |
| | 336 | **0.069** | **0.212** | 0.081 | 0.226 | 0.083 | 0.229 | 0.825 | 0.846 | 0.119 | 0.270 | 0.107 | 0.258 | 0.202 | 0.381 | 0.086 | 0.231 |
| | 720 | **0.076** | **0.222** | 0.080 | 0.226 | 0.090 | 0.240 | 0.190 | 0.355 | 0.142 | 0.299 | 0.126 | 0.283 | 0.183 | 0.355 | 0.103 | 0.253 |
| ETTh2 | 96 | **0.119** | **0.268** | 0.129 | 0.278 | 0.127 | 0.272 | 0.381 | 0.501 | 0.128 | 0.271 | 0.153 | 0.306 | 0.213 | 0.373 | 0.273 | 0.407 |
| | 192 | **0.151** | **0.306** | 0.169 | 0.324 | 0.182 | 0.335 | 0.332 | 0.458 | 0.185 | 0.330 | 0.204 | 0.351 | 0.227 | 0.387 | 0.315 | 0.446 |
| | 336 | **0.169** | **0.332** | 0.194 | 0.355 | 0.204 | 0.367 | 0.655 | 0.670 | 0.231 | 0.378 | 0.246 | 0.389 | 0.242 | 0.401 | 0.367 | 0.488 |
| | 720 | **0.188** | **0.352** | 0.225 | 0.381 | 0.241 | 0.396 | 0.630 | 0.662 | 0.278 | 0.420 | 0.268 | 0.409 | 0.291 | 0.439 | 0.413 | 0.519 |
| ETTm1 | 96 | **0.026** | **0.121** | **0.026** | 0.122 | 0.029 | 0.127 | 0.651 | 0.733 | 0.033 | 0.140 | 0.056 | 0.183 | 0.109 | 0.277 | 0.033 | 0.136 |
| | 192 | **0.039** | 0.152 | **0.039** | **0.149** | 0.041 | 0.153 | 0.190 | 0.372 | 0.058 | 0.186 | 0.081 | 0.216 | 0.151 | 0.310 | 0.049 | 0.169 |
| | 336 | **0.051** | 0.173 | 0.052 | **0.172** | 0.053 | 0.175 | 0.428 | 0.581 | 0.084 | 0.231 | 0.076 | 0.218 | 0.427 | 0.591 | 0.065 | 0.196 |
| | 720 | 0.074 | 0.213 | 0.073 | 0.207 | **0.071** | **0.205** | 0.254 | 0.433 | 0.102 | 0.250 | 0.110 | 0.267 | 0.438 | 0.586 | 0.089 | 0.231 |
| ETTm2 | 96 | **0.060** | **0.179** | 0.063 | 0.182 | 0.065 | 0.189 | 0.153 | 0.318 | 0.067 | 0.198 | 0.065 | 0.189 | 0.088 | 0.225 | 0.211 | 0.340 |
| | 192 | **0.090** | **0.222** | **0.090** | 0.223 | 0.094 | 0.233 | 0.183 | 0.350 | 0.102 | 0.245 | 0.118 | 0.256 | 0.132 | 0.283 | 0.237 | 0.371 |
| | 336 | **0.113** | **0.255** | 0.117 | 0.259 | 0.124 | 0.274 | 0.204 | 0.367 | 0.130 | 0.279 | 0.154 | 0.305 | 0.180 | 0.336 | 0.264 | 0.396 |
| | 720 | **0.166** | **0.318** | 0.170 | 0.318 | 0.173 | 0.323 | 0.482 | 0.567 | 0.178 | 0.325 | 0.182 | 0.335 | 0.300 | 0.435 | 0.310 | 0.441 |
| Count | | **14** | **11** | 4 | 4 | 1 | 1 | 0 | 0 | 0 | 0 | 0 | 0 | 0 | 0 | 0 | 0 |

Table 2: **ECG statement classification** on PTB-XL (100 Hz version). Baseline AUROC from Strodthoff et al. (2021) (error bars in App. D.4).

| Task AUROC | All | Diag | Sub-diag | Super-diag | Form | Rhythm |
|---|---|---|---|---|---|---|
| **SPACETIME** | 0.936 | **0.941** | **0.933** | 0.929 | 0.883 | 0.967 |
| S4 | **0.938** | 0.939 | 0.929 | **0.931** | 0.895 | **0.977** |
| Inception-1D | 0.925 | 0.931 | 0.930 | 0.921 | **0.899** | 0.953 |
| xRN-101 | 0.925 | 0.937 | 0.929 | 0.928 | 0.896 | 0.957 |
| LSTM | 0.907 | 0.927 | 0.928 | 0.927 | 0.851 | 0.953 |
| Transformer | 0.857 | 0.876 | 0.882 | 0.887 | 0.771 | 0.831 |
| Wavelet + NN | 0.849 | 0.855 | 0.859 | 0.874 | 0.757 | 0.890 |

Table 3: **Speech audio classification**

| Method | Acc. (%) |
|---|---|
| SPACETIME | 97.29 |
| S4 | **98.32** |
| LSSL | ✗ |
| WaveGan-D | 96.25 |
| Transformer | ✗ |
| Performer | 30.77 |
| CKConv | 71.66 |

results in Table 20 (App. D.3). We compare against 13 classical and deep learning baselines. SPACE-TIME achieves best RMSE on 7 tasks and sets new state-of-the-art average performance across all 32 datasets. SPACETIME's relative improvements also notably grow on long horizon tasks (Fig. 6).

**ECG (multi-label classification).** Beyond forecasting, we show that SPACETIME can also perform state-of-the-art time series classification. To classify sequences, we use the same sequence model architecture in Sec. 3.1. Like prior work (Gu et al., 2021a), we simply use the last-layer FFN to project from number of SSMs to number of classes, and mean pooling over length before a softmax to output class logits. In Table 2, we find that SPACETIME obtains best or second-best AUROC on five out of six tasks, outperforming both general sequence models and specialized architectures.

**Speech Audio (single-label classification).** We further test SPACETIME on long-range audio classification on the Speech Commands dataset (Warden, 2018). The task is classifying raw audio sequences of length 16,000 into 10 word classes. We use the same pooling operation for classification as in ECG. SPACETIME outperforms domain-specific architectures, *e.g.*, WaveGan-D (Donahue et al., 2018) and efficient Transformers, *e.g.*, Performer (Choromanski et al., 2020) (Table 3).

## 4.2 IMPROVEMENT ON CRITERIA FOR EFFECTIVE TIME SERIES MODELING

For further insight into SPACETIME's performance, we now validate that our contributions improve expressivity (4.2.1), forecasting ability (4.2.2), and efficiency (4.2.3) over existing approaches.

### 4.2.1 EXPRESSIVITY

To first study SPACETIME's expressivity, we test how well SPACETIME can fit controlled autoregressive processes. To validate our theory on SPACETIME's expressivity gains in Sec. 3.1, we compare against recent related SSM architectures such as S4 (Gu et al., 2021a) and S4D (Gu et al., 2022).

For evaluation, we generate noiseless synthetic AR($p$) sequences. We test if models learn the true process by inspecting whether the trained model weights recover *transfer functions* specified by the AR coefficients (Oppenheim, 1999). We use simple 1-layer 1-SSM models, with state-space size equal to AR $p$, and predict one time-step given $p$ lagged inputs (the smallest sufficient setting).

In Fig. 3 we compare the trained forecasts and transfer functions (as frequency response plots) of SPACETIME, S4, and S4D models on a relatively smooth AR(4) process and sharp AR(6) process.

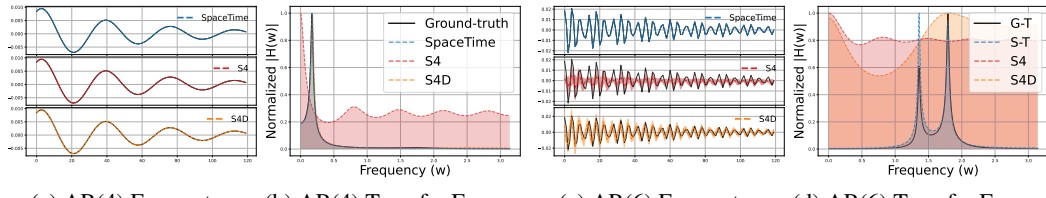

| (a) AR(4) Forecast | (b) AR(4) Transfer Func. | (c) AR(6) Forecast | (d) AR(6) Transfer Func. |

Figure 3: **AR($p$) expressiveness benchmarks**. SPACETIME captures AR processes more precisely than similar deep SSM models, forecasting future samples and learning ground-truth transfer functions more accurately.

Table 4: **Longer horizon forecasting** on Informer ETTh datasets. Mean standardized MSE reported. SPACETIME obtains lower MSE when trained to forecast longer horizons.

| Dataset | Horizon | 720 | 960 | 1080 | 1440 | 1800 | 1920 |
|---------|---------|-----|-----|------|------|------|------|
| ETTh1 | NLinear | 0.080 | 0.089 | 0.085 | 0.094 | 0.102 | 0.104 |
|       | SPACETIME | **0.075** | **0.074** | **0.072** | **0.080** | **0.081** | **0.088** |
| ETTh2 | NLinear | 0.224 | 0.273 | 0.290 | 0.329 | 0.450 | 0.493 |
|       | SPACETIME | **0.188** | **0.225** | **0.265** | **0.299** | **0.438** | **0.459** |

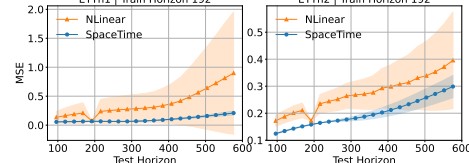

Figure 4: **Forecasting transfer**. Mean MSE ($\pm 1$ SD). SPACETIME transfers more accurately and consistently to new horizons than NLinear.

Our results support the relative expressivity of SPACETIME's companion matrix SSM. While all models accurately forecast the AR(4) time series, only SPACETIME recovers the ground-truth transfer functions for both, and notably forecasts the AR(6) process more accurately (Fig. 3c, d).

### 4.2.2 LONG-HORIZON FORECASTING

To next study SPACETIME's improved long horizon forecasting capabilities, we consider two additional long horizon tasks. First, we test on much longer horizons than prior settings (*c.f.*, Table 1). Second, we test a new forecasting ability: how well methods trained to forecast one horizon transfer to longer horizons at test-time. For both, we use the popular Informer ETTh datasets. We compare SPACETIME with NLinear—the prior state-of-the-art on longer-horizon ETTh datasets—an FCN that learns a dense linear mapping between every lag input and horizon output (Zeng et al., 2022).

We find SPACETIME outperforms NLinear on both long horizon tasks. On training to predict long horizons, SPACETIME consistently obtains lower MSE than NLinear on all settings (Table 4). On transferring to new horizons, SPACETIME models trained to forecast 192 time-step horizons transfer more accurately and consistently to forecasting longer horizons up to 576 time-steps (Fig. 4). This suggests SPACETIME's autoregressive forecasting more convincingly learns the time series process; rather than only fitting to the specified horizon, the same model can generalize to new horizons.

### 4.2.3 EFFICIENCY

To finally study if our companion matrix algorithm enables efficient training on long sequences, we conduct two speed benchmarks. We (1) compare the wall-clock time per training epoch of SPACETIME to standard sequence models, *e.g.*, LSTMs and Transformers, with similar pararemeter counts, and (2) empirically test our theory in Sec. 3.3, which suggests SPACETIME trains near-linearly with sequence length and state dimension. For (1), we use ETTh1 data with lag and horizon 720 time-steps long. For (2), we use synthetic data, scaling sequences from $100-2000$ time-steps long.

On (1) we find SPACETIME reduces clock time on ETTh1 by 73% and 80% compared to Transformers and LSTMs (Table 5). Our efficient algorithm (Sec. 3.3) is also important; it speeds up training by $2\times$, and makes SPACETIME's training time competitive with efficient models such as S4. On (2), we find SPACETIME also scales near-linearly with input sequence length, achieving 91% faster training time versus similarly recurrent LSTMs (Fig. 5).

Table 5: **Train wall-clock time**. ETTh1 seconds per epoch.

| Method | # params | sec/epoch |
|--------|----------|-----------|
| SPACETIME | 148k | 66 |
| → No Algorithm 1 | 148k | 132 |
| S4 | 151k | 49 |
| Transformer | 155k | 240 |
| LSTM | 145k | 336 |

Figure 5: **Wall-clock scaling**. SPACETIME scales near-linearly.

## 5 CONCLUSION

We introduce SPACETIME, a state-space time series model. We achieve high expressivity by modeling SSMs with the companion matrix, long-horizon forecasting with a closed-loop SSM variant, and efficiency with a new algorithm to compute the companion SSM. We validate SPACETIME's proposed components on extensive time series forecasting and classification tasks.

## 6 ETHICS STATEMENT

A main objective of our work is to improve the ability to classify and forecast time series, which has real-world applications in many fields. These applications may have high stakes, such as classifying abnormalities in medical time series. In these situations, incorrect predictions may lead to harmful patient outcomes. It is thus critical to understand that while we aim to improve time series modeling towards these applications, we do not solve these problems. Further analysis and development into where models fail in time series modeling is necessary, including potentials intersections with research directions such as robustness and model biases when aiming to deploy machine learning models in real world applications.

## 7 REPRODUCIBILITY

We include code to reproduce our main results in Table 1 in the supplementary material. We provide training hyperparameters and dataset details for each benchmark in Appendix C, discussing the Informer forecasting benchmark in Appendix C.1, the Monash forecasting benchmark in Appendix C.2, and the ECG and speech audio classification benchmarks in Appendix C.3. We provide proofs for all propositions and algorithm complexities in Appendix B.

## 8 ACKNOWLEDGEMENTS

We thank Albert Gu, Yining Chen, Ke Alexander Wang, and Rose Wang for helpful discussions and feedback. We also gratefully acknowledge the support of NIH under No. U54EB020405 (Mobilize), NSF under Nos. CCF1763315 (Beyond Sparsity), CCF1563078 (Volume to Velocity), and 1937301 (RTML); US DEVCOM ARL under No. W911NF-21-2-0251 (Interactive Human-AI Teaming); ONR under No. N000141712266 (Unifying Weak Supervision); ONR N00014-20-1-2480: Understanding and Applying Non-Euclidean Geometry in Machine Learning; N000142012275 (NEPTUNE); NXP, Xilinx, LETI-CEA, Intel, IBM, Microsoft, NEC, Toshiba, TSMC, ARM, Hitachi, BASF, Accenture, Ericsson, Qualcomm, Analog Devices, Google Cloud, Salesforce, Total, the HAI-GCP Cloud Credits for Research program, the Stanford Data Science Initiative (SDSI), and members of the Stanford DAWN project: Facebook, Google, and VMWare. The U.S. Government is authorized to reproduce and distribute reprints for Governmental purposes notwithstanding any copyright notation thereon. Any opinions, findings, and conclusions or recommendations expressed in this material are those of the authors and do not necessarily reflect the views, policies, or endorsements, either expressed or implied, of NIH, ONR, or the U.S. Government.

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

# Effectively Modeling Time Series with Simple Discrete State Spaces

## Table of Contents

## A   RELATED WORK

### A.1   CLASSICAL APPROACHES

Classical approaches in time series modeling include the Box-Jenkins method (Box and Jenkins, 1968), exponential smoothing (Hyndman et al., 2008; Winters, 1960), autoregressive integrated moving average (ARIMA) (Box et al., 1970), and state-space models (Hamilton, 1994). In such approaches, the model is usually manually selected based analyzing time series features (e.g., seasonality and order of non-stationarity), where the selected model is then fitted for each individual time series. While classical approaches may be more interpretable than recent deep learning techniques, the domain expertise and manual labor needed to succesfully apply them renders them infeasible to the common setting of modeling thousands, or millions, of time series.

### A.2   DEEP LEARNING APPROACHES

**Recurrent models.** Common deep learning architectures for modeling sequence data are the family of recurrent neural networks, which include GRUs (Chung et al., 2014), LSTMs (Hochreiter and Schmidhuber, 1997), and DeepAR (Salinas et al., 2020). However, due to the recurrent nature of RNNs, they are slow to train and may suffer from vanishing/exploding gradients, making them difficult to train (Pascanu et al., 2013).

**Deep State Space models.** Recent work has investigated combining the expressive strengths of SSMs with the scalable strengths of deep neural networks (Rangapuram et al., 2018; Gu et al., 2021a). Rangapuram et al. (2018) propose to train a global RNN that transforms input covariates to sequence-spcific SSM parameters; however, one downside of this approach is that they inherit the drawbacks of RNNs. More recent approaches, such as LSSL (Gu et al., 2021b), S4 (Gu et al., 2021a), S4D (Gu et al., 2022), and S5 (Smith et al., 2022), directly parameterize the layers of a neural network with multiple linear SSMs, and overcome common recurrent training drawbacks by leveraging the convolutional view of SSMs. While deep SSM models have been shown great promise in time series modeling, we show in our work – which builds off deep SSMs – that current deep SSM approaches are not able to capture autoregressive processes due to their continuous nature.

**Neural differential equations as nonlinear state spaces.** (Chen et al., 2018) parametrizes the vector field of continuous–time autonomous systems. These models, termed *Neural Differential Equations* (NDEs) have seen extensive application to time series and sequences, first by Rubanova et al. (2019) and then by Kidger et al. (2020); Morrill et al. (2021); Massaroli et al. (2021) with the notable extension to *Neural Controlled Differential Equations* (Neural CDEs). Neural CDEs can be considered the continuous–time, nonlinear version of state space models and RNNs (Kidger, 2022). Rather than introducing nonlinearity between linear state space layers, Neural CDEs model nonlinear systems driven by a control input.

The NDE framework has been further applied by Poli et al. (2019) to model graph time series via *Neural Graph Differential Equations*. In Queiruga et al. (2020), a continuous-depth ResNet generalization based on ODEs is proposed, and in Kim et al. (2021) numerical techniques to enable learning of stiff dynamical systems with Neural ODEs are investigated. The idea of parameterizing the vector field of a differential equation with a neural network, popularized by NDEs, can be traced back to earlier works (Funahashi and Nakamura, 1993; Zhang et al., 2014; Weinan, 2017).

**Transformers.** While RNNs and its variants have shown some success at time series modeling, a major limitation is their applicability to long input sequences. Since RNNs are recurrent by nature, they require long traversal paths to access past inputs, which leads to vanishing/exploding gradients and as a result struggle with capturing long-range dependencies.

To counteract the long-range dependency problem with RNNs, a recent line of work considers Transformers for time series modeling. The motivation is that due to the attention mechanism, a Transformer can directly model dependencies between any two points in the input sequence, independently of how far apart the points are. However, the high expressivity of the attention mechanism comes at the cost of the time and space complexity being quadratic in sequence length, making Transformers infeasible for very long sequences. As a result, many works consider specialized Transformer architectures with sparse attention mechanisms to bring down the quadratic complexity. For example, Beltagy et al. (2020) propose LogSparse self-attention, where a cell attends to a subset of past cells (as opposed to all cells), where closer cells are attended to more frequently, proportional to the log of their distance, which brings down complexity from $\mathcal{O}(\ell^2)$ to $\mathcal{O}(\ell(\log \ell)^2)$. Zhou et al. (2021) propose ProbSparse self-attention, which achieves $\mathcal{O}(\ell \log \ell)$ time and memory complexity, where they propose a generative style decoder to speed inference. Liu et al. (2022) propose a pyramidal attention mechanism which shows linear time and space complexity with sequence length. Autoformer (Wu et al., 2021) suggests more specialization is needed in time series with a decomposition forecasting architecture, which extracts long-term stationary trend from the seasonal series and utilizes an auto-correlation mechanism, which discovers the period-based dependencies. Zhou et al. (2022b) believes previous attempts of Transformer-based architectures do not capture global statistical properties, and to do so requires an attention mechanism in the frequency domain. Conformer (Gulati et al., 2020) stacks convolutional and self-attention modules into a shared layer to combine the strengths of local interactions from convolutional modules and global interactions from self-attention modules. Perceiver AR (Hawthorne et al., 2022) builds on the Perceiver architecture, which reduces the computational complexity of transformers by performing self-attention in a latent space, and extends Perceiver's applicability to causal autoregressive generation.

While these works have shown exciting progress on time series forecasting, their proposed architectures are specialized to handle specific time series settings (e.g., long input sequences, or seasonal sequences), and are commonly trained to output a fixed target horizon length (Zhou et al., 2021), *i.e.*, as *direct multi-step forecasting* (DMS) Chevillon (2007). Thus, while effective at specific fore-

casting tasks, their setups are not obviously applicable to a broad range of time series settings (such as forecasting arbitrary horizon lengths, or generalizing to classification or regression tasks).

Moreover, Zeng et al. (2022) showed that simpler alternatives to Transformers, such as data normalization plus a single linear layer (NLinear), can outperform these specialized Transformer architectures when similarly trained to predict the entire fixed forecasting horizons. Their results suggest that neither the attention mechanism nor the proposed modifications of these time series Transformers may be best suited for time series modeling. Instead, the success of these prior works may just be from learning to forecast the entire horizon with fully connected dependencies between prior time-step inputs and future time-step outputs, where a fully connected linear layer is sufficient.

**Other deep learning methods.** Other works also investigate pure deep learning architectures with no explicit temporal components, and show these models can also perform well on time series forecasting. Oreshkin et al. (2019) propose N-BEATS, a deep architecture based on backward and forward residual links. Even simpler, Zeng et al. (2022) investigate single linear layer models for time series forecasting. Both works show that simple architectures are capable of achieving high performance for time series forecasting. In particular, with just data normalization, the NLinear model in Zeng et al. (2022) obtained state-of-the-art performance on the popular Informer benchmark Zhou et al. (2021). Given an input sequence of past lag terms and a target output sequence of future horizon terms, for every horizon output their model simply learns the fully connected dependencies between that output and every input lag sample. However, FCNs such as NLinear also carry inefficient downsides. Unlike Transformers and SSM-based models, the number of parameters for FCNs scales directly with input and output sequence length, *i.e.*, $\mathcal{O}(\ell h)$ for $\ell$ inputs and $h$ outputs. Meanwhile, SPACETIME shows that the SSM can improve the modeling quality of deep architectures, while maintaining constant parameter count regardless of input or output length. Especially when forecasting long horizons, we achieve higher forecasting accuracy with smaller models.

## B  PROOFS AND THEORETICAL DISCUSSION

### B.1  EXPRESSIVITY RESULTS

**Proposition 1.** *An SSM with a companion state matrix can represent*

 i.  ○ *ARIMA (Box et al., 1970)*

 ii.  ○ *Exponential smoothing*

 iii.  ○ *Controllable LTI systems (Chen, 1984)*

*Proof of Proposition 1.* We show each case separately. We either provide a set of algebraic manipulations to obtain the desired model from a companion SSM, or alternatively invoke standard results from signal processing and system theory.

*i.* ○ We start with a standard ARMA$(p, q)$ model

$$y_k = u_k + \sum_{i=1}^{q} \theta_i u_{k-i} + \sum_{i=1}^{p} \phi_i y_{k-i} p_i$$

We consider two cases:

**Case (1): Outputs $y$ are a shifted (lag–1) version of the inputs $u$**

$$
\begin{aligned}
y_{k+1} &= y_k + \sum_{i=1}^{q} \theta_i y_{k-i} + \sum_{i=1}^{p} \phi_i y_{k-i+1} p_i \\
&= (1 + \phi_1 y_k) + \sum_{i=1}^{q} (\theta_i + \phi_{i+1}) y_{k-i} + \sum_{i=q+1}^{p} \theta_i y_{k-i}
\end{aligned}
\tag{16}
$$

where, without loss of generality, we have assumed that $p > q$ for notational convenience. The autoregressive system (16) is equivalent to

$$
\begin{bmatrix} \boldsymbol{A} & \boldsymbol{B} \\ \boldsymbol{C} & \boldsymbol{D} \end{bmatrix} = \begin{bmatrix} 0 & 0 & \dots & 0 & 0 & 1 \\ 1 & 0 & \dots & 0 & 0 & 0 \\ \vdots & \vdots & \dots & \vdots & \vdots & \vdots \\ 0 & 0 & \dots & 0 & 0 & 0 \\ 0 & 0 & \dots & 1 & 0 & 0 \\ (1+\phi_1) & (\theta_1+\phi_2) & \dots & \theta_{d-1} & \theta_d & 0 \end{bmatrix}.
$$

in state-space form, with $x \in \mathbb{R}^d$ and $d = \max(p, q)$. Note that the state-space formulation is not unique.

**Case (2): Outputs $y$ are "shaped noise".** The ARMA(p,q) formulation (classically) defines inputs $u$ as white noise samples[1], $\forall k : p(u_k)$ is a normal distribution with mean zero and some variance. In this case, we can decompose the output as follows:

$$
y_k^{\mathrm{ar}} = \sum_{i=1}^{p} \phi_i y_{k-i} p_i \qquad y_k^{\mathrm{ma}} = u_k + \sum_{i=1}^{q} \theta_i u_{k-i}
$$

such that $y_k = y_k^{\mathrm{ar}} + y_k^{\mathrm{ma}}$. The resulting state-space models are:

$$
\begin{bmatrix} \boldsymbol{A}^{\mathrm{ar}} & \boldsymbol{B}^{\mathrm{ar}} \\ \boldsymbol{C}^{\mathrm{ar}} & \boldsymbol{D}^{\mathrm{ar}} \end{bmatrix} = \begin{bmatrix} 0 & 0 & \dots & 0 & 0 & 1 \\ 1 & 0 & \dots & 0 & 0 & 0 \\ \vdots & \vdots & \dots & \vdots & \vdots & \vdots \\ 0 & 0 & \dots & 0 & 0 & 0 \\ 0 & 0 & \dots & 1 & 0 & 0 \\ \phi_1 & \phi_2 & \dots & \phi_{p-1} & \phi_p & 0 \end{bmatrix}.
$$

and

$$
\begin{bmatrix} \boldsymbol{A}^{\mathrm{ma}} & \boldsymbol{B}^{\mathrm{ma}} \\ \boldsymbol{C}^{\mathrm{ma}} & \boldsymbol{D}^{\mathrm{ma}} \end{bmatrix} = \begin{bmatrix} 0 & 0 & \dots & 0 & 0 & 1 \\ 1 & 0 & \dots & 0 & 0 & 0 \\ \vdots & \vdots & \dots & \vdots & \vdots & \vdots \\ 0 & 0 & \dots & 0 & 0 & 0 \\ 0 & 0 & \dots & 1 & 0 & 0 \\ \theta_1 & \theta_2 & \dots & \theta_{q-1} & \theta_q & 1 \end{bmatrix}.
$$

Note that $\boldsymbol{A}^{\mathrm{ar}} \in \mathbb{R}^{p \times p}$, $\boldsymbol{A}^{\mathrm{ma}} \in \mathbb{R}^{q \times q}$. More generally, our method can represent any ARMA process as the sum of two SPACETIME *heads*: one taking as input the time series itself, and one the driving signal $u$.

**ARIMA** ARIMA processes are ARMA$(p, q)$ applied to differenced time series. For example, first-order differencing $y_k = u_k - u_{k-1}$. Differencing corresponds to high–pass filtering of the signal $y$, and can be thus be realized via a convolution (Strang and Nguyen, 1996).

Any digital filter that can be expressed as a difference equation admits a state–space representation in companion form (Oppenheim, 1999), and hence can be learned by SPACETIME.

*ii.* ○ Simple exponential smoothing (SES) (Brown, 1959)

$$
y_k = \alpha y_{k-1} + \alpha(1-\alpha)y_{k-2} + \cdots + \alpha(1-\alpha)^{p-1}y_{k-p} \tag{17}
$$

is an AR process with a parametrization involving a single scalar $0 < \alpha < 1$ and can thus be represented in companion form as shown above.

*iii.* ○ Let $(\boldsymbol{A}, \boldsymbol{B}, \boldsymbol{C})$ be any controllable linear system. Controllability corresponds to invertibility of the Krylov matrix (Chen, 1984, Thm 6.1, p145)

$$
\mathcal{K}(\boldsymbol{A}, \boldsymbol{B}) = [\boldsymbol{B}, \boldsymbol{A}\boldsymbol{B}, \dots, \boldsymbol{A}^{d-1}\boldsymbol{B}], \qquad \mathcal{K}(\boldsymbol{A}, \boldsymbol{B}) \in \mathbb{R}^{d \times d}.
$$

From $\mathrm{rank}(\mathcal{K}) = d$, it follows that there exists a $\boldsymbol{a} \in \mathbb{R}^d$

$$
a_0 \boldsymbol{B} + a_1 \boldsymbol{A}\boldsymbol{B} + \cdots + a_{d-1} \boldsymbol{A}^{d-1}\boldsymbol{B} + \boldsymbol{A}^d \boldsymbol{B} = 0.
$$

---

[1]Other formulations with forecast residuals are also common.

Thus

$$\begin{aligned}
A\mathcal{K} &= [AB, A^2B, \ldots, A^dB] \\
&= [\underbrace{AB, A^2B, \ldots, A^{d-1}B}_{\text{column left shift of } \mathcal{K}}, \quad \underbrace{-(a_0B + a_1Ab + \cdots + a_{d-1}A^{d-1}B)}_{\text{linear combination, columns of } \mathcal{K}}] \\
&= \mathcal{K}(S^f - ae_{d-1}^\top)
\end{aligned}$$

where $G = (S^f - ae_{d-1}^\top)$ is a companion matrix.

$$A\mathcal{K} = \mathcal{K}G \iff G = \mathcal{K}^{-1}A\mathcal{K}.$$

Therefore $G$ is similar to $A$. We can then construct a companion form state space $(G, B, C, D)$ from $A$ using the relation above.

$\square$

**Proposition 2.** *No class of continuous-time LSSL SSMs can represent the noiseless AR(p) process.*

*Proof of Proposition 2.* Recall from Sec. 3.1.1 that a noiseless AR($p$) process is defined by

$$y_t = \sum_{i=1}^{p} \phi_i y_{t-i} = \phi_1 y_{t-1} + \ldots + \phi_p y_{t-p} \tag{18}$$

with coefficients $\phi_1, \ldots, \phi_p$. This is represented by the SSM

$$x_{t+1} = Sx_t + Bu_t \tag{19}$$
$$y_t = Cx_t + Du_t \tag{20}$$

when $S \in \mathbb{R}^{p \times p}$ is the shift matrix, $B \in \mathbb{R}^{p \times 1}$ is the first basis vector $e_1$, $C \in \mathbb{R}^{1 \times p}$ is a vector of coefficients $\phi_1, \ldots, \phi_p$, and $D = 0$, *i.e.*,

$$S = \begin{bmatrix} 0 & 0 & \ldots & 0 & 0 \\ 1 & 0 & \ldots & 0 & 0 \\ 0 & 1 & \ldots & 0 & 0 \\ \vdots & & \ddots & \vdots & \vdots \\ 0 & 0 & \ldots & 1 & 0 \end{bmatrix}, \quad B = \begin{bmatrix} 1 & 0 & \ldots & 0 \end{bmatrix}^T, \quad C = \begin{bmatrix} \phi_1 & \ldots & \phi_p \end{bmatrix} \tag{21}$$

We prove by contradiction that a continuous-time LSSL SSM cannot represent such a process. Consider the following solutions to a continuous-time system and a system (18), both in autonomous form

$$x_{t+1}^{\text{cont}} = e^A x_t \qquad x_{t+1}^{\text{disc}} = Sx_t.$$

It follows

$$\begin{aligned}
x_{t+1}^{\text{cont}} = x_{t+1}^{\text{disc}} &\iff e^A = S \\
&\iff A = \log(S).
\end{aligned}$$

we have reached a contradiction by (Culver, 1966, Theorem 1), as $S$ is singular by definition and thus its matrix logarithm does not exist.

$\square$

## B.2 EFFICIENCY RESULTS

We first prove that Algorithm 1 yields the correct output filter $F^y$. We then analyze its time complexity, showing that it takes time $O(\ell \log \ell + d \log d)$ for sequence length $\ell$ and state dimension $d$.

**Theorem 1.** *Algorithm 1 returns the filter $F^y = (CB, \ldots, CA^{\ell-1}B)$.*

*Proof.* We follow the outline of the proof in Section 3.3. Instead of computing $\boldsymbol{F}^y$ directly, we compute its spectrum (its discrete Fourier transform):

$$\tilde{\boldsymbol{F}}^y[m] := \mathcal{F}(\boldsymbol{F}^y) = \sum_{j=0}^{\ell-1} \boldsymbol{C}\boldsymbol{A}^j \omega^{mj} \boldsymbol{B} = \boldsymbol{C}(\boldsymbol{I}-\boldsymbol{A}^\ell)(\boldsymbol{I}-\boldsymbol{A}\omega^m)^{-1}\boldsymbol{B} = \tilde{\boldsymbol{C}}(\boldsymbol{I}-\boldsymbol{A}\omega^m)^{-1}\boldsymbol{B}, \quad m = 0, 1, \ldots, \ell-1.$$

where $\omega = \exp(-2\pi i/\ell)$ is the $\ell$-th root of unity.

This reduces to computing the quadratic form of the resolvent $(\boldsymbol{I} - \boldsymbol{A}\omega^m)^{-1}$ on the roots of unity (the powers of $\omega$). Since $\boldsymbol{A}$ is a companion matrix, we can write $\boldsymbol{A}$ as a shift matrix plus a rank-1 matrix, $\boldsymbol{A} = \boldsymbol{S} + a e_d^T$, where $e_d$ is the $d$-th basis vector $[0, \ldots, 0, 1]$ and the shift matrix $\boldsymbol{S}$ is:

$$\boldsymbol{S} = \begin{bmatrix} 0 & 0 & \ldots & 0 & 0 \\ 1 & 0 & \ldots & 0 & 0 \\ 0 & 1 & \ldots & 0 & 0 \\ \vdots & \vdots & \ddots & \vdots & \vdots \\ 0 & 0 & \ldots & 1 & 0 \end{bmatrix}.$$

Thus Woodbury's matrix identity (i.e., Sherman–Morrison formula) yields:

$$\begin{aligned} (\boldsymbol{I} - \boldsymbol{A}\omega^m)^{-1} &= (\boldsymbol{I} - \omega^m \boldsymbol{S} - \omega^m a e_d^\top)^{-1} \\ &= (\boldsymbol{I} - \omega^m \boldsymbol{S})^{-1} + \frac{(\boldsymbol{I} - \omega^m \boldsymbol{S})^{-1} \omega^m a e_d^\top (\boldsymbol{I} - \omega^m \boldsymbol{S})^{-1}}{1 - \omega^m e_d^\top (\boldsymbol{I} - \omega^m \boldsymbol{S})^{-1} a}. \end{aligned}$$

This is the resolvent of the shift matrix $(\boldsymbol{I} - \omega^m \boldsymbol{S})^{-1}$, with a rank-1 correction. Hence

$$\tilde{\boldsymbol{F}}^y = \tilde{\boldsymbol{C}}(\boldsymbol{I} - \omega^m \boldsymbol{S})^{-1}\boldsymbol{B} + \frac{\tilde{\boldsymbol{C}}(\boldsymbol{I} - \omega^m \boldsymbol{S})^{-1} a e_d^\top (\boldsymbol{I} - \omega^m \boldsymbol{S})^{-1}\boldsymbol{B}}{\omega^{-m} - e_d^\top (\boldsymbol{I} - \omega^m \boldsymbol{S})^{-1} a}. \tag{22}$$

We now need to derive how to compute the quadratic form of a resolvent of the shift matrix efficiently. Fortunately the resolvent of the shift matrix has a very special structure that closely relates to the Fourier transform. We show analytically that:

$$(\boldsymbol{I} - \omega^m \boldsymbol{S})^{-1} = \begin{bmatrix} 1 & 0 & \ldots & 0 & 0 \\ \omega^m & 1 & \ldots & 0 & 0 \\ \omega^{2m} & \omega^m & \ldots & 0 & 0 \\ \vdots & \vdots & \ddots & \vdots & \vdots \\ \omega^{(d-1)m} & \omega^{(d-2)m} & \ldots & \omega^m & 1 \end{bmatrix}.$$

It is easy to verify by multiplying this matrix with $I - \omega^m S$ to see if we obtain the identity matrix. Recall that multiplying with $S$ on the left just shifts all the columns down by one index. Therefore:

$$
\begin{bmatrix}
1 & 0 & \dots & 0 & 0 \\
\omega^m & 1 & \dots & 0 & 0 \\
\omega^{2m} & \omega^m & \dots & 0 & 0 \\
\vdots & \vdots & \ddots & \vdots & \vdots \\
\omega^{(d-1)m} & \omega^{(d-2)m} & \dots & \omega^m & 1
\end{bmatrix}
(I - \omega^m S)
$$

$$
=
\begin{bmatrix}
1 & 0 & \dots & 0 & 0 \\
\omega^m & 1 & \dots & 0 & 0 \\
\omega^{2m} & \omega^m & \dots & 0 & 0 \\
\vdots & \vdots & \ddots & \vdots & \vdots \\
\omega^{(d-1)m} & \omega^{(d-2)m} & \dots & \omega^m & 1
\end{bmatrix}
- \omega^m S
\begin{bmatrix}
1 & 0 & \dots & 0 & 0 \\
\omega^m & 1 & \dots & 0 & 0 \\
\omega^{2m} & \omega^m & \dots & 0 & 0 \\
\vdots & \vdots & \ddots & \vdots & \vdots \\
\omega^{(d-1)m} & \omega^{(d-2)m} & \dots & \omega^m & 1
\end{bmatrix}
$$

$$
=
\begin{bmatrix}
1 & 0 & \dots & 0 & 0 \\
\omega^m & 1 & \dots & 0 & 0 \\
\omega^{2m} & \omega^m & \dots & 0 & 0 \\
\vdots & \vdots & \ddots & \vdots & \vdots \\
\omega^{(d-1)m} & \omega^{(d-2)m} & \dots & \omega^m & 1
\end{bmatrix}
- \omega^m
\begin{bmatrix}
0 & 0 & \dots & 0 & 0 \\
1 & 0 & \dots & 0 & 0 \\
\omega^m & 1 & \dots & 0 & 0 \\
\vdots & \vdots & \ddots & \vdots & \vdots \\
\omega^{(d-2)m} & \omega^{(d-3)m} & \dots & 1 & 0
\end{bmatrix}
$$

$$
=
\begin{bmatrix}
1 & 0 & \dots & 0 & 0 \\
\omega^m & 1 & \dots & 0 & 0 \\
\omega^{2m} & \omega^m & \dots & 0 & 0 \\
\vdots & \vdots & \ddots & \vdots & \vdots \\
\omega^{(d-1)m} & \omega^{(d-2)m} & \dots & \omega^m & 1
\end{bmatrix}
-
\begin{bmatrix}
0 & 0 & \dots & 0 & 0 \\
\omega^m & 0 & \dots & 0 & 0 \\
\omega^{2m} & \omega^m & \dots & 0 & 0 \\
\vdots & \vdots & \ddots & \vdots & \vdots \\
\omega^{(d-1)m} & \omega^{(d-2)m} & \dots & \omega & 0
\end{bmatrix}
$$

$$
= I.
$$

Thus the resolvent of the shift matrix indeed has the form of a lower-triangular matrix containing the roots of unity.

Now that we have the analytic formula of the resolvent, we can derive its quadratic form, given some vectors $u, v \in \mathbb{R}^d$. Substituting in, we have

$$
u^T (I - \omega^m S)^{-1} v = u_1 v_1 + u_2 v_1 \omega^m + u_2 v_2 + u_3 v_1 \omega^{2m} + u_3 v_2 \omega^m + u_3 v_1 + \dots .
$$

Grouping terms by powers of $\omega$, we see that we want to compute $u_1 v_1 + u_2 v_2 + \dots + u_d v_d$, then $u_2 v_1 + u_3 v_2 + \dots + u_d v_{d-1}$, and so on. The term corresponding to $\omega^{km}$ is exactly the $k$-th element of the linear convolution $u * v$. Define $q = u * v$, then $u^T (I - \omega^m S)^{-1} v$ is just the Fourier transform of $u * v$. To deal with the case where $d > \ell$, we note that the powers of roots of unity will repeat, so we just need to extend the output of $u * v$ to be multiples of $\ell$, then split them into chunk of size $\ell$, then sum them up and take the length-$\ell$ Fourier transform. This is exactly the procedure $\mathrm{quad}(u, v)$ defined in Algorithm 1.

Once we have derived the quadratic form of the resolvent $(I - \omega^m S)^{-1}$, simply plugging it into the Woodbury's matrix identity (Equation (22)) yields Algorithm 1.

$\square$

We analyze the algorithm's complexity.

**Theorem 2.** *Algorithm 1 has time complexity $O(\ell \log \ell + d \log d)$ for sequence length $\ell$ and state dimension $d$.*

*Proof.* We see that computing the quadratic form of the resolvent $(I - \omega^m S)^{-1}$ involves a linear convolution of size $d$ and a Fourier transform of size $\ell$. The linear convolution can be done by performing an FFT of size $2d$ on both inputs, multiply them pointwise, then take the inverse FFT of size $2d$. This has time complexity $O(d \log d)$. The Fourier transform of size $\ell$ has time complexity $O(\ell \log \ell)$.

The whole algorithm needs to compute four such quadratic form, hence it takes time $O(\ell \log \ell + d \log d)$.

$\square$

**Remark.** We see that the algorithm easily extends to the case where the matrix $\boldsymbol{A}$ is a companion matrix plus low-rank matrix (of some rank $k$). We can write $\boldsymbol{A}$ as the sum of the shift matrix and a rank-$(k+1)$ matrix (since $\boldsymbol{A}$ itself is the sum of a shift matrix and a rank-1 matrix). Using the same strategy, we can use the Woodbury's matrix identity for the rank-$(k+1)$ case. The running time will then scale as $O(k(\ell \log \ell + d \log d))$.

### B.3 COMPANION MATRIX STABILITY

**Normalizing companion parameters for bounded gradients**

**Proposition 3** (Bounded SPACETIME Gradients). *Given $s$, the norm of the gradient of a* SPACE-TIME *layer is bounded for all $k < s$ if*

$$\sum_{i=0}^{d-1} |\boldsymbol{a}_i| = 1$$

*Proof.* Without loss of generality, we assume $x_0 = 0$. Since the solution at time $s$ is

$$y_s = \boldsymbol{C} \sum_{i-1}^{s-1} \boldsymbol{A}^{s-i-1} \boldsymbol{B} u_i$$

we compute the gradient w.r.t $u_k$ as

$$\frac{\mathrm{d}y_s}{\mathrm{d}u_k} = \boldsymbol{C}\boldsymbol{A}^{s-k-1}\boldsymbol{B}. \tag{23}$$

The largest eigenvalue of $\boldsymbol{A}$

$\max\{\texttt{eig}(\boldsymbol{A})\} =$

$$\leq \max\Big\{1, \sum_{i=0}^{d-1}|\boldsymbol{a}_i|\Big\} \qquad \text{Corollary of Gershgorin (Hirst and Macey, 1997, Theorem 1)}$$

$$= 1 \qquad\qquad \text{using } \sum_{i}|\boldsymbol{a}_i| = 1$$

is 1, which implies convergence of the operator $\boldsymbol{C}\boldsymbol{A}^{s-k-1}\boldsymbol{B}$. Thus, the gradients are bounded.

$\square$

We use the proposition above to ensure gradient boundedness in SPACETIME layers by normalizing $\boldsymbol{a}$ every forward pass.

## C EXPERIMENT DETAILS

### C.1 INFORMER FORECASTING

**Dataset details**. In Table 1, we evaluate all methods with datasets and horizon tasks from the Informer benchmark (Zhou et al., 2021). We use the datasets and horizons evaluated on in recent works (Wu et al., 2021; Zhou et al., 2022b;a; Zeng et al., 2022), which evaluate on electricity transformer temperature time series (ETTh1, ETTh2, ETTm1, ETTm2) with forecasting horizons $\{96, 192, 336, 720\}$. We extend this comparison in Appendix D.2 to all datasets and forecasting horizons in Zhou et al. (2021), which also consider weather and electricity (ECL) time series data.

**Training details.** We train SPACETIME on all datasets for 50 epochs using AdamW optimizer (Loshchilov and Hutter, 2017), cosine scheduling, and early stopping based on best validation standardized MSE. We performed a grid search over number of SSMs $\{64, 128\}$ and weight decay $\{0, 0.0001\}$. Like prior forecasting works, we treat the input lag sequence as a hyperparameter, and train to predict each forecasting horizon with either 336 or 720 time-step-long input sequences for all datasets and horizons. For all datasets, we use a 3-layer SPACETIME network with 128 SSMs per layer. We train with learning rate 0.01, weight decay 0.0001, batch size 32, and dropout 0.25.

**Hardware details.** All experiments were run on a single NVIDIA Tesla P100 GPU.

## C.2 MONASH FORECASTING

The Monash Time Series Forecasting Repository (Godahewa et al., 2021) provides an extensive benchmark suite for time series forecasting models, with over 30 datasets (including various configurations) spanning finance, traffic, weather and medical domains. We compare SPACETIME against 13 baselines provided by the Monash benchmark: simple exponential smoothing (SES) (Gardner Jr, 1985), Theta (Assimakopoulos and Nikolopoulos, 2000), TBATS (De Livera et al., 2011), ETS (Winters, 1960), DHR–ARIMA (Hyndman and Athanasopoulos, 2018), Pooled Regression (PR) (Trapero et al., 2015), CatBoost (Dorogush et al., 2018), FFNN, DeepAR (Salinas et al., 2020), N-BEATS Oreshkin et al. (2019), WaveNet (Oord et al., 2016), vanilla Transformer (Vaswani et al., 2017). A complete list of the datasets considered and baselines, including test results (average RMSE across 3 seeded runs) is available in Table 20.

**Training details.** We optimize SPACETIME on all datasets using Adam optimizer for 40 epochs with a linear learning rate warmup phase of 20 epochs and cosine decay. We initialize learning rate at 0.001, reach 0.004 after warmup, and decay to 0.0001. We do not use weight decay or dropout.

We perform a grid search over number of layers $\{3, 4, 5, 6\}$, number of SSMs per layer $\{8, 16, 32, 64, 128\}$ and number of channels (width of the model) $\{1, 4, 8, 16\}$. Hyperparameter tuning is performed for each dataset. We pick the model based on best validation RMSE performance.

**Hardware details.** All experiments were run on a single NVIDIA GeForce RTX 3090 GPU.

## C.3 TIME SERIES CLASSIFICATION

**ECG classification (motivation and dataset description).** Electrocardiograms (ECG) are commonly used as one of the first examination tools for assessing and diagnosing cardiovascular diseases, which are a major cause of mortality around the world (Amini et al., 2021). However, ECG interpretation remains a challenging task for cardiologists and general practitioners (Jablonover et al., 2014; Cook et al., 2020). Incorrect interpretation of ECG can result in misdiagnosis and delayed treatment, which can be potentially life-threatening in critical situations such as emergency rooms, where an accurate interpretation is needed quickly.

To mitigate these challenges, deep learning approaches are increasingly being applied to interpret ECGs. These approaches have been used for predicting the ECG rhythm class (Hannun et al., 2019), detecting atrial fibrillation (Attia et al., 2019b), rare cardiac diseases like cardiac amyloidosis (Goto et al., 2021), and a variety of other abnormalities (Attia et al., 2019a; Siontis et al., 2021). Deep learning approaches have shown preliminary promise in matching the performance of cardiologists and emergency residents in triaging ECGs, which would permit accurate interpretations in settings where specialists may not be present (Ribeiro et al., 2020; Hannun et al., 2019).

We use the publicly available PTB-XL dataset (Wagner et al., 2020a;b; Goldberger et al., 2000), which contains 21,837 12-lead ECG recordings of 10 seconds each obtained from 18,885 patients. Each ECG recording is annotated by up to two cardiologists with one or more of the 71 ECG statements (labels). These ECG statements conform to the SCP-ECG standard (Secretary, 2009). Each statement belongs to one or more of the following three categories – diagnostic, form, and rhythm statements. The diagnostic statements are further organised in a hierarchy containing 5 superclasses and 24 subclasses.

This provides six sets of annotations for the ECG statements based on the different categories and granularities: all (all ECG statements), diagnostic (only diagnostic statements including both subclass and superclass statements), diagnostic subclass (only diagnostic subclass statements), diagnostic superclass (only diagnostic superclass statements), form (only form statements), and rhythm (only rhythm statements). These six sets of annotations form different prediction tasks which are referred to as all, diag, sub-diag, super-diag, form, and rhythm respectively. The diagnostic superclass task is multi-class classification, and the other tasks are multi-label classification.

**ECG classification training details.** To tune SPACETIME and S4, we performed a grid search over the learning rate $\{0.01, 0.001\}$, model dropout $\{0.1, 0.2\}$, number of SSMs per layer $\{128, 256\}$, and number of layers $\{4, 6\}$, and chose the parameters that resulted in highest validation AUROC. The SSM state dimension was fixed to 64, with gated linear units as the non-linearity between

stacked layers. We additionally apply layer normalization. We use a cosine learning rate scheduler, with a warmup period of 5 epochs. We train all models for 100 epochs.

**Speech Commands training details.** To train SPACETIME, we use the same hyperparameters used by S4: a learning rate of 0.01 with a plateau scheduler with patience 20, dropout of 0.1, 128 SSMs per layer, 6 layers, batch normalization, trained for 200 epochs.

**Hardware details.** For both ECG and Speech Commands, all experiments were run on a single NVIDIA Tesla A100 Ampere 40 GB GPU.

# D EXTENDED EXPERIMENTAL RESULTS

## D.1 EXPRESSIVITY ON DIGITAL FILTERS

We experimentally verify whether SPACETIME can approximate the input–output map of digital filter admitting a state–space representation, with improved generalization over baseline models given test inputs of unseen frequencies.

We generate a dataset of 1028 sinusoidal signals of length 200

$$x(t) = \sin(2\pi\omega t)$$

where $\omega \in [2, 40] \bigcup [50, 100]$ in the training set and $\omega \in (40, 50)$ in the test set. The outputs are obtained by filtering $x$, i.e., $y = \mathcal{F}(x)$ where $\mathcal{F}$ is in the family of digital filters.

We introduce common various sequence-to-sequence layers or models as baselines: the original S4 diagonal plus low–rank (Gu et al., 2021a), a single-layer LSTM, a single 1d convolution (Conv1d), a dense linear layer (NLinear), a single self–attention layer. All models are trained for 800 epochs with batch size 256, learning rate $10^{-3}$ and Adam. We repeat this experiment for digital filters of different orders (Oppenheim, 1999). The results are shown in Figure 8. SPACETIME learns to match the frequency response of the target filter, producing the correct output for inputs at test frequencies.

Table 6: Comparing sequence models on the task of approximating the input–output map defined by digital filters of different orders. Test RMSE on held-out inputs at unseen frequencies.

| Filter | Order | SPACETIME | S4 | Conv1D | LSTM | NLinear | Transformer |
|---|---|---|---|---|---|---|---|
| Butterworth | 2 | 0.0055 | 0.0118 | 0.0112 | 0.0115 | 1.8420 | 0.5535 |
| | 3 | 0.0057 | 0.3499 | 0.0449 | 0.0231 | 1.7085 | 0.6639 |
| | 10 | 0.0039 | 0.8077 | 0.4747 | 0.2753 | 1.5162 | 0.7191 |
| Chebyshev 1 | 2 | 0.0187 | 0.0480 | 0.0558 | 0.0285 | 1.9313 | 0.2452 |
| | 3 | 0.0055 | 0.0467 | 0.0615 | 0.0178 | 1.8077 | 0.4028 |
| | 10 | 0.0620 | 0.6670 | 0.1961 | 0.1463 | 1.5069 | 0.7925 |
| Chebyshev 2 | 2 | 0.0112 | 0.0121 | 0.0067 | 0.0019 | 0.4101 | 0.0030 |
| | 3 | 0.0201 | 0.0110 | 0.0771 | 0.0102 | 0.4261 | 0.0088 |
| | 10 | 0.0063 | 0.6209 | 0.3361 | 0.1911 | 1.5584 | 0.7936 |
| Elliptic | 2 | 0.0001 | 0.0300 | 0.0565 | 0.0236 | 1.9150 | 0.2445 |
| | 3 | 0.0671 | 0.0868 | 0.0551 | 0.0171 | 1.8782 | 0.4198 |
| | 10 | 0.0622 | 0.0909 | 0.1352 | 0.1344 | 1.4901 | 0.7368 |

## D.2 INFORMER FORECASTING

**Univariate long horizon forecasts with Informer splits.** Beyond the ETT datasets and horizons evaluated on in Table 7, we also compare SPACETIME to alternative time series methods on the complete datasets and horizons used in the original Informer paper (Zhou et al., 2021). We compare against recent architectures which similarly evaluate on these settings, including ETSFormer (Woo et al., 2022), SCINet (Liu et al., 2021), and Yformer (Madhusudhanan et al., 2021), and other comparison methods found in the Informer paper, such as Reformer (Kitaev et al., 2020) and ARIMA. SPACETIME obtains best results on 20 out of 25 settings, the most of any method.

Table 7: **Univariate forecasting** results on Informer datasets. **Best** results in **bold**. SPACETIME obtains best MSE on 19 out of 25 and best MAE on 20 out of 25 dataset and horizon tasks.

| Methods | | SPACETIME | | ETSFormer | | SCINet | | S4 | | Yformer | | Informer | | LogTrans | | Reformer | | N-BEATS | | DeepAR | | ARIMA | | Prophet | |
|---|---|---|---|---|---|---|---|---|---|---|---|---|---|---|---|---|---|---|---|---|---|---|---|---|---|
| Metric | | MSE | MAE | MSE | MAE | MSE | MAE | MSE | MAE | MSE | MAE | MSE | MAE | MSE | MAE | MSE | MAE | MSE | MAE | MSE | MAE | MSE | MAE | MSE | MAE |
| ETTh1 | 24 | **0.026** | **0.124** | 0.031 | 0.132 | 0.031 | 0.132 | 0.061 | 0.191 | 0.082 | 0.230 | 0.098 | 0.247 | 0.103 | 0.259 | 0.222 | 0.389 | 0.042 | 0.156 | 0.107 | 0.280 | 0.108 | 0.284 | 0.115 | 0.275 |
| | 48 | **0.038** | **0.153** | 0.041 | 0.154 | 0.051 | 0.173 | 0.079 | 0.220 | 0.139 | 0.308 | 0.158 | 0.319 | 0.167 | 0.328 | 0.284 | 0.445 | 0.065 | 0.200 | 0.162 | 0.327 | 0.175 | 0.424 | 0.168 | 0.330 |
| | 168 | 0.066 | 0.209 | **0.065** | **0.203** | 0.081 | 0.222 | 0.104 | 0.258 | 0.111 | 0.268 | 0.183 | 0.346 | 0.207 | 0.375 | 1.522 | 1.191 | 0.106 | 0.255 | 0.239 | 0.422 | 0.396 | 0.504 | 1.224 | 0.763 |
| | 336 | **0.069** | **0.212** | 0.071 | 0.215 | 0.094 | 0.242 | 0.080 | 0.229 | 0.195 | 0.365 | 0.222 | 0.387 | 0.230 | 0.398 | 1.860 | 1.124 | 0.127 | 0.284 | 0.445 | 0.552 | 0.468 | 0.593 | 1.549 | 1.820 |
| | 720 | **0.075** | **0.226** | 0.079 | 0.227 | 0.176 | 0.343 | 0.116 | 0.271 | 0.226 | 0.394 | 0.269 | 0.435 | 0.273 | 0.463 | 2.112 | 1.436 | 0.269 | 0.422 | 0.658 | 0.707 | 0.659 | 0.766 | 2.735 | 3.253 |
| ETTh2 | 24 | **0.064** | **0.189** | 0.087 | 0.232 | 0.070 | 0.194 | 0.095 | 0.234 | 0.082 | 0.221 | 0.093 | 0.240 | 0.102 | 0.255 | 0.263 | 0.437 | 0.078 | 0.210 | 0.098 | 0.263 | 3.554 | 0.445 | 0.199 | 0.381 |
| | 48 | **0.095** | **0.230** | 0.112 | 0.263 | 0.102 | 0.242 | 0.191 | 0.346 | 0.172 | 0.334 | 0.155 | 0.314 | 0.169 | 0.348 | 0.458 | 0.545 | 0.123 | 0.271 | 0.163 | 0.341 | 3.190 | 0.474 | 0.304 | 0.462 |
| | 168 | **0.144** | **0.300** | 0.169 | 0.325 | 0.157 | 0.311 | 0.167 | 0.333 | 0.174 | 0.337 | 0.232 | 0.389 | 0.246 | 0.422 | 1.029 | 0.879 | 0.244 | 0.393 | 0.255 | 0.414 | 2.800 | 0.595 | 2.145 | 1.068 |
| | 336 | **0.169** | **0.333** | 0.216 | 0.379 | 0.177 | 0.340 | 0.189 | 0.361 | 0.224 | 0.391 | 0.263 | 0.417 | 0.267 | 0.437 | 1.668 | 1.228 | 0.270 | 0.418 | 0.604 | 0.607 | 2.753 | 0.738 | 2.096 | 2.543 |
| | 720 | 0.188 | **0.352** | 0.226 | 0.385 | 0.253 | 0.403 | **0.187** | 0.358 | 0.211 | 0.382 | 0.277 | 0.431 | 0.303 | 0.493 | 2.030 | 1.721 | 0.281 | 0.432 | 0.429 | 0.580 | 2.878 | 1.044 | 3.355 | 4.664 |
| ETTm1 | 24 | **0.010** | **0.074** | 0.013 | 0.084 | 0.019 | 0.088 | 0.024 | 0.117 | 0.024 | 0.118 | 0.030 | 0.137 | 0.065 | 0.202 | 0.095 | 0.228 | 0.031 | 0.117 | 0.091 | 0.243 | 0.090 | 0.206 | 0.120 | 0.290 |
| | 48 | **0.019** | **0.101** | 0.020 | 0.107 | 0.045 | 0.143 | 0.051 | 0.174 | 0.048 | 0.173 | 0.069 | 0.203 | 0.078 | 0.220 | 0.249 | 0.390 | 0.056 | 0.168 | 0.219 | 0.362 | 0.179 | 0.306 | 0.133 | 0.305 |
| | 96 | **0.026** | **0.121** | 0.030 | 0.132 | 0.072 | 0.198 | 0.086 | 0.229 | 0.143 | 0.311 | 0.194 | 0.372 | 0.199 | 0.386 | 0.920 | 0.767 | 0.095 | 0.234 | 0.364 | 0.496 | 0.272 | 0.399 | 0.194 | 0.396 |
| | 288 | **0.051** | **0.176** | 0.053 | 0.179 | 0.117 | 0.266 | 0.160 | 0.327 | 0.150 | 0.316 | 0.401 | 0.554 | 0.411 | 0.572 | 1.108 | 1.245 | 0.157 | 0.311 | 0.948 | 0.795 | 0.462 | 0.558 | 0.452 | 0.574 |
| | 672 | 0.078 | 0.220 | **0.075** | **0.214** | 0.180 | 0.328 | 0.292 | 0.466 | 0.305 | 0.476 | 0.512 | 0.644 | 0.598 | 0.702 | 1.793 | 1.528 | 0.207 | 0.370 | 2.437 | 1.352 | 0.639 | 0.697 | 2.747 | 1.174 |
| Weather | 24 | **0.088** | **0.205** | - | - | - | - | 0.125 | 0.254 | - | - | 0.117 | 0.251 | 0.136 | 0.279 | 0.231 | 0.401 | - | - | 0.128 | 0.274 | 0.219 | 0.355 | 0.302 | 0.433 |
| | 48 | **0.134** | **0.258** | - | - | - | - | 0.181 | 0.305 | - | - | 0.178 | 0.318 | 0.206 | 0.356 | 0.328 | 0.423 | - | - | 0.203 | 0.353 | 0.273 | 0.409 | 0.445 | 0.536 |
| | 168 | 0.221 | 0.349 | - | - | - | - | **0.198** | **0.333** | - | - | 0.266 | 0.398 | 0.309 | 0.439 | 0.654 | 0.634 | - | - | 0.293 | 0.451 | 0.503 | 0.599 | 2.441 | 1.142 |
| | 336 | **0.268** | **0.380** | - | - | - | - | 0.300 | 0.417 | - | - | 0.297 | 0.416 | 0.359 | 0.484 | 1.792 | 1.093 | - | - | 0.585 | 0.644 | 0.728 | 0.730 | 1.987 | 2.468 |
| | 720 | 0.345 | 0.451 | - | - | - | - | **0.245** | **0.375** | - | - | 0.359 | 0.466 | 0.388 | 0.499 | 2.087 | 1.534 | - | - | 0.499 | 0.596 | 1.062 | 0.943 | 3.859 | 1.144 |
| ECL | 48 | **0.184** | **0.306** | - | - | - | - | 0.222 | 0.350 | 0.194 | 0.322 | 0.239 | 0.359 | 0.280 | 0.429 | 0.971 | 0.884 | - | - | 0.204 | 0.357 | 0.879 | 0.764 | 0.524 | 0.595 |
| | 168 | **0.250** | **0.353** | - | - | - | - | 0.331 | 0.421 | 0.260 | 0.361 | 0.447 | 0.503 | 0.454 | 0.529 | 1.671 | 1.587 | - | - | 0.315 | 0.436 | 1.032 | 0.833 | 2.725 | 1.273 |
| | 336 | 0.288 | 0.382 | - | - | - | - | 0.328 | 0.422 | **0.269** | **0.375** | 0.489 | 0.528 | 0.514 | 0.563 | 3.528 | 2.196 | - | - | 0.414 | 0.519 | 1.136 | 0.876 | 2.246 | 3.077 |
| | 720 | **0.355** | **0.446** | - | - | - | - | 0.428 | 0.494 | 0.427 | 0.479 | 0.540 | 0.571 | 0.558 | 0.609 | 4.891 | 4.047 | - | - | 0.563 | 0.595 | 1.251 | 0.933 | 4.243 | 1.415 |
| | 960 | **0.393** | **0.478** | - | - | - | - | 0.432 | 0.497 | 0.595 | 0.573 | 0.582 | 0.608 | 0.624 | 0.645 | 7.019 | 5.105 | - | - | 0.657 | 0.683 | 1.370 | 0.982 | 6.901 | 4.260 |
| Count | | **19** | **20** | 2 | 2 | 0 | 0 | 3 | 2 | 1 | 1 | 0 | 0 | 0 | 0 | 0 | 0 | 0 | 0 | 0 | 0 | 0 | 0 | 0 | 0 |

Table 8: **Multivariate forecasting** results on Informer datasets. **Best** results in **bold**. SPACETIME obtains MSE and MAE competitive with NLinear, the prior state-of-the-art.

| Methods | | SPACETIME | | NLinear | | FiLM | | S4 | | FEDformer | | Autoformer | | Informer | |
|---|---|---|---|---|---|---|---|---|---|---|---|---|---|---|---|
| Metric | | MSE | MAE | MSE | MAE | MSE | MAE | MSE | MAE | MSE | MAE | MSE | MAE | MSE | MAE |
| ETTh1 | 720 | 0.499 | 0.480 | **0.440** | **0.453** | 0.465 | 0.472 | 1.074 | 0.814 | 0.506 | 0.507 | 0.514 | 0.512 | 1.181 | 0.865 |
| ETTh2 | 720 | 0.402 | **0.434** | **0.394** | 0.436 | 0.439 | 0.456 | 2.973 | 1.333 | 0.463 | 0.474 | 0.515 | 0.511 | 3.647 | 1.625 |
| ETTm1 | 720 | **0.408** | **0.415** | 0.433 | 0.422 | 0.420 | 0.420 | 0.738 | 0.655 | 0.543 | 0.49 | 0.671 | 0.561 | 1.166 | 0.823 |
| ETTm2 | 720 | **0.358** | **0.378** | 0.368 | 0.384 | 0.393 | 0.422 | 2.074 | 1.074 | 0.421 | 0.415 | 0.433 | 0.432 | 3.379 | 1.338 |

**Multivariate signals.** We additionally compare the performance of SPACETIME to state-of-the-art comparison methods on ETT multivariate settings. We focus on horizon length 720, the longest evaluated in prior works. In Table 8, we find SPACETIME is competitive with NLinear, which achieves best performance among compparison methods. SPACETIME also notably outperforming S4 by large margins, supporting the companion matrix representation once more.

## D.3 MONASH FORECASTING

We report the results across all datasets in Table 20. We also investigate the performance of models by aggregating datasets based on common characteristics. Concretely, we generate sets of tasks[2] based on the following properties:

- *Large dataset:* the dataset contains more than 2000 effective training samples.

- *Long context:* the models are provided a context of length greater than 20 as input.

- *Long horizon:* and the models are asked to forecast longer than 20 steps in the future.

Figure 6 shows the average $x/13$ model ranking in terms of test RMSE across splits. We contextualize SPACETIME results with best classical and deep learning methods (TBATS and DeepAR). SPACETIME relative performance is noticeably higher when context and forecasting horizons are longer, and when a larger number of samples is provided during training.

---

[2]A task can belong to multiple splits, resulting in overlapping splits. For example, a task can involve both long context as well as long forecasting horizon.

**Monash Model Rankings**

Figure 6: Relative test RMSE rankings (∗/13 models) across different slices of the 33 datasets in the Monash repository (Godahewa et al., 2021). SPACETIME sets best overall ranking across all tasks and is significantly more accurate on tasks involving long forecast horizon and larger number of training samples.

## D.4 ECG CLASSIFICATION

In addition to our results table in the main paper, we also provide the mean and standard deviations of the two models we ran in house (SPACETIME and S4) in Table 9.

Table 9: **ECG statement classification** on PTB-XL (100 Hz version). We report the mean and standard deviation over three random seeds for the three methods we ran in house.

| Task AUROC | All | Diag | Sub-diag | Super-diag | Form | Rhythm |
|---|---|---|---|---|---|---|
| SPACETIME | 93.6(0.13) | 94.1(0.12) | 93.3(0.34) | 92.9(0.09) | 88.3(0.63) | 96.7(0.05) |
| S4 | 93.8(0.38) | 93.9(0.15) | 92.9(0.11) | 93.1(0.07) | 89.5(0.66) | 97.7(0.04) |
| Transformer | 85.7(0.30) | 87.6(0.41) | 88.2(0.20) | 88.7(0.28) | 77.1(0.45) | 83.1(0.72) |

## D.5 EFFICIENCY RESULTS

We additionally empirically validate that SPACETIME trains in near-linear time with horizon sequence length. We also use synthetic data, scaling horizon from $1 - 1000$.

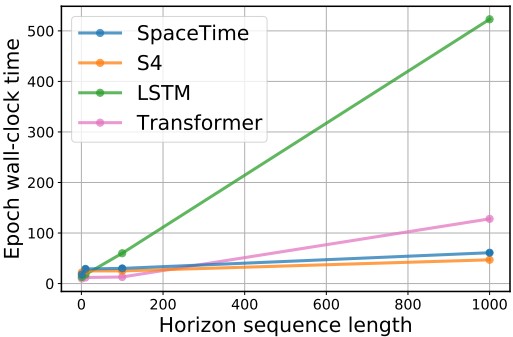

Figure 7: Train wall-clock time as we scale horizon length for SPACETIME, S4, LSTM, and Transformer.

### D.6 SPACETIME ABLATIONS

To better understand how the proposed SPACETIME SSMs lead to the improved empirical performance, we include ablations on the individual closed-loop forecasting SSM (Section 3.2) and preprocessing SSMs (Section 3.1.3).

#### D.6.1 CLOSED-LOOP FORECASTING SSM

To study how the closed-loop SSM improves long horizon forecasting accuracy, we remove the closed-loop SSM component in our default SPACETIME forecasting architecture (*c.f.*, Appendix D.7, and compare the default SPACETIME with one without any closed-loop SSMs on Informer forecasting tasks. For models without closed-loop SSMs, we replace the last layer with the standard "open-loop" SSM framework in Section 3.1.2), and keep all other layers the same. Finally, for baseline comparison against another SSM without the closed-loop component, we compare against S4.

In Table 10, we report standardized MSE on Informer ETT datasets. Adding the closed-loop SSM consistently improves forecasting accuracy, on average lowering relative MSE by 33.2%. Meanwhile, even without the closed-loop SSM, SPACETIME outperforms S4, again suggesting that the companion matrix parameterization is beneficial for autoregressive time series forecasting.

Table 10: **Closed-loop SSM Ablation** We ablate the closed-loop SSM component in SPACETIME, comparing against the prior S4 SSM on four Informer time series forecasting tasks. Removing the closed-loop SSM consistently hurts forecasting accuracy for SPACETIME.

| Method / Ablation | ETTh1 (720) | | ETTh2 (720) | | ETTm1 (720) | | ETTm2 (720) | |
|---|---|---|---|---|---|---|---|---|
| | MSE | MAE | MSE | MAE | MSE | MAE | MSE | MAE |
| SPACETIME | 0.076 | 0.222 | 0.188 | 0.352 | 0.074 | 0.213 | 0.166 | 0.318 |
| SPACETIME No Closed-loop | 0.114 | 0.271 | 0.278 | 0.431 | 0.156 | 0.310 | 0.213 | 0.365 |
| S4 (No Closed-loop) | 0.190 | 0.355 | 0.630 | 0.662 | 0.254 | 0.433 | 0.482 | 0.567 |

#### D.6.2 PREPROCESSING SSM

To study how the preprocessing SSM improves long horizon forecasting accuracy, we next compare how SPACETIME performs with and without the weight-initializing preprocessing SSMs introduced in Section 3.1.3. We compare the default SPACETIME architecture (Table 12 with (1) replacing the preprocessing SSMs with randomly initialized default companion SSMs, and (2) removing the preprocessing SSMs altogether. For the former, we preserve the number of layers, but now train the first-layer SSM weights. For the latter, there is one-less layer, but the same number of trainable parameters (as we fix and freeze the weights for each preprocessing SSM).

In Table 11, we report standardized MSE on Informer ETT datasets. We find fixing the first layer SSMs of a SPACETIME network to preprocessing SSMs consistently improves forecasting performance, achieving 4.55% lower MSE on average than the ablation with just trainable companion matrices. Including the preprocessing layer also improves MSE by 9.26% on average compared to removing the layer altogether. These results suggest that preprocessing SSMs are beneficial for time series forecasting, *e.g.*, by performing classic time series modeling techniques on input data. Unlike other approaches, SPACETIME is able to flexibly and naturally incorporate these operations into its network layers via simple weight initializations of the same general companion SSM structure.

Table 11: **Preprocessing SSM Ablation** We ablate the preprocessing SSM layer in SPACETIME, comparing against either replacing the SSMs with companion SSMs (Companion) or removing the layer (Removed). Including preprocessing SSMs consistently improves forecasting accuracy.

| Method / Ablation | ETTh1 (720) | | ETTh2 (720) | | ETTm1 (720) | | ETTm2 (720) | |
|---|---|---|---|---|---|---|---|---|
| | MSE | MAE | MSE | MAE | MSE | MAE | MSE | MAE |
| SpaceTime | 0.076 | 0.222 | 0.188 | 0.352 | 0.074 | 0.213 | 0.166 | 0.318 |
| SpaceTime No Preprocessing (Companion) | 0.076 | 0.224 | 0.194 | 0.358 | 0.079 | 0.218 | 0.182 | 0.336 |
| SpaceTime No Preprocessing (Removed) | 0.078 | 0.227 | 0.204 | 0.367 | 0.087 | 0.232 | 0.188 | 0.326 |

### D.7 SPACETIME ARCHITECTURES

We provide the specific SPACETIME architecture configurations used for forecasting and classification tasks. Each configuration follows the general architecture presented in Section 3.1 and Figure 2, and consists of repeated Multi-SSM SPACETIME layers. We first provide additional details on specific instantiations of the companion SSMs we use in our models, *e.g.*, how we instantiate preprocessing SSMs to recover specific techniques (Section 3.1.3). We then include the layer-specific details of the number and type of SSM used in each network.

#### D.7.1 SPECIFIC SSM PARAMETERIZATIONS

In Section 3.1.1, we described the general form of the companion SSM used in this work. By default, for any individual SSM we learn the $a$ column in $\boldsymbol{A}$ and the vectors $\boldsymbol{B}, \boldsymbol{C}$ as trainable parameters in a neural net module. We refer to these SSMs specifically as **companion SSMs**.

In addition, as discussed in Sections 3.1.1 and 3.1.3, we can also fix $a$, $\boldsymbol{B}$, or $\boldsymbol{C}$ to specific values to recover useful operations when computing the SSM outputs. We describe specific instantiations of the companion SSM used in our models below (with dimensionality referring to one SSM).

**Shift SSM.** We fix the $\boldsymbol{a}$ vector in the companion state matrix $\boldsymbol{A} \in \mathbb{R}^{d \times d}$ to the $\boldsymbol{0}$ vector $\in \mathbb{R}^{d}$, such that $\boldsymbol{A}$ is the shift matrix (see Eq. 21 for an example). This is a generalization of a 1-D "sliding window" convolution with fixed kernel size equal to SSM state dimension $d$. To see how, note that if $\boldsymbol{B}$ is also fixed to the first basis vector $\boldsymbol{e_1} \in \mathbb{R}^{d \times 1}$, then this exactly recovers a 1-D convolution with kernel determined by $\boldsymbol{C}$.

**Differencing SSM.** As a specific version of the preprocessing SSM discussed in Section 3.1.3, we fix $\boldsymbol{a} = \boldsymbol{0}$, $\boldsymbol{B} = \boldsymbol{e_1}$, and set $\boldsymbol{C}$ to recover various order differencing when computing the SSM, *i.e.*,

$$\boldsymbol{C} = [1 \quad 0 \quad 0 \quad 0 \quad 0 \quad \ldots \quad 0] \quad \text{(0-order differencing, \textit{i.e.}, an identity function)} \quad (24)$$

$$\boldsymbol{C} = [1 \quad -1 \quad 0 \quad 0 \quad 0 \quad \ldots \quad 0] \quad \text{(1st-order differencing)} \quad (25)$$

$$\boldsymbol{C} = [1 \quad -2 \quad 1 \quad 0 \quad 0 \quad \ldots \quad 0] \quad \text{(2nd-order differencing)} \quad (26)$$

$$\boldsymbol{C} = [1 \quad -3 \quad 3 \quad -1 \quad 0 \quad \ldots \quad 0] \quad \text{(3rd-order differencing)} \quad (27)$$

In this work, we only use the above 0, 1st, 2nd, or 3rd-order differencing instantiations. With multiple differencing SSMs in a multi-SSM SPACETIME layer, we initialize differencing SSMs by running through the orders repeatedly in sequence. For example, given five differencing SSMs, the first four SSMs perform 0, 1st, 2nd, and 3rd-order differencing respectively, while the fifth performs 0-order differencing again.

**Moving Average Residual (MA residual) SSM.** As another version of the preprocessing SSM, we can fix $\boldsymbol{a} = \boldsymbol{0}$, $\boldsymbol{B} = \boldsymbol{e_1}$, and set $\boldsymbol{C}$ such that the SSM outputs sample residuals from a moving average applied over the input sequence. For an $n$-order moving average, we compute outputs with $\boldsymbol{C}$ specified as

$$\boldsymbol{C} = [1 - 1/n, \quad -1/n, \quad \ldots \quad -1/n, \quad 0 \quad \ldots \quad 0] \quad \text{($n$-order moving average residual)} \quad (28)$$

For each MA residual SSM, we randomly initialize the order by uniform-randomly sampling an integer in the range $[4, d]$, where $d$ is again the state-space dimension size (recall $\boldsymbol{C} \in \mathbb{R}^{1 \times d}$). We pick 4 as a heuristic which was not finetuned; we leave additional optimization here for further work.

#### D.7.2 TASK-SPECIFIC SPACETIME ARCHITECTURES

Here, we provide layer-level details on the SPACETIME networks used in this work. For each task, we describe number of layers, number of SSMs per layer, state-space dimension (fixed for all SSMs in a network), and which SSMs are used in each layer.

Expanding on this last detail, as previously discussed in Section 3.1.2, in each SPACETIME layer we can specify multiple SSMs in each layer, computing their outputs in parallel to produce a multidimensional output that is fed as the input to the next SPACETIME layer. The "types" of SSMs do not all have to be the same per layer, and we list the type (companion, shift, differencing, MA residual) and closed-loop designation (standard, closed-loop) of the SSMs in each layer below.

For an additional visual overview of a SPACETIME network, please refer back to Figure 2.

Table 12: SPACETIME forecasting architecture. For all SSMs, we keep state-space dimension $d = 128$. Repeated Identity denotes repeating the input to match the number of SSMs in the next layer, *i.e.*, 128 SSMs in this case. For each forecasting task, $d'$ denotes time series samples' number of features, $\ell$ denotes the lag size (number of past samples given as input), and $h$ denotes the horizon size (number of future samples to be predicted).

| Layer | Details | Input Size | Output Size |
|---|---|---|---|
| Decoder | Linear | $128 \times \ell$ | $d' \times h$ |
| SSM Layer 3 | $\begin{bmatrix} \text{Companion} \\ \text{(closed-loop)} \end{bmatrix} \times 128$ | $128 \times \ell$ | $128 \times \ell$ |
| SSM Layer 2 | $\begin{bmatrix} \text{Companion} \\ \text{(standard)} \end{bmatrix} \times 128$ | $128 \times \ell$ | $128 \times \ell$ |
| SSM Layer 1 | $\begin{bmatrix} \text{Differencing} \\ \text{(standard)} \end{bmatrix} \times 64$  $\begin{bmatrix} \text{MA Residual} \\ \text{(standard)} \end{bmatrix} \times 64$ | $128 \times \ell$ | $128 \times \ell$ |
| Encoder | Repeated Identity | d' $\times \ell$ | $128 \times \ell$ |

**Forecasting: Informer and Monash.** We describe the architecture in Table 12. We treat the first SPACETIME layer as "preprocessing" layer, which performs differencing and moving average residual operations on the input sequence. We treat the last SPACETIME layer as a "forecasting" layer, which autoregressively outputs future horizon predictions given the second-to-last layer's outputs as an input sequence.

**Classification: ECG.** We describe the architectures for each ECG classification task in Tables 13–18. For all models, we use state-space dimension $d = 64$. As described in the experiments, for classification we compute logits with a mean pooling over the output sequence, where pooling is computed over the sequence length.

**Classification: Speech Audio.** We describe the architecture for the Speech Audio task in Table 19. We use state-space dimension $d = 1024$. As described in the experiments, for classification we compute logits with a mean pooling over the output sequence, where pooling is computed over the sequence length.

Table 13: SPACETIME architecture for ECG SuperDiagnostic classification. For all SSMs, we keep state-space dimension $d = 64$. Input samples have $d' = 12$ features and are length $\ell = 1000$ time-steps long. The number of classes $c = 5$.

| Layer | Details | Input Size | Output Size |
|---|---|---|---|
| Classifier | Mean Pooling | $c \times \ell$ | $c \times 1$ |
| Decoder | Linear | $256 \times \ell$ | $c \times \ell$ |
| SSM Layer 5 | $\begin{bmatrix} \text{Companion} \\ \text{(standard)} \end{bmatrix} \times 256$ | $256 \times \ell$ | $256 \times \ell$ |
| SSM Layer 4 | $\begin{bmatrix} \text{Companion} \\ \text{(standard)} \end{bmatrix} \times 256$ | $256 \times \ell$ | $256 \times \ell$ |
| SSM Layer 3 | $\begin{bmatrix} \text{Companion} \\ \text{(standard)} \end{bmatrix} \times 256$ | $256 \times \ell$ | $256 \times \ell$ |
| SSM Layer 2 | $\begin{bmatrix} \text{Companion} \\ \text{(standard)} \end{bmatrix} \times 256$ | $256 \times \ell$ | $256 \times \ell$ |
| SSM Layer 1 | $\begin{bmatrix} \text{Differencing} \\ \text{(standard)} \end{bmatrix} \times 256$ | $256 \times \ell$ | $256 \times \ell$ |
| Encoder | Linear | $d' \times \ell$ | $256 \times \ell$ |

Table 14: SPACETIME architecture for ECG SubDiagnostic classification. For all SSMs, we keep state-space dimension $d = 64$. Input samples have $d' = 12$ features and are length $\ell = 1000$ time-steps long. The number of classes $c = 23$.

| Layer | Details | Input Size | Output Size |
|---|---|---|---|
| Classifier | Mean Pooling | $c \times \ell$ | $c \times 1$ |
| Decoder | Linear | $256 \times \ell$ | $c \times \ell$ |
| SSM Layer 5 | $\begin{bmatrix} \text{Shift} \\ \text{(standard)} \end{bmatrix} \times 256$ | $256 \times \ell$ | $256 \times \ell$ |
| SSM Layer 4 | $\begin{bmatrix} \text{Shift} \\ \text{(standard)} \end{bmatrix} \times 256$ | $256 \times \ell$ | $256 \times \ell$ |
| SSM Layer 3 | $\begin{bmatrix} \text{Shift} \\ \text{(standard)} \end{bmatrix} \times 256$ | $256 \times \ell$ | $256 \times \ell$ |
| SSM Layer 2 | $\begin{bmatrix} \text{Shift} \\ \text{(standard)} \end{bmatrix} \times 256$ | $256 \times \ell$ | $256 \times \ell$ |
| SSM Layer 1 | $\begin{bmatrix} \text{Differencing} \\ \text{(standard)} \end{bmatrix} \times 256$ | $256 \times \ell$ | $256 \times \ell$ |
| Encoder | Linear | $d' \times \ell$ | $256 \times \ell$ |

Table 15: SPACETIME architecture for ECG Diagnostic classification. For all SSMs, we keep state-space dimension $d = 64$. Input samples have $d' = 12$ features and are length $\ell = 1000$ time-steps long. The number of classes $c = 44$.

| Layer | Details | Input Size | Output Size |
|---|---|---|---|
| Classifier | Mean Pooling | $c \times \ell$ | $c \times 1$ |
| Decoder | Linear | $256 \times \ell$ | $c \times \ell$ |
| SSM Layer 5 | $\begin{bmatrix} \text{Shift} \\ \text{(standard)} \end{bmatrix} \times 256$ | $256 \times \ell$ | $256 \times \ell$ |
| SSM Layer 4 | $\begin{bmatrix} \text{Shift} \\ \text{(standard)} \end{bmatrix} \times 256$ | $256 \times \ell$ | $256 \times \ell$ |
| SSM Layer 3 | $\begin{bmatrix} \text{Shift} \\ \text{(standard)} \end{bmatrix} \times 256$ | $256 \times \ell$ | $256 \times \ell$ |
| SSM Layer 2 | $\begin{bmatrix} \text{Shift} \\ \text{(standard)} \end{bmatrix} \times 256$ | $256 \times \ell$ | $256 \times \ell$ |
| SSM Layer 1 | $\begin{bmatrix} \text{Differencing} \\ \text{(standard)} \end{bmatrix} \times 256$ | $256 \times \ell$ | $256 \times \ell$ |
| Encoder | Linear | $d' \times \ell$ | $256 \times \ell$ |

Table 16: SPACETIME architecture for ECG Form classification. For all SSMs, we keep state-space dimension $d = 64$. Input samples have $d' = 12$ features and are length $\ell = 1000$ time-steps long. The number of classes $c = 19$.

| Layer | Details | Input Size | Output Size |
|---|---|---|---|
| Classifier | Mean Pooling | $c \times \ell$ | $c \times 1$ |
| Decoder | Linear | $256 \times \ell$ | $c \times \ell$ |
| SSM Layer 5 | $\begin{bmatrix} \text{Companion} \\ \text{(standard)} \end{bmatrix} \times 256$ | $256 \times \ell$ | $256 \times \ell$ |
| SSM Layer 4 | $\begin{bmatrix} \text{Companion} \\ \text{(standard)} \end{bmatrix} \times 256$ | $256 \times \ell$ | $256 \times \ell$ |
| SSM Layer 3 | $\begin{bmatrix} \text{Companion} \\ \text{(standard)} \end{bmatrix} \times 256$ | $256 \times \ell$ | $256 \times \ell$ |
| SSM Layer 2 | $\begin{bmatrix} \text{Companion} \\ \text{(standard)} \end{bmatrix} \times 256$ | $256 \times \ell$ | $256 \times \ell$ |
| SSM Layer 1 | $\begin{bmatrix} \text{Differencing} \\ \text{(standard)} \end{bmatrix} \times 192$ 
 $\begin{bmatrix} \text{MA Residual} \\ \text{(standard)} \end{bmatrix} \times 64$ | $256 \times \ell$ | $256 \times \ell$ |
| Encoder | Linear | $d' \times \ell$ | $256 \times \ell$ |

Table 17: SPACETIME architecture for ECG Rhythm classification. For all SSMs, we keep state-space dimension $d = 64$. Input samples have $d' = 12$ features and are length $\ell = 1000$ time-steps long. The number of classes $c = 12$.

| Layer | Details | Input Size | Output Size |
|---|---|---|---|
| Classifier | Mean Pooling | $c \times \ell$ | $c \times 1$ |
| Decoder | Linear | $256 \times \ell$ | $c \times \ell$ |
| SSM Layer 5 | $\begin{bmatrix} \text{Companion} \\ \text{(standard)} \end{bmatrix} \times 128$ $\begin{bmatrix} \text{Shift} \\ \text{(standard)} \end{bmatrix} \times 128$ | $256 \times \ell$ | $256 \times \ell$ |
| SSM Layer 4 | $\begin{bmatrix} \text{Companion} \\ \text{(standard)} \end{bmatrix} \times 128$ $\begin{bmatrix} \text{Shift} \\ \text{(standard)} \end{bmatrix} \times 128$ | $256 \times \ell$ | $256 \times \ell$ |
| SSM Layer 3 | $\begin{bmatrix} \text{Companion} \\ \text{(standard)} \end{bmatrix} \times 128$ $\begin{bmatrix} \text{Shift} \\ \text{(standard)} \end{bmatrix} \times 128$ | $256 \times \ell$ | $256 \times \ell$ |
| SSM Layer 2 | $\begin{bmatrix} \text{Companion} \\ \text{(standard)} \end{bmatrix} \times 128$ $\begin{bmatrix} \text{Shift} \\ \text{(standard)} \end{bmatrix} \times 128$ | $256 \times \ell$ | $256 \times \ell$ |
| SSM Layer 1 | $\begin{bmatrix} \text{Differencing} \\ \text{(standard)} \end{bmatrix} \times 256$ | $256 \times \ell$ | $256 \times \ell$ |
| Encoder | Linear | $d' \times \ell$ | $256 \times \ell$ |

Table 18: SPACETIME architecture for ECG All classification. For all SSMs, we keep state-space dimension $d = 64$. Input samples have $d' = 12$ features and are length $\ell = 1000$ time-steps long. The number of classes $c = 71$.

| Layer | Details | Input Size | Output Size |
|---|---|---|---|
| Classifier | Mean Pooling | $c \times \ell$ | $c \times 1$ |
| Decoder | Linear | $256 \times \ell$ | $c \times \ell$ |
| SSM Layer 5 | $\begin{bmatrix} \text{Shift} \\ \text{(standard)} \end{bmatrix} \times 256$ | $256 \times \ell$ | $256 \times \ell$ |
| SSM Layer 4 | $\begin{bmatrix} \text{Shift} \\ \text{(standard)} \end{bmatrix} \times 256$ | $256 \times \ell$ | $256 \times \ell$ |
| SSM Layer 3 | $\begin{bmatrix} \text{Shift} \\ \text{(standard)} \end{bmatrix} \times 256$ | $256 \times \ell$ | $256 \times \ell$ |
| SSM Layer 2 | $\begin{bmatrix} \text{Shift} \\ \text{(standard)} \end{bmatrix} \times 256$ | $256 \times \ell$ | $256 \times \ell$ |
| SSM Layer 1 | $\begin{bmatrix} \text{Differencing} \\ \text{(standard)} \end{bmatrix} \times 192$ 
 $\begin{bmatrix} \text{MA Residual} \\ \text{(standard)} \end{bmatrix} \times 64$ | $256 \times \ell$ | $256 \times \ell$ |
| Encoder | Linear | $d' \times \ell$ | $256 \times \ell$ |

Table 19: SPACETIME architecture for Speech Audio classification. For all SSMs, we keep state-space dimension $d = 1024$. Input samples have $d' = 1$ features and are length $\ell = 16000$ time-steps long. The number of classes $c = 10$.

| Layer | Details | Input Size | Output Size |
|-------|---------|------------|-------------|
| Classifier | Mean Pooling | $c \times \ell$ | $c \times 1$ |
| Decoder | Linear | $256 \times \ell$ | $c \times \ell$ |
| SSM Layer 6 | $\begin{bmatrix} \text{Companion} \\ \text{(standard)} \end{bmatrix} \times 256$ | $256 \times \ell$ | $256 \times \ell$ |
| SSM Layer 5 | $\begin{bmatrix} \text{Companion} \\ \text{(standard)} \end{bmatrix} \times 256$ | $256 \times \ell$ | $256 \times \ell$ |
| SSM Layer 4 | $\begin{bmatrix} \text{Companion} \\ \text{(standard)} \end{bmatrix} \times 256$ | $256 \times \ell$ | $256 \times \ell$ |
| SSM Layer 3 | $\begin{bmatrix} \text{Companion} \\ \text{(standard)} \end{bmatrix} \times 256$ | $256 \times \ell$ | $256 \times \ell$ |
| SSM Layer 2 | $\begin{bmatrix} \text{Companion} \\ \text{(standard)} \end{bmatrix} \times 256$ | $256 \times \ell$ | $256 \times \ell$ |
| SSM Layer 1 | $\begin{bmatrix} \text{Companion} \\ \text{(standard)} \end{bmatrix} \times 256$ | $256 \times \ell$ | $256 \times \ell$ |
| Encoder | Linear | $d' \times \ell$ | $256 \times \ell$ |

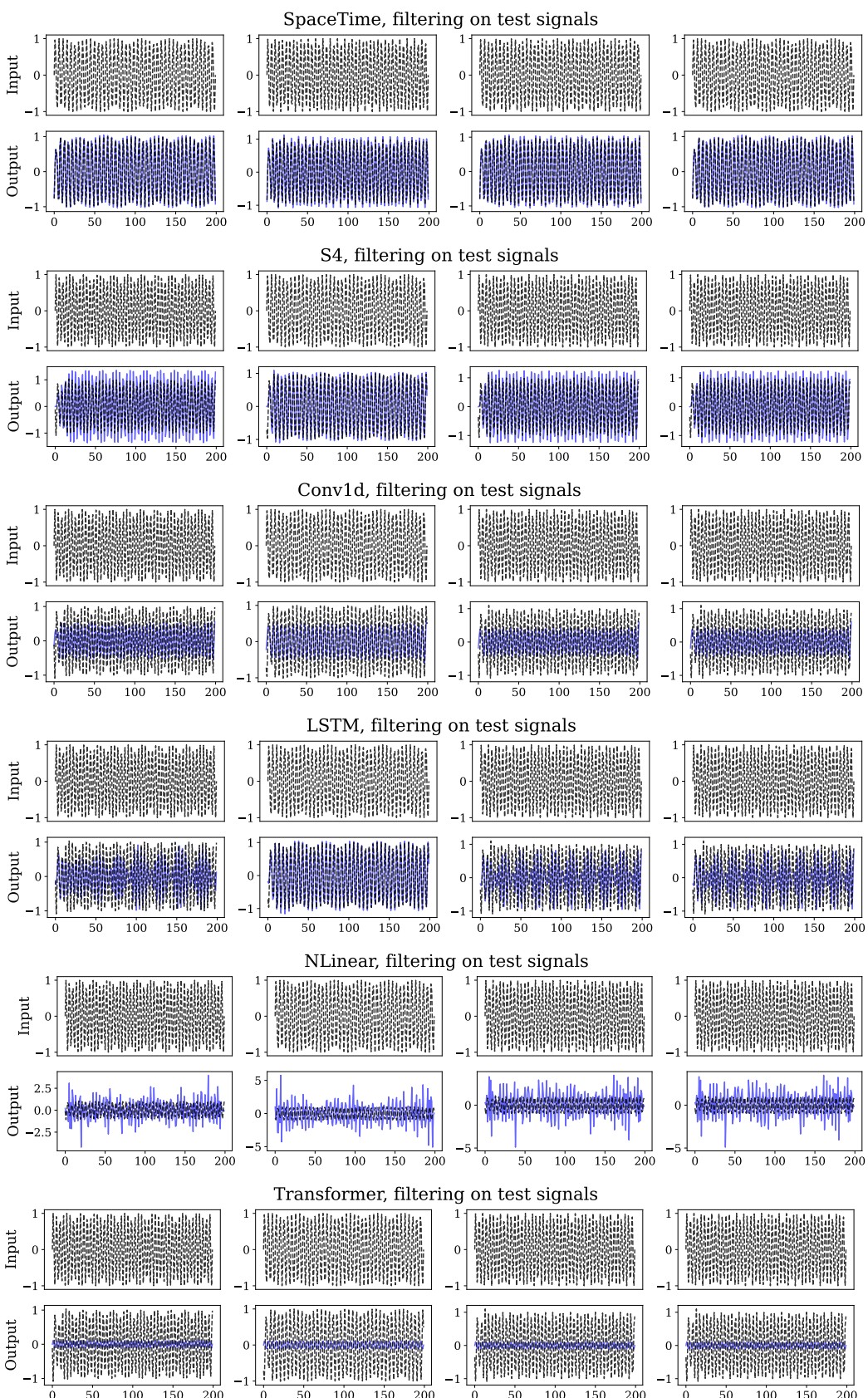

Figure 8: Testing the capability of different sequence–to–sequence models to approximate the input–output map of digital filters. In blue, we show the output signal filtered by each model. The ground–truth digital filter is a Butterworth of order 10.

Table 20: Monash forecasting. We report test RMSE of SPACETIME for each dataset (best result selected via validation RMSE, average of 3 runs).

| Dataset | SPACETIME | SES | Theta | TBATS | ETS | (DHR-)ARIMA | PR | CatBoost | DeepAR | N-BEATS | WaveNet | Transformer |
|---|---|---|---|---|---|---|---|---|---|---|---|---|
| M1 Yearly | 135508.3 | 193829.5 | 171458.1 | **116850.9** | 167739.0 | 175343.8 | 152038.7 | 237644.5 | 173075.1 | 192489.8 | 312821.8 | 182850.6 |
| M1 Quarterly | 2200.3 | 2545.7 | 2282.7 | 2673.9 | 2408.5 | 2538.5 | 1909.3 | 2161.0 | 2313.3 | 2267.3 | 2271.7 | 2231.5 |
| M1 Monthly | 2601.1 | 2725.8 | 2564.9 | 2594.5 | 2264.0 | 2450.6 | 2478.8 | 2461.7 | 2202.2 | **2183.4** | 2578.9 | 3129.8 |
| M3 Yearly | 1412.4 | 1172.9 | 1106.1 | 1386.3 | 1189.2 | 1662.2 | 1181.8 | 1341.7 | 1157.9 | 1117.4 | 1147.6 | 1084.8 |
| M3 Quarterly | 676.1 | 670.6 | 567.7 | 653.6 | 598.7 | 650.8 | 605.5 | 698.0 | 606.6 | 582.8 | 606.8 | 819.2 |
| M3 Monthly | 897.12 | 893.9 | 754.0 | 765.2 | 755.3 | 790.8 | 830.0 | 874.2 | 873.7 | 796.9 | 845.3 | 948.4 |
| M3 Other | 265.56 | 309.7 | 242.1 | 217.0 | 224.1 | 220.8 | 262.3 | 349.9 | 277.7 | 248.5 | 277.0 | 271.0 |
| M4 Quarterly | 718.2 | 732.8 | 673.2 | 672.7 | 674.3 | 710.0 | 711.9 | 714.2 | 700.3 | 684.7 | 697.0 | 739.1 |
| M4 Monthly | 1092.2 | 755.5 | 683.7 | 743.4 | 705.7 | 702.1 | 720.5 | 734.8 | 740.3 | 705.2 | 787.9 | 902.4 |
| M4 Weekly | **348.3** | 412.6 | 405.2 | 356.7 | 408.5 | 386.3 | 350.3 | 420.8 | 422.2 | 330.8 | 437.3 | 456.9 |
| M4 Daily | **183.2** | 209.8 | 210.4 | 208.4 | 230.0 | 212.6 | 213.0 | 263.1 | 343.5 | 221.7 | 220.5 | 233.6 |
| M4 Hourly | **255.2** | 1476.8 | 1483.7 | 469.9 | 3830.4 | 1563.1 | 313.0 | 344.6 | 1095.1 | 501.2 | 468.1 | 391.2 |
| Tourism Yearly | **74799.2** | 106665.2 | 99914.2 | 105799.4 | 104700.5 | 106082.6 | 89645.6 | 87489.0 | 78470.7 | 78241.7 | 77581.3 | 80089.3 |
| Tourism Quarterly | 11608.32 | 15000.0 | 9254.6 | 12001.5 | 10812.3 | 12564.8 | 11746.9 | 12788.0 | 11762.0 | 11306.0 | 11546.6 | 11724.1 |
| Tourism Monthly | 3181.2 | 7039.4 | 2702.0 | 3661.5 | 2543.0 | 3132.4 | 2739.4 | 3102.8 | 2359.9 | 2596.2 | 2694.2 | 2660.1 |
| Pedestrian | 69.6 | 228.1 | 228.2 | 261.3 | 278.3 | 820.3 | 61.8 | 60.8 | 65.8 | 99.3 | 68.0 | 70.2 |
| Weather | **2.7** | 2.9 | 3.3 | 2.9 | 3.0 | 3.1 | 9.1 | 3.1 | **2.7** | 3.1 | 3.0 | 2.8 |
| NN5 Weekly | **16.9** | 18.8 | 18.7 | 18.5 | 18.8 | 18.6 | 18.6 | 18.7 | 18.5 | 17.4 | 24.2 | 24.0 |
| Solar 10 min | 7.4 | 7.2 | 7.2 | 10.7 | 7.2 | 5.6 | 7.2 | 8.7 | 7.2 | 6.6 | 8.0 | 7.2 |
| Solar Weekly | 1423.7 | 1331.3 | 1341.6 | 1049.0 | 1264.4 | 967.9 | 1168.2 | 1754.2 | 873.6 | 1307.8 | 2569.3 | 693.8 |
| Electricity Hourly | **475.1** | 1026.3 | 1026.4 | 743.4 | 1524.9 | 1082.4 | 689.9 | 582.7 | 478.0 | 510.9 | 489.9 | 514.7 |
| Electricity Weekly | 37802.2 | 77067.9 | 76935.6 | 28039.7 | 70369.0 | 32594.8 | 47802.1 | 37289.7 | 53100.3 | 35576.8 | 63916.9 | 78894.7 |
| Fred-MD | 3743.6 | 3103.0 | 3898.7 | 2295.7 | 2341.7 | 3312.5 | 9736.9 | 2679.4 | 4638.7 | 2813.0 | 2779.5 | 5098.9 |
| Traffic Hourly | 0.03 | 0.04 | 0.04 | 0.05 | 0.04 | 0.04 | 0.03 | 0.03 | 0.02 | 0.02 | 0.03 | 0.02 |
| Traffic Weekly | **1.3** | 1.5 | 1.5 | 1.5 | 1.5 | 1.5 | 1.5 | 1.5 | 1.5 | 1.4 | 1.6 | 1.9 |
| Hospital | 40.1 | 26.6 | 22.6 | 21.3 | 22.0 | 23.7 | 23.5 | 23.5 | 22.0 | 24.2 | 23.4 | 40.5 |
| Covid | 490.1 | 403.4 | 370.1 | 113.0 | 102.1 | 100.5 | 394.1 | 607.9 | 230.5 | 186.5 | 1135.4 | 480.0 |
| Saugeen | **24.0** | 39.8 | 39.8 | 42.6 | 50.4 | 43.2 | 47.7 | 39.3 | 45.3 | 48.9 | 43.0 | 49.1 |
| US Births | 630.2 | 1369.5 | 735.5 | 606.5 | 607.2 | 705.5 | 732.1 | 618.4 | 684.0 | 627.7 | 768.8 | 686.5 |
| Sunspot | 3.1 | 5.0 | 5.0 | 3.0 | 5.0 | 3.0 | 4.0 | 2.4 | 1.1 | 14.5 | 0.7 | 0.5 |
| Car Parts | 0.64 | 0.71 | 0.65 | 0.71 | 0.71 | 0.71 | 0.58 | 0.71 | **0.50** | 1.0 | 0.58 | 0.5 |
| Vehicle Trips | 30.4 | 36.5 | 37.4 | 25.7 | 37.6 | 35.0 | 31.7 | 27.3 | 26.5 | 33.6 | 29.0 | 33.0 |

