# OpenReview forum: "Effectively Modeling Time Series with Simple Discrete State Spaces"
_ICLR.cc/2023/Conference — ICLR 2023 poster_

### Official Review · Reviewer_BK6G · 2022-10-15

**Confidence:** 3
**Correctness:** 3
**Technical Novelty And Significance:** 2
**Empirical Novelty And Significance:** 2
**Recommendation:** 3

**Clarity, Quality, Novelty And Reproducibility:**

The paper is written in a clear language. The overall scientific quality of the paper is decent but the studied topic is not really a machine learning problem. The reported results look reproducible. The presented idea of maintaining a companion matrix is novel in its context, while being a straightforward allocation of existing ideas.

**Strength And Weaknesses:**

**Strengths:**

  * The paper reports a rigorous empirical analysis of the proposed method, though the experiments are designed from a signal processing point of view rather than machine learning.

**Weaknesses:**

  * The proposed method assumes a linear state space model, which is a major restriction for the wide-scope applicability of the method to general machine learning problems. It performs on par with an LSTM on the ECG problem probably because a preprocessing step on the signal, such as a Fourier Transform, makes the problem linearly solvable. This will not be the case for any other arbitrary machine learning problem. It is also likely that an LSTM cascade or GRU would give even better results, both of which are way below state of the art methods.
  * The experimental results are rather inconclusive. The proposed method appears to work favorably on univariate forecasting tasks, which are not so much within the scope of machine learning research. In more challenging tasks such as ECG and speech audio classification, the proposed method does not appear to meet the state of the art. The paper also does not explain why a comparison against a transformer in Tables 2 and 3 was not possible. They are currently operational in much bigger processing pipelines than the ones used in this paper.


**Summary Of The Paper:**

The paper proposes to improve a linear state space model’s expressive power by maintaining a companion matrix, which encodes essential properties of the dynamics into a discrete set of quantities.

**Summary Of The Review:**

I do not have the impression that the paper speaks to the target audience of ICLR. Its algorithmic details could be better fit to a signal processing venue. From a machine learning research perspective, what I can only see is a model that is not working charmingly well even in much simpler data sets than an average machine learning research paper uses.

After rebuttal
---

I read the author response and decided to keep my score. The reply to the comparative strength of the proposed method to LSTM/GRU and the degree of difficulty of the studied data sets is not convincing. I still do not think the paper's content is relevant for the machine learning community.

---

> ### Author Response · Authors · 2022-11-17
> **Author Response (1/2)**
>
> Thank you for your review. We appreciate your acknowledgement of our rigorous empirical analysis, paper quality, and novelty of the companion matrix.
>
> However, in response to your concerns, we also want to clarify several important points on our model architecture, the grounding of our study + our contributions, and our experimental evaluation, to support how our work contributes advances in time series modeling *and* deep learning.
>
> We address each of your concerns below. We give further clarification, and point to the relevant parts of the original submission as evidence for our responses.
>
> **"Linear state space model is a major restriction for wide-scope applicability"**
> We believe this is a misunderstanding of our architecture. While our core building block uses linear state-space models (a linear layer with weights that parameterize an SSM; c.f. Section 2, Section 3.1), to actually build our networks, **we stack multiple such layers with nonlinearities in-between (Figure 2, Section 3.1.2)**. This is **similar to how Transformers intersperse attention with nonlinearities** [1]. It also makes our proposed networks *nonlinear*.
>
> As counterpoints to the claimed “major restriction”, we note that **this is also similar to how CNNs are built** [2]. They incorporate a core linear component (the convolution over different kernels is a *linear* operation) with nonlinear activations. Needless to say, CNNs are a widely-applied architecture.
>
> Furthermore, even more similarly, other versions of our linear SSM framework have been applied in related architectures that achieve state-of-the-art results on a wide range of tasks and modalities, including image classification [3, 4], language modeling [4], audio classification & generation [5], and offline RL policy learning [6].
>
>
> **“Studied topic is not really a machine learning problem”**
> We believe our work tackles multiple machine learning problems, both from the angle of time series modeling, as well as the angle of making fundamental improvements to a **growing line of research on deep SSM architectures**.
>
> Regarding time series modeling, please see our later response on your comment that "univariate forecasting tasks are not so much within the scope of machine learning research".
>
> Regarding work on deep SSM architectures, we study and make contributions on how to **fundamentally improve the expressivity and efficiency of a recent class of deep SSM models**, popularized by S4 [4], which won an ICLR 2022 Outstanding Paper Honorable Mention award.
>
> While these contributions are motivated by challenges in time series modeling, **improving these deep SSM’s autoregressive capabilities is a general advance** widely applicable to any causal modality.
> * Please find our discussion and contributions for improving expressivity in Section 3.1 and Propositions 1 and 2, validated in Section 4.2.2 (Figure 3); and our discussion and contributions for improving efficiency in Sections 3.2 and 3.3 and Algorithm 1, validated in Section 4.2.3 (Table 5, Figure 5).
>
> Furthermore, rather than being a “straightforward allocation of existing ideas”, our contributions uncover fundamental limitations with these prior SSM formulations. **We provide new neural net architectures and training & inference algorithms to solve these issues**.
> * Prior works such as S4 show one way to compute the SSM efficiently and practically in a deep learning architecture, but we provably show its limitations.
> * We thus propose an alternative that improves the expressivity of the learned SSMs, while coming up with the necessary algorithms and architectures to maintain prior benefits.

---

> > ### Author Response · Authors · 2022-11-17
> > **Author Response (2/2)**
> >
> > **(Responses and clarifying details to "Experimental results are rather inconclusive")**
> >
> > ***Clarification on preprocessing and linear solvability - "It performs on par with an LSTM on the ECG problem probably because a preprocessing step on the signal, such as a Fourier Transform, makes the problem linearly solvable"***
> > We think this is a misunderstanding of our method. We take no processing steps for ECG classification [^1]. The noted Fourier Transform is not used to preprocess data, but rather to compute our model’s layer outputs efficiently as a convolution (Section 3.1.2).
> >
> > We also clarify that in Table 2 we consistently outperform LSTMs across all 6 ECG tasks (+0.016 points on average), getting best AUROC on 2/6 tasks and second-best AUROC on the remaining 3/4 tasks. The best method (S4, also a linear SSM-based model [4]) improves AUROC by +0.019 points on average.
> >
> > Finally, we note that despite the SSM being a linear model, our work and others [3, 4, 5, 6] have shown how deep networks based on SSMs can achieve state-of-the-art results on a wide range of modalities [5] and established image and text sequence modeling benchmarks [14].
> >
> > [^1] The only “preprocessing steps” we take are not for classification, but *time series forecasting*. These are either standard linear transformations widely applied on these benchmarks (see the Informer [7] or Monash [13] benchmarks), or linear differencing and moving average capabilities that can be baked into our model via simple weight initializations (as an added benefit and capability of our proposed SSM; see Section 3.1.3 for details).
> >
> > ***“Univariate forecasting tasks are not so much within the scope of machine learning research”***
> > We clarify that a substantial number of recent deep learning works evaluate on univariate forecasting [7, 8, 9, 10, 11, 12], which propose methods building on modern architectures such as Transformers. Univariate forecasting also makes up a substantial part of popular time series forecasting benchmarks such as Informer [7] and Monash [13], which have been used for evaluation of multiple deep learning methods [8, 9, 10, 11, 12].
> >
> > We also included evaluations for multivariate forecasting in Table 8, Appendix D.2. We outperform competitive (Transformer-based) time series architectures in all evaluation settings (e.g., compare to FedFormer, ICML 2022 [10]).
> >
> > ***“In more challenging tasks such as ECG and speech audio classification, the proposed method does not appear to meet the state of the art.” “It is also likely that an LSTM cascade or GRU would give even better results, both of which are way below state of the art methods”***
> >
> > We believe this is a mischaracterization of our results. Our method outperforms the state-of-the-art on 2 / 6 ECG classification tasks, and gets second-best performance on the remaining 3 / 4 tasks by a significant margin (Table 2 in paper; we report average AUROC across all tasks below). In this evaluation, we outperform Transformers, LSTMs and other deep architectures.
> >
> > | Method | SpaceTime | S4 | Inception-1D | xRN-101 | LSTM | Transformer  | Wavelet + NN |
> > |---|:---:|:---:|:---:|:---:|:---:|:---:|:---:|
> > | Average AUROC | 0.932 | **0.935** | 0.927 | 0.929 | 0.916 | 0.851 | 0.847 |
> > | Delta from SoTA | **0.003** | N/A | 0.008 | 0.006 | 0.019 | 0.084 | 0.088 |
> >
> >
> > ***“The paper also does not explain why a comparison against a transformer in Tables 2 and 3 was not possible”***
> > We apologize for the confusing presentation. We compared against Transformers in Table 3; the X means that Transformers failed to perform the task (did not perform better than chance, in this case because we could not train due to memory constraints). We clarified this in the revision (Section 4.1).
> >
> > We reason this is due to Transformer’s quadratic space and time complexity with respect to input length. Table 3’s speech audio classification task requires processing sequences up to 16,000 time-steps, which could not fit in the allotted GPU memory constraints (kept the same for all methods for fair comparison). This suggests that, controlling for practical resource constraints, our method can perform tasks that Transformers cannot. We have clarified this in the revision.
> >
> > In our revision we also added results for Transformers on the ECG tasks in Table 2. We report the numbers below. Our method substantially outperforms Transformers on all 6 tasks.
> >
> > | Task AUROC | All | Diag | Sub-diag | Super-diag | Form | Rhythm |
> > |---|:---:|:---:|:---:|:---:|:---:|:---:|
> > | SpaceTime | **93.6 ± 0.13** | **94.1 ± 0.12** | **93.3 ± 0.34** | **92.9 ± 0.09** | **88.3 ± 0.63** | **96.7 ± 0.05** |
> > | Transformer | 85.7 ± 0.30 | 87.6 ± 0.41 | 88.2 ± 0.20 | 88.7 ± 0.28 | 77.1 ± 0.45 | 83.1 ± 0.72 |

---

> > > ### Author Response · Authors · 2022-11-17
> > > **References**
> > >
> > > 1. A. Vaswani, N. Shazeer, N. Parmar, J. Uszkoreit, L. Jones, A. Gomez, L. Kaiser. and I. Polosukhin. Attention is all you need. NeurIPS 2017.
> > > 2. A. Krizhevsky, I. Sutskever, G. Hinton. ImageNet Classification with Deep Convolutional Neural Networks. NeurIPS 2012.
> > > 3. E. Nguyen, K. Goel, A. Gu, G. Downs, P. Shah, T. Dao, S. Baccus, and C. Ré. S4ND: Modeling Images and Videos as Multidimensional Signals Using State Spaces. NeurIPS 2022.
> > > 4. A. Gu, K. Goel, and C. Ré. Efficiently Modeling Long Sequences with Structured State Spaces. ICLR 2022.
> > > 5. K. Goel, A. Gu, C. Donahue, and C. Ré. “It’s Raw! Audio Generation with State-Space Models. ICML 2022.
> > > 6. Decision S4: Efficient Sequence-Based RL via State Spaces Layers (ICLR 2023 submission)
> > > 7. H. Zhou, S. Zhang, J. Peng, S. Zhang, J. Li, H. Xiong, and W. Zhang. Informer: Beyond Efficient Transformer for Long Sequence Time-Series Forecasting. AAAI 2021.
> > > 8. H. Wu, J. Xu, J. Wang, and M. Long. Autoformer: Decomposition transformers with auto-correlation for long-term series forecasting. NeurIPS 2021.
> > > 9. T. Zhou, Z. Ma, Q. Wen, L. Sun, T. Yao, R. Jin, et al. Film: Frequency improved legendre memory model for long-term time series forecasting. arXiv preprint arXiv:2205.08897 2022.
> > > 10. T. Zhou, Z. Ma, Q. Wen, X. Wang, L. Sun, and R. Jin. Fedformer: Frequency enhanced decomposed transformer for long-term series forecasting. ICML 2022.
> > > 11.  D. Salinas, V. Flunkert, J. Gasthaus, and T. Januschowski. DeepAR: Probabilistic forecasting with autoregressive recurrent networks. International Journal of Forecasting, 2020.
> > > 12.  S. L, X. Jin, Y. Xuan, X. Zhou, W. Chen, Y. Wang, and X. Yan. Enhancing the Locality and Breaking the Memory Bottleneck of Transformer on Time Series Forecasting. NeurIPS 2019.
> > > 13.  R. Godahewa, C. Bergmeir, G. Webb, R. Hyndman, and P. Montero-Manso. Monash Time Series Forecasting Archive. arXiv preprint arXiv:2105.06643 2021.
> > > 14.  Y. Tay, M. Dehghani, S. Abnar, Y. Shen, D. Bahri, P. Pham, J. Rao, L. Yang, S. Ruder, and D. Metzler. Long Range Arena : A Benchmark for Efficient Transformers. ICLR 2021

---

### Official Review · Reviewer_YRVv · 2022-10-25

**Confidence:** 2
**Correctness:** 3
**Technical Novelty And Significance:** 3
**Empirical Novelty And Significance:** 3
**Recommendation:** 6

**Clarity, Quality, Novelty And Reproducibility:**

The writing of the paper leaves questions (see details above) regarding the quality of the technical work presented. This also hampers the reproducibiilty of the work.


**Strength And Weaknesses:**

Strength

The formulation of the SSM presented appeared to be new.

The experiments considered a large number of forecasting and classification tasks.

The detailed breakdown analyses of each of the stated goals in 4.2 are appreciated.

Weakness

-The state-space formulation in (1-3) is somewhat unconventional. In existing SSM works, u_k in (1-2) would be corresponding to external controls applied to the system, and y_k the observations. In what presented in (1-3), u_k and y_k are essentially the same variables, one being what observed in the look-back windows, and one to be forecasted. This somehow mixes the “estimation” of the state-space modes with the “generative” formulation of the state-space models. The intuition for such a mix — as the premise of the work — is not clear.

- The statement of existing SSMs not being able to capture the autoregressive process is not valid. Perhaps the authors are referring to when the state transition model is formulated as in (1-2) with a Markovian assumption? But the transition function can be much more general than by introducing a larger lag in time for the dependency.


- The rationale behind the closed-loop modeling is also somewhat counter-intuitive. Essentially it is suggesting that the forecasting of y_K+1 is not going to be good enough for it to be used as the input for the future forecasting? What is the fundamental rationale to believe that (11) and (12) can be identified as two different models? If y_k’s are accurately forecasted, it should be sufficient to be used as inputs for future forecasting; the fact that another variable needs to be predicted to be used as "input" to the model means that y's would not be well estimated?

- Some of the strong AR baselines such as N-BEATs and DeepAR are missed in the experiments. Since the paper has an emphasis of taking a SSM approach, comparison to existing SSM-based time-series forecasting models is also necessary. This again speaks to the need to demonstrate the benefits of the presented modeling that appears to be a unusual mix of a SSM and AR formulation.


**Summary Of The Paper:**

This paper presents a new SSM formulation for time-series modeling that utilizes some explicit modeling of “companion” matrices to model the autoregressive nature of the time-series, and a “closed-loop” formulation that outputs the “input” for the next frames in forecasting. Experiments are conducted on several benchmark datasets for time-series forecasting and classification, in comparison to selected baseline models.


**Summary Of The Review:**

This paper presents a SSM formulation for time-series forecasting that includes "autoregressive"m modeling in a state-space formulation and adds the prediction of the "input" to the forecasting model as an additional output. While improvements were demonstrated with respect to selected baselines, I found the premise for both of the two primary contributions unclear. Missing necessary baselines to either line of the related works, especially those in deep SSMs, further add to the concern.

Post-rebuttal update: The extensive discussion and revision during the rebuttal phase helped me appreciate the contribution of the work better, because of which I'm raising my score.

---

> ### Author Response · Authors · 2022-11-17
> **Author Response (1/3)**
>
> Thank you for your review and the clarifying questions! To address your concerns, in our responses below, we:
> * Explain the reasoning and motivation behind our state-space model formulation
> * Clarify our claims on the limitations of prior SSMs for time series modeling. **We do not state that existing SSMs are not able to capture the autoregressive process**, but rather that the specific SSMs used in ***prior related deep learning models*** cannot capture autoregressive dependencies (see Proposition 2).
>   * These SSMs offer numerous benefits for deep learning (e.g., in S4 [2]), and we build on these ideas to inherit the benefits. However, they also have fundamental autoregressive modeling limitations, which serve as motivation for our work to improve this framework.
> * Clarify the mechanics of our closed-loop SSM, which ***does* autoregressively predict future time series samples**, but aims to do this ***efficiently*** by also separately predicting future time-steps of its own input process (which are not necessarily the same as the target time series in a multi-layer network)
> * Clarify that **we do compare against strong AR baselines such as DeepAR in the original submission (Table 7)**, and have added comparisons to N-BEATS. These baselines were not in the main results due to their poor relative performance to more modern methods.
>   * In addition, we compared against the strong S4 SSM baseline, which recently outperformed Transformers and LSTMs on general long sequence modeling tasks [2]. Our SpaceTime models significantly improve performance over S4 (Table 1).
>
> **Clarification of state-space formulation and conventions**
> Thanks for pointing out these conventional SSM modeling choices. We clarify and give some intuition for our setup.
>
> We agree that in SSMs, $u_k$ are typically modeled as “inputs” or “controls” applied to the system, i.e. samples from an input sequence, and $y_k$ are “outputs” or “observations”, i.e. samples from an output sequence. The SSM matrices specify how the output sequence is a function of the input sequence; we thus aim to learn this function as a mapping between the two sequences. However, it is also common to reason about controls as functions of the state – if available – or the output e.g., in closed-loop systems.
>
> For intuition of our setup, we note that time series processes can also be formulated as SSMs, where observations at future time-steps are a function of observations at prior  time-steps. In particular, there is an autoregressive relationship, i.e. $y_t$ depends on some $L$ prior time-steps $y_{t - 1} \ldots y_{t - L}$. Note that in this case the relationship is not necessarily order-1 Markovian: $y_{t}$ can in general depend on $y_{t-k}$ for $k >  1$. One way to handle longer dependencies is to embed observations at different time steps into a higher-dimensional state and have it follow regular 1-Markov dynamics (see e.g., Taken’s embedding theorem [cite: "Detecting strange attractors in turbulence".]).
>
> If we can learn the SSM that describes the dynamics of such an embedding, we can reliably predict future values for accurate forecasting.
>
> This choice is also motivated by the observation that time series data is typically given as a single sequence, from which we specify inputs, outputs and controls. Besides the reasoning above, our choice borrows from prior work such as LSSL and S4 [1, 2] that demonstrates numerous advances in sequence modeling using the SSM. We note that equations (1-3) are not a new formulation for deep learning (we clarified this in our revision, but cited [1, 2] in our introduction and presented this modeling setup as “Preliminaries” (Section 2) in the original submission). The authors of S4 [2] formulate SSM inputs and outputs for predictions similarly to us.
>
> We acknowledge that other classical techniques (e.g., ARIMA, Kalman Filters) rely on a formulation where the input sequence is given by ID samples of noise (e.g., from a Gaussian distribution), independent from the output observations [3, 4]. We make one such connection in the proof of Proposition 1 in Appendix B, where we show the similarity between the SSM when inputs are i.i.d samples of noise or shifted (delay 1) versions of the observations. We think these connections could be fruitful ground for interesting future work.

---

> > ### Author Response · Authors · 2022-11-17
> > **Author Response (2/3)**
> >
> > **“The statement of existing SSMs not being able to capture the autoregressive process is not valid”**
> > We apologize for the confusion; we believe this is a misunderstanding of our claims. We never state that all existing SSMs cannot capture AR processes; in fact we reference the SSMs used in ARIMA models as an example of SSMs that *do* capture AR processes (Section 3.1.1).
> >
> > We only state that existing SSMs used in prior related works such as S4 [1, 2, 5] cannot capture the autoregressive process due to their representation choice for the A matrix. We discuss this in Section 3.1.1, and proved this claim in Proposition 2 (Appendix B.1).
> >
> > Please see our general response for motivation on why we focus on the SSMs presented in these related works. We have clarified this grounding setup in the revision (Introduction).
> >
> > **“The rationale behind the closed-loop modeling is also somewhat counter-intuitive. Essentially it is suggesting that the forecasting of y_K+1 is not going to be good enough for it to be used as the input for the future forecasting?”**
> > This is a misunderstanding of the closed-loop component and Section 3.2. We emphasize that we are still predicting future values $y_{K + 1}$ given $L$ prior values $y_{K}, y_{K - 1}, \ldots, y_{K - L + 1}$. **However, a core benefit and aim of the closed-loop SSM is to efficiently forecast these values**. Treating this naively as the next-time-step input would hurt the *efficiency* of our network for forecasting long horizons (as discussed in Section 3.2.).
> > * Note this is still better than Transformers, but not the best we could do; hence the closed loop component. We discuss more details on this below.
> >
> > Recall that in our architecture, the outputs of an SSM correspond to the outputs of a single layer (Figure 2, Section 3.1). In a *multi-layer setting*—as standard for modern deep learning architectures—the output $y_{K + 1}$ of the last layer is trained to be the predicted future time-step, but this itself is *not the input* of the *same last layer*. Instead, it would be the input of the *first* layer in our network.
> > * We could still autoregressively feed in the output $y_{K + 1}$ as the next step input to the first layer (as discussed in Section 3.2), but this would hurt the efficiency of our network for forecasting long horizons.
> >
> > To see why, note that we’d need to compute $H$ forward passes through the entire network to predict each future time-step given target horizon length $H$. Because we train our network with $L$ input time-steps for each forward pass (as we have to learn the underlying time series process, which dictates how a function of some number of prior time-steps determines a future time-step), this results in $\mathcal{O}(LH)$ time complexity.
> >
> > To improve our efficiency, we discussed in Section 3.1.2 and 3.2 how the recurrent view of the SSM can be used to compute outputs in $O(L + H)$ time. If the outputs of a single layer can be fed back as inputs to that layer, then we can “rollout” these outputs autoregressively as a fast RNN (c.f. Eq. (14), (15); Algorithm 1).
> >
> > But to motivate the closed-loop component, recall that this requires that we can feed an **output of the last layer** as an accurate prediction for **its next time-step input** (or the *2nd-to-last layer’s outputs*).
> > * The standard framework only allocates the outputs of the last layer to be accurate predictions for the ground-truth observed $y$ value (or the *1st layer’s* next-time-step inputs).
> >
> > Hence the closed-loop component (Eq. 12), which is separately dedicated to specifically predicting the last layer’s next time-step inputs.

---

> > > ### Author Response · Authors · 2022-11-17
> > > **Author Response (3/3)**
> > >
> > > **“Some of the strong AR baselines such as N-BEATs and DeepAR are missed in the experiments”**
> > >
> > > Thanks for bringing up the important comparison to past SSM-based models.
> > >
> > > To clarify, in our original submission, we included comparisons to DeepAR on the Informer and Monash forecasting benchmarks (see Table 7, Appendix D.2; Figure 7, Appendix D.3). SpaceTime outperforms DeepAR on both benchmarks. In our revision, we added comparisons to N-BEATS, where SpaceTime also substantially outperforms N-BEATs on the Informer benchmark.
> > > * On 25 Informer benchmark settings (Table 7), SpaceTime achieves **9.6% to 96.8% lower MSE** than DeepAR on all settings.
> > > * On Monash, SpaceTime notably outperforms DeepAR when needing to process longer input context or forecast longer future horizons (Figure 7). On longer context tasks, SpaceTime achieves **24.4% lower RMSE** than DeepAR on average. On forecasting long horizons, SpaceTime achieves **27.5% lower RMSE** than DeepAR on average.
> > >   * SpaceTime also outperforms DeepAR on tasks with more training samples, achieving 57.6% lower RMSE.
> > >   * However, we do note that there are settings where DeepAR outperforms SpaceTime. For shorter horizons, DeepAR achieves 11.5% lower RMSE on average. When only needing to process shorter contexts, DeepAR gets 19.2% lower RMSE.
> > >
> > > We included these comparisons in the Appendix due to DeepAR’s poor performance relative to other more recent methods. For example, across methods in our main results (Table 1) and extended results (Table 7), DeepAR is outperformed by 14/15 other methods on the ETTh1 dataset when forecasting 720 time-steps ahead, and 13/15 methods on ETTh1 when forecasting 336 time-steps ahead.
> > >
> > > Finally, in our revision, we added N-BEATS results on 15 Informer evaluation settings from [6]. SpaceTime also substantially outperforms N-BEATS on all settings here, achieving **17.8% to 72.8% lower MSE** (Table 7).
> > >
> > > **References**
> > > 1. A. Gu, I. Johnson, K. Goel, K. Saab, T. Dao, A. Rudra, and C. R ́e. Combining Recurrent, Convolutional, and Continuous-time Models with Linear State Space Layers. NeurIPS 2021.
> > > 2. A. Gu, K. Goel, and C. R ́e. Efficiently Modeling Long Sequences with Structured State Spaces. ICLR 2022.
> > > 3. G. Box, G. Jenkins, G. Reinsel, and G. Ljung. Time Series Analysis Forecasting and Control. John Wiley & Sons 1970.
> > > 4. R. Kalman. A New Approach to Linear Filtering and Prediction Problems. Journal of Basic Engineering 1960.
> > > 5. A. Gu, A. Gupta, K. Goel, and C. Re. On the Parameterization and Initialization of Diagonal State Space Models. arXiv preprint arXiv:2206:11893 2022.
> > > 6. M. Liu, A. Zeng, M. Chen, Z. Xu, Q. Lai, L. Ma, Q. Xu. SCINet: Time Series Modeling and Forecasting with Sample Convolution and Interaction. NeurIPS 2022.

---

> > > > ### Comment · Reviewer_YRVv · 2022-11-30
> > > > **Thanks for the response**
> > > >
> > > > I want to thank the authors for the detailed response and explanations. It helped with a better understanding of the work, especially in the line of related SSM formulations prior to this. I have two lingering questions.
> > > >
> > > > 1. In my original review regarding "comparison to existing SSM-based time-series forecasting models", I was referring to the line of deep SSMs that follow a more standard SSM formulation (i.e., inputs do not come from the past outputs) and used in time-series modeling/forecasting. Some examples are below. My main confusion was regarding the advantages of the presented SSM formulations compared to this general line of SSMs. Clarifying this would be very helpful in helping me formulate my final understanding of the contribution of the paper.
> > > >
> > > > - Deep Kalman filters
> > > > - DEEP VARIATIONAL BAYES FILTERS: UNSUPERVISED LEARNING OF STATE SPACE MODELS FROM RAW DATA
> > > > - A Disentangled Recognition and Nonlinear Dynamics Model for Unsupervised Learning
> > > > - Latent Matters: Learning Deep State-Space Models
> > > >
> > > > 2. The clarification of "closed-loop" modeling was helpful for me to better understand the intention of this component. This may be a naive question, but -- after step l and for forecasting the future h steps, if we model the SSM as: x_k+1 = Ax_k + By_k; y_k+1 = Cx_k+1; wouldn't we be able to achieve the same effect for close-loop modeling (such that x_l+i = (A + B)^ix_l  for i = 1, . . . , h-1, and y_l+i  = C(A + B)^ix_l)? As I mentioned, I may be missing something important, but this would really help me understand the significance of this component.
> > > >
> > > > Thanks very much!

---

> > > > > ### Author Response · Authors · 2022-12-08
> > > > > **Response (1/2) - clarification to Q1 on SSM advantages**
> > > > >
> > > > > Thanks for the great clarifying questions and concrete references!
> > > > >
> > > > > **(1)** For your 1st question, please find clarifications of our SSM advantages in the section below. A key benefit is our *practical boost* in **fast and efficient training**. We explain the differences behind this, and support our reasoning by implementing two of the suggested methods. We find SpaceTime:
> > > > > * Trains **up to 12x faster**.
> > > > > * Is *substantially more accurate*. For forecasting 96 future time-steps on ETTh1 real-world data, we get 0.054 MSE; the others fail to achieve <1.8 MSE.
> > > > >
> > > > > **(2)** For your 2nd question, please see our next comment in the thread. You're 100% correct for a *single-layer* model, but we expand on why the closed-loop SSM is needed for *multilayer* deep models.
> > > > >
> > > > > We are happy to provide further clarity. If we've addressed your concerns, we kindly ask that you consider raising your score.
> > > > >
> > > > > ---
> > > > >
> > > > > ### Advantages over existing deep SSMs
> > > > >
> > > > > Compared to prior “deep SSMS that follow a more standard SSM formulation (i.e., *inputs  do not come from the past outputs*)”, we note three advantages:
> > > > >
> > > > > **(1) Simplicity + intuitive connection to the task**
> > > > > For time series forecasting, the task is predict a sequence of future values from a sequence of prior values (where values are from the same time series). We thus naturally model this task as a sequence-to-sequence mapping: between a sequence of past and future values as inputs and outputs.
> > > > > * This is not taken by prior SSMs as the general SSM also models settings with defined control inputs (i.e., in well-governed physical systems, or where explicit actions over time impact observations).
> > > > >   * However, with just time series data, we are only given a single sequence (no controls).
> > > > > * We could try to *infer* control inputs that impact the process (as additional dynamics) from the observed data, but this adds (difficult) modeling complexity (esp. for real-world time series data).
> > > > >   * In fact, many existing SSMs that stick with inputs as controls also avoid this. They simply ignore the control inputs for time series, c.f. Kalman Filters for time series analysis [1, 2]
> > > > >
> > > > > **(2) Fast + expressive training and inference**
> > > > > Treating inputs as past outputs also *enables* the fast training and inference in our framing (derived in Sec. 3.1.2, 3.2).
> > > > > * We uniquely get fast processing over input sequences, and fast recurrent forecasting over future values. (Sec 3.2 for forecasting, Eq. 6-9, Eq. 15 show this computation as parallelizable matrix multiplications on GPUs).
> > > > >
> > > > > * The prior methods only process inputs recurrently through the hidden state. Implemented as an RNN / LSTM, this requires more slowly processing inputs and outputs sequentially (even on GPUs, this stateful inference is a sequential / non-parallelizable operation).
> > > > >
> > > > > As empirical evidence, we implemented the referenced Deep Kalman Filters (DKF) [1] and Deep Variational Bayes Filters (DVBF) [3]. We compare the training time / epoch on ETTh1 data. For forecasting 96 future steps, SpaceTime is 7x and 12x faster than DVBF and DKF, respectively:
> > > > >
> > > > > ||DKF|DVBF|SpaceTime|
> > > > > |---|---|---|---|
> > > > > |Epoch sec.|264|155|21|
> > > > >
> > > > > **It's also unclear how to speed-up the prior SSM framings** Under our framing we treat SSM variables $\boldsymbol{A}, \boldsymbol{B}, \boldsymbol{C}, \boldsymbol{D}$ as distinct trainable parameters in a neural net layer, enabling fast computations derived in Sec. 3.1.2, 3.2. We make our model **more expressive** by stacking these layers, interspersing with standard nonlinearities (e.g., ReLU, Fig 2)
> > > > > * However, prior work makes the SSM more expressive by replacing the linear relations, e.g. $x_{t+1} = \boldsymbol{A}x_t + \boldsymbol{B}u_t$ with monolithic neural nets, e.g. $x_{t + 1} = f_\theta(x_t, u_t)$ [1, 3, 4].
> > > > > * Any recurrences or autoregressions thus need to pass through the entire model (we avoid this again with distinct parameters that enable the closed-loop SSM).
> > > > >
> > > > > **(3) Avoiding expensive variational inference**
> > > > > Finally, prior SSM frameworks frequently model noise terms in the transition Eq., i.e. $x_{t+1} = Ax_{t} + Bu_{t} + \epsilon_t$
> > > > > where noise is sampled from a 0-mean Gaussian with (learnable) covariance.
> > > > >
> > > > > We currently do not explicitly model this noise. However, we can still handle noisy data via the companion matrix (Sec. 3.1.3) (but computed as a deterministic mapping between inputs and outputs). This allows us to avoid relatively more expensive training and inference steps related to past deep SSMs, e.g., maximizing ELBO and Monte Carlo sampling [1, 3, 4].
> > > > > * These variational steps may lead to slower and more difficult training in prior deep SSMs. On the same ETTh1 96 task, we list MSE in the table below. We were unable to get prior SSM models to forecast this real-world data accurately:
> > > > >
> > > > > ||DKF|DVBF|SpaceTime|
> > > > > |---|---|---|---|
> > > > > |MSE|1.868|2.752|0.054|
> > > > >
> > > > > Meanwhile, despite not explicitly modeling noise terms, SpaceTime gets significantly better accuracy: up to 98% lower MSE.

---

> > > > > > ### Author Response · Authors · 2022-12-08
> > > > > > **Response (2/2) - clarification to Q2 on closed-loop SSM**
> > > > > >
> > > > > > Thanks for this great clarifying question. If our model has a *single* SSM layer, then you’re correct that we’d be able to get the same effect with your stated recurrence.
> > > > > >
> > > > > > However, the recurrence only applies for a single SSM. This breaks down when our neural networks have *multiple SSM layers*. We thus introduce the closed-loop component to handle this setting (where we scale up expressivity with multiple SSM layers in a deep network)[^1].
> > > > > >
> > > > > > To clarify why, please see additional details below:
> > > > > >
> > > > > > ---
> > > > > >
> > > > > > **Review of single SSM layer**
> > > > > > First, as review for the single SSM layer setting, recall that we get the same effect with your stated recurrence because for a model with a single SSM layer, the model’s output for a given time series input $u_k$ is the next time-step $y_{k+1}$ in the time series. Because the model only has one SSM layer, this is the same as a prediction for *that same SSM layer’s* next time-step input $u_{k+1}$ (a “closed loop”).
> > > > > >
> > > > > > **Dealing with multiple layers**
> > > > > > However, to clarify why this same recurrence does not simply apply to a multiple layer model, recall that **in the multi-layer setting**, the output of the model is the output of the *last layer’s* SSM (which is again a prediction for a sample time-step in the time series). However, the recurrence requires an SSM’s output to be a prediction for the *next input* of **that same last-layer SSM**, and this is no longer the case. Instead, the next time-step in the time series would be the *next input* for the **first-layer** SSM.
> > > > > >
> > > > > > Naively, without the closed loop recurrence we could compute the model output as a composition of multiple SSMs and nonlinearities given an input, e.g., for an $n$-layer model:
> > > > > > $$y\_{k + 1} = \hat{u}\_{k + 1} = \text{SSM}\_{n}(\text{SSM}\_{n-1}( \ldots \text{SSM}\_{1}(u\_{k}))$$
> > > > > >
> > > > > > **Problem**
> > > > > > Unfortunately, computing a fast recurrence for this composition is not trivial:
> > > > > >
> > > > > > * As discussed in Section 5, as a pure recurrence the sequential operation would not take advantage of parallel matrix multiplications.
> > > > > > * Furthermore, the fast computation as a convolution (Eq. 6 - 9) is only derived for a single SSM.
> > > > > >
> > > > > > **Solution**
> > > > > > Our closed-loop SSM approach gets around these limitations. We retain fast recurrence by explicitly also outputting predictions for the next time-step inputs of the *last-layer SSM*. This recovers the recurrence as we now again have a closed loop for a single SSM.
> > > > > >
> > > > > > To expand on the contributions here:
> > > > > >
> > > > > > * We recognized that this distinction was not addressed in prior related work (S4 [5]), which prevented flexible autoregressive forecasting.
> > > > > > * We proposed the closed loop contribution to address this gap, achieve these new time series capabilities (generalization and flexible forecasting of multiple horizons - Sec 4.2.2) and improve performance (Table D.6.1).
> > > > > >
> > > > > > We will clarify this distinction and motivation in Section 3.2 of the final paper.
> > > > > >
> > > > > > ---
> > > > > >
> > > > > > We thank you again for the questions and references, and will incorporate the responses in these two comments to further clarify our contributions in the final paper.
> > > > > >
> > > > > > We also think that this clarification motivated interesting further directions, such as efficiently modeling a noise component with our SSM framework. We will discuss this in the final paper as well, and think this could inspire future related work.
> > > > > >
> > > > > > If we have addressed your concerns, we again kindly ask that you consider raising your score. We are happy to provide further clarification.
> > > > > >
> > > > > > ---
> > > > > >
> > > > > > [^1] Recall that we use multiple layers with the usual motivation of deep neural nets. More layers can scale up the expressivity of our model (and the SSMs) in a deep architecture.
> > > > > >
> > > > > > ---
> > > > > >
> > > > > > **References**
> > > > > > [1] Deep Kalman Filters
> > > > > > [2] KalmanNet: Neural Network Aided Kalman Filtering for Partially Known Dynamics
> > > > > > [3] Deep Variational Bayes Filters: Unsupervised Learning of State Space Models from Raw Data
> > > > > > [4] Latent Matters: Learning Deep State-Space Models
> > > > > > [5] Efficiently Modeling Long Sequences with Structured State Spaces

---

> > > > > > > ### Comment · Reviewer_YRVv · 2022-12-12
> > > > > > > **Thanks**
> > > > > > >
> > > > > > > Thanks for the detailed response. My questions are answered. I'll raise my score.

---

### Official Review · Reviewer_hBe1 · 2022-10-25

**Confidence:** 3
**Correctness:** 3
**Technical Novelty And Significance:** 3
**Empirical Novelty And Significance:** 3
**Recommendation:** 5

**Clarity, Quality, Novelty And Reproducibility:**

The paper is well written but some claims and experiments seem not persuadable to me. I did not find the source code shared by this paper. Moreover, the overall framework is not introduced.

**Strength And Weaknesses:**

Strength
1. The experimental result is impressively good


Weaknesses
1. Using SSM in the seq2seq is sort of ad-hoc, lacks intuition, and just applies some existing techniques.
2. The simplified approach here (e.g. constrained A) is ad-hoc to me without sufficient explaination.
3. Some claims seem incorrect and the experimental result seems weird to me.

**Summary Of The Paper:**

The paper proposed a layer named SpaceTime. The proposed layer uses a State Space Model-based latent space structure. SpaceTime layers could achieve better performance in the task of time series forecasting and classification.

**Summary Of The Review:**

1. Some claims are wired and maybe even incorrect. For example, some claims like”  “Recent Transformer-based models reduce this complexity to O(ℓ+h), but do not always outperform the above FCNs on forecasting benchmarks”, but authors only cited an arXiv paper with limited citation. Moreover, the cited article seems like only focuses on ablation tests.

2. Every component in the proposed model is ad-hoc and lacks a reason to do that. For example, the model used SSM embedded into the autoregressive model. But what about using Kalman Filter structure? Or any other things such as gaussian process. I barely seen the reason why SSM is used here compared with using classical embedding space.

3. The simplified approach is also ad-hoc. The model uses a special A. I understand that it may have the ability to approximate a lot of dynamic systems, but that neither explain if we can ``learn’’ this solution nor why any formula is not working.

4. The Closed-loop Recurrent is a very interesting structure, but I have a hard time understanding why it could get a better fit compared with the original seq2seq-based decoding process. Is it increasing the ability to capture nonlinearity in the system? But transformers already have the ability to capture such nonlinearity. Then why did you achieve better performance? Any explanation?

5. Is there any reason why you haven’t compared with time series classification approaches such as Rocket[1]?

6. Besides, I believe an overall structure for the forecasting and classification model should be included in the supplement material.

7. In the comparison experiment, the best baseline is NLinear (LSTM + Conv). Any code, reason, or explanation? Since this result is conflicted with existing published work (e.g. Informer, NBeats, FEDFormer results because they claim that they are better than LSTM because it cannot capture long-term data dependency). As a rule of thumb, I am led to believe existing published work compared with the result in this paper.

Overall, the experiment is impressively good. But given all the concerns here, I believe the proposed structure is ad-hoc and how to get such an impressive performance is not convincing.

---

> ### Author Response · Authors · 2022-11-17
> **Author Response (1/4)**
>
> Thank you for your review. In our responses below, please find clarifications on (1) how our work is not ad-hoc, and (2) how our claims and experiment results are correct.
>
> On **ad-hocness**, we first expand on our general response to show why our work is well-motivated by limitations in prior work. We then describe intuitions for how our contributions overcome these prior issues. Our methods are not merely applications of existing techniques, but alternative proposals motivated by past frameworks that solve fundamental limitations. These are general advances for both deep learning for time series modeling and SSM-based deep learning.
>
> We also address individual comments on the benefits of the closed-loop forecasting SSM, and details on the overall architectures.
>
> On **claim and experiment validity**, we provide additional details below. Hopefully this clears up misunderstandings with our results and baseline comparisons. In particular, we clarify that the prior state-of-the-art *does not* contradict past Transformer results, as it **not an LSTM** (as previously claimed by the reviewer). Furthermore, **our method frequently outperforms *both* the prior state-of-the-art and prior Transformer architectures** (e.g., in Table 1, we get best forecasting MSE on 14 / 16 tasks).
>
> **Adhoc 1: Grounding and intuition for SSM in time series sequence-to-sequence modeling**
> As described in our introduction (page 2) and theoretically proved in Proposition 1 (Appendix B.1), the SSM is a powerful and general structure that captures many fundamental models used in time series modeling. These include ARIMA, exponential smoothing, and the *suggested Kalman Filter*. Hence there are strong historical and mathematical roots for building a model around SSMs.
>
> Furthermore, a recent line of work (e.g., S4, see general response) shows that SSM-based sequence-to-sequence (seq2seq) models can lead to large improvements over Transformers and LSTMs on a wide range of sequence modeling tasks and modalities [1, 2].
>
> As many recent deep learning works involving Transformers and LSTMs cast time-series forecasting as a seq2seq problem [3, 4], this suggests that SSMs can also be useful structures for our setting.
>
> **Adhoc 2: Motivation for our contributions**
> However, we *do not* simply apply existing ideas for time series modeling. To motivate our work, we rigorously demonstrate that these prior state-of-the-art deep SSM-based architectures carry fundamental limitations for time series modeling. In our paper, we:
> * Discussed how prior deep SSMs cannot provably model autoregressive patterns in time series data, due to their specific continuous-time or diagonal state representations (Section 3.1.1)
> * Proved these limitations in Proposition 2 (Section 3.1.1, proof in Appendix B.1)
> * Showed that these limitations hold in practice via extensive controlled experiments (Section 4.2.1): prior deep SSM-based networks cannot learn the ground-truth models behind synthetic ARIMA processes through our transfer function analysis (Figure 3).
>
> **Adhoc 3: Ties between our proposed companion SSM contribution and overcoming prior work’s limitations**
> Our first core contribution is thus directly tied to how to get around these limitations:
> 1. We theoretically proved that how we model the SSM gets around these limitations in Proposition 1 (Appendix B.1).
> 2. We showed that our companion SSM leads to state-of-the-art performance on popular benchmarks (where prior related SSMs did not, c.f. Table 1). We also empirically demonstrate increased expressivity over prior deep SSMs via controlled ARIMA synthetics (Section 4.2.1), where our SSM uniquely enables learning the ground-truth processes (Figure 3).

---

> > ### Author Response · Authors · 2022-11-17
> > **Author Response (2/4)**
> >
> > **Adhoc 4: Intuition behind “special A” companion matrix to overcome prior SSM limitations**
> > To clarify why the companion matrix is not “ad-hoc”, we explain why prior choices for A are problematic, and how we arrived at the companion matrix solution. We also discuss how the companion matrix is a general solution that for the first time enables the SSM framework presented in prior work to provably learn autoregressive processes.
> >
> > In Section 3.1.1, we described key criteria for how to parameterize the SSM. **For intuition, we noted that we should be able to express autoregressive processes using the SSM**. This is because time series processes are inherently autoregressive (future samples are dependent on past samples). Thus we should be able to learn the specific matrix weights behind an SSM that specifies the ground-truth time series processes.
> >
> > However, as discussed above and in Section 3.1.1, **we proved that prior SSM-based deep learning models could not do this; the core culprit is how they parameterize the A matrix** in their architectures (see Proposition 2).
> >
> > Expanding on this point, note that to model an autoregressive process with an SSM, we require that A have the offset diagonal structure (see Eq. 5). We can see this by walking through the SSM equations (1) and (2) and give an example with the AR(p) process in Section 3.1.1; prior time series analysis texts also note this structure when presenting SSMs for classical time series models (e.g., ARIMA processes, see Hamilton 1994 [5]).
> >   * *However*, this form is not necessarily learnable via prior SSM representations; these prior representations either (i) use continuous-time SSMs, but intuitively the solution to these SSMs (a differential equation with some $e^Ax$ term) cannot represent this offset diagonal, or (ii) use diagonal SSMs, but the offset diagonal A matrix is not necessarily diagonalizable. We discuss these limitations in Section 3.1.1 when introducing Proposition 2.
> >
> > Luckily, **in our example in Section 3.1.1 we describe one simple solution**.  We can achieve this autoregressive SSM by simply instantiating A with the offset diagonal structure, i.e., setting A as the companion matrix[^1].
> >
> > As opposed to being ad-hoc, **the companion matrix is the most general class of real matrices that carries this desirable property** (offset diagonal structure) (note that for a $d \times d$ matrix, the “a column” can be any real-valued vector $\in \mathbb{R}^{d \times 1}$). Multiple textbooks also reference the companion matrix as a “canonical form” for representing discrete state-space models (see Hamilton 1994 [5]).
> >
> > [^1]: Note that we also discuss how we cannot in general use any matrix for A, as prior work demonstrates that specific matrices are needed to make the SSM practical to train (Gu et al., ICLR 2022 [2]). However, we also choose the companion SSM because we can provide the necessary machinery to make the companion SSM practical to train (and we do this via an efficient training and inference algorithm as our third core contribution).
> >
> > **Adhoc 5: “This neither explains if we can learn this solution, nor why any formula is not working”**
> > We think this is a misunderstanding of our results. In Proposition 1, **we proved that we can represent autoregressive processes *exactly* with our proposed companion matrix SSM**. To learn the ground-truth autoregressive solution, we just need to find the right matrix values for A, B, C, and D. As explained in Section 2, we can do this by setting A, B, C, D as trainable parameters in a neural net layer, and optimizing these parameters via gradient descent. Later in Section 4.2.1, **we validate that we *do* learn the solution in practice (Figure 3)**.
> >
> > Meanwhile, **we also proved that this cannot be done via prior SSM parameterizations in Proposition 2, due to their specific choices of A matrices**. Similarly, we demonstrated this inability *in practice* via our controlled synthetic experiments. We tested whether the simplest version of our architecture (1 layer, 1 SSM) could learn AR solutions in practice. Our results (Figure 3) confirmed this, while we also found that alternate “formulas” for the SSM could not learn such AR solutions.

---

> > > ### Author Response · Authors · 2022-11-17
> > > **Author Response (3/4)**
> > >
> > > **Benefit of closed-loop recurrent component over prior seq2seq decoding**
> > > In Section 3.2, we stated the issues with prior seq2seq approaches for time series forecasting.  The primary issues we found were (1) generalization to new horizons, or (2) inefficient inference over long horizons. Our closed-loop recurrent SSM is designed to directly overcome both of these limitations.
> > >
> > > On (1), as we described on page 6, for popular time series forecasting benchmarks, Transformers and the prior state-of-the-art method (a linear fully connected network) [6] train to forecast specific horizons; they learn a seq2seq mapping from a sequence of historical time-steps to a *specific length* of future time-steps. However, *because of this explicit supervision for a specific horizon*, these models may not actually learn the underlying time series process, and merely the seq2seq map to a specific horizon. We validate this hypothesis for the prior state-of-the-art method and show poorer generalization to other horizons vs our method in Section 4.2.2 and Figure 4.
> > >
> > > We can get around this by training to forecast the next time-step, recurrently using our predicted outputs as future inputs (and thus training to learn the true autoregressive process).
> > >
> > > However, doing this naively brings up issue (2). We also discussed this on page 6, where without the closed-loop SSM we would need to perform inference through the entire model, for every time-step in the horizon, using the “convolutional view” of the SSM (Eq. 13).
> > >
> > > Our closed-loop recurrent component allows us to avoid this inefficient inference through the entire model by isolating this step to the last layer, and enabling the last layer to output future time-step predictions similarly to an RNN. We clarify that the key idea is that, given a time series we want to model, we can train the last layer to output not just predictions for future time-steps of that time series (as normal), but also the last layer’s future *inputs* (that would lead to accurate predictions for that time series, once processed through the last layer). Thus we are explicitly training the last layer to predict its future inputs, introducing a “closed-loop” setup. Because the last layer predicts its own inputs, we do not need to interact with the rest of the model to recurrently generate future time-step predictions, allowing us to do this efficiently.
> > >
> > > To further demonstrate the benefits of this closed loop component, in our revision we include an additional ablation in Appendix D.6 (Table 10) where we remove this component. We hypothesize that the closed-loop recurrent approach allows us to better learn the true autoregressive process and not overfit to predicting a specific horizon. We validate this by noting that when we train our model without the closed-loop recurrent layer, we obtain less accurate forecasting:
> > >
> > > |                                   |  ETTh1 (720)  |  ETTh2 (720)  |  ETTm1 (720)  |  ETTm2 (720)  |
> > > |-----------------------------------|:-------------:|:-------------:|:-------------:|:-------------:|
> > > | SpaceTime (Default)               | 0.076 ± 0.006 | 0.188 ± 0.013	 | 0.074 ± 0.000 | 0.166 ± 0.001 |
> > > | SpaceTime (No closed-loop SSM)    | 0.114 ± 0.026 | 0.278 ± 0.027 | 0.156 ± 0.069 | 0.194 ± 0.026 |
> > > | S4 (Default - no closed-loop SSM) |     0.190     |     0.630     |     0.254     |     0.482     |

---

> > > > ### Author Response · Authors · 2022-11-17
> > > > **Author Response (4/4)**
> > > >
> > > > **Comparison to Rocket for time series classification**
> > > > Thanks for the suggestion. We think Rocket is an interesting time series classification method, and tried to run it for our ECG and Speech Audio classification tasks. However, we were unable to obtain satisfactory results. To explain why, we note several limitations. First, the available Rocket code (https://github.com/angus924/rocket) is designed for univariate time series; however ECG is multivariate (App. C.3). Second, using the available code, we were unable to get Rocket to run with practical compute constraints on these tasks without the Numba package (due to nested for loops over each sample and kernel). We tried to extend the Numba implementation to the multivariate setting, which is a specific dependency of Rocket not required by other methods. However, we have not been able to get this to work on our machine (A100 supplied by GCloud).
> > > >
> > > >
> > > > **Structure for forecasting and classification model in the supplementary material**
> > > > In our revision, please find detailed architecture diagrams for the forecasting and classification models in Appendix D.7. Our forecasting architecture is also depicted in Figure 2. We describe the simple and standard techniques to adapt this for classification in Section 4.1, (ECG (multi-label classification) paragraph.)
> > > >
> > > > **“In the comparison experiment, the best baseline is NLinear (LSTM + Conv). Any code, reason, or explanation? … This result is conflicted with existing published work”**
> > > > We believe there are several misunderstandings here. First, NLinear is **not** an LSTM + Conv, but rather a fully-connected network with input size = lag (or number historical time-steps), and output size = horizon (number of future time-steps to forecast) [11]. We discuss this on page 2 and Related Work (Appendix A.2). Such an architecture can thus capture long-term data dependencies, as each future time-step is modeled as a linear combination of every historical sample in the input.
> > > >
> > > > This result is thus **not conflicted with existing published work** describing the benefits of capturing long-range dependencies for time series forecasting, as *NLinear does densely capture long-range dependencies*.
> > > >
> > > > Regarding further reasons or explanation, the NLinear authors state that the success of Transformers on time series forecasting benchmarks may just be due to learning to forecast the entire horizon with these fully connected dependencies between prior time-step inputs and future time-step outputs, rather than anything unique to Transformers or attention. They test and verify this hypothesis with the fully-connected network, and demonstrate improved results.
> > > >
> > > > On code, we reproduced their results on the Informer benchmark using the publicly available code (https://github.com/cure-lab/LTSF-Linear). Our results vs the reported numbers on the Informer benchmark are in the Table below.
> > > >
> > > > **“Some claims are wired and maybe even incorrect.”**
> > > > We hope that the above clarifications have clarified that our claims are not incorrect.
> > > >
> > > > Specifically on the concern that the prior state-of-the-art is an "arXiv paper with limited citation.”, we report this merely as faithful acknowledgement of recent developments in the field. Importantly, we reproduced their numbers using their publicly available code.
> > > >
> > > > **“I did not find the source code shared by this paper”**
> > > > We uploaded the source code in our supplementary materials (as acknowledged by Reviewer 61Ty).
> > > >
> > > > | Dataset | Horizon | NLinear (Paper) MSE | Our Repro (5 seeds) MSE | NLinear (Paper) MAE | Our Repro (5 seeds) MAE |
> > > > |---------|---------|:-------------------:|:----------------------:|:-------------------:|:----------------------:|
> > > > | ETTh1   |      96 |0.053|      0.053 ± 0.000     |0.177|      0.177 ± 0.001     |
> > > > | |     192 |0.069|      0.070 ± 0.001     |0.204|      0.205 ± 0.001     |
> > > > | |     336 |0.081|      0.081 ± 0.000     |0.226|      0.225 ± 0.000     |
> > > > | |     720 |0.080|      0.080 ± 0.000     |0.226|      0.227 ± 0.001     |
> > > > | ETTh2   |      96 |0.129|      0.131 ± 0.002     |0.278|      0.279 ± 0.002     |
> > > > | |     192 |0.169|      0.176 ± 0.012     |0.324|      0.330 ± 0.011     |
> > > > | |     336 |0.194|      0.195 ± 0.001     |0.355|      0.356 ± 0.000     |
> > > > | |     720 |0.225|      0.225 ± 0.001     |0.381|      0.381 ± 0.001     |
> > > > | ETTm1   |      96 |0.026|      0.026 ± 0.000     |0.122|      0.121 ± 0.000     |
> > > > | |     192 |0.039|      0.038 ± 0.000     |0.149|      0.149 ± 0.000     |
> > > > | |     336 |0.052|      0.051 ± 0.000     |0.172|      0.172 ± 0.000     |
> > > > | |     720 |0.073|      0.072 ± 0.000     |0.207|      0.206 ± 0.000     |
> > > > | ETTm2   |      96 |0.063|      0.064 ± 0.003     |0.182|      0.185 ± 0.006     |
> > > > | |     192 |0.090|      0.090 ± 0.000     |0.223|      0.223 ± 0.000     |
> > > > | |     336 |0.117|      0.116 ± 0.000     |0.259|      0.258 ± 0.000     |
> > > > | |     720 |0.170|      0.170 ± 0.001     |0.318|      0.319 ± 0.001     |

---

> > > > > ### Author Response · Authors · 2022-11-17
> > > > > **References**
> > > > >
> > > > > 1. A. Gu, I. Johnson, K. Goel, K. Saab, T. Dao, A. Rudra, and C. R ́e. Combining Recurrent, Convolutional, and Continuous-time Models with Linear State Space Layers. NeurIPS 2021.
> > > > > 2. A. Gu, K. Goel, and C. R ́e. Efficiently Modeling Long Sequences with Structured State Spaces. ICLR 2022.
> > > > > 3. H. Zhou, S. Zhang, J. Peng, S. Zhang, J. Li, H. Xiong, and W. Zhang. Informer: Beyond Efficient Transformer for Long Sequence Time-Series Forecasting. AAAI 2021.
> > > > > 4. M. Liu, A. Zeng, M. Chen, Z. Xu, Q. Lai, L. Ma, Q. Xu. SCINet: Time Series Modeling and Forecasting with Sample Convolution and Interaction. NeurIPS 2022.
> > > > > 5. J. D. Hamilton. State-space Models. Handbook of Econometrics, 4:3039–3080, 1994.
> > > > > 6. A. Zeng, M. Chen, L. Zhang, and Q. Xu. Are Transformers Effective for Time Series Forecasting? arXiv 2022.

---

> > > > > ### Comment · Reviewer_hBe1 · 2022-11-18
> > > > > **Response to the authors**
> > > > >
> > > > > Thank you very much for the valuable feedback and clarification. The discussion helps me understand the paper a lot. However, I still believe the technical contribution is not meet the standard of ICLR. Here are the major concerns:
> > > > >
> > > > > 1. Deep State Space originally is invented in the time series forecasting task, while recently, they start to use in a lot of more wide tasks, it is a well-known baseline in time series. As a result, I believe it is natural to work to model: ARIMA, exponential smoothing, and controllable linear time-invariant systems. The authors reinvent the Deep State Space model into time series modeling based on the newly proposed model such as S4, and LSST. So the logic seems like the newly proposed model does not provide any theoretical advance. For example, any non-stationary evolving system can be modeled by the newly invented model?
> > > > >
> > > > > 2. While I understand that they are indeed different (through the way it combines with deep learning structure), it just uses the framework proposed by previous work with a newly proposed companion matrix fashion. Therefore, while it is better than I expected, I still think the technique contribution in the time series modeling part is incremental (Note: the paper has some technique contribution in the efficient algorithm part). Moreover, is any arbitrary companion matrix other than the diag matrix that could make S4 and LSST work?
> > > > >
> > > > > Overall, I believe that this paper has some degree of contribution, but in the theoretical part, it is not enough to reach the acceptance bar of ICLR.
> > > > >
> > > > > Related to NLinear baseline:
> > > > > Note: Discussion related to benchmark, NLinear, is not related to my paper evaluation
> > > > >
> > > > > It is interesting to see the experiment result, again, my doubt is the idea of why this result is contracted to the claim of existing work (not reproduce this result, since we always have more time series data). is there any reason that can justify the claim that a fully connected neuron network can beat a transformer in time series other than some assumption?

---

> > > > > > ### Author Response · Authors · 2022-11-18
> > > > > > **Response on theoretical advances + technical contributions**
> > > > > >
> > > > > > Thanks for the quick response!
> > > > > >
> > > > > > First, we're glad that you recognize our companion SSM lets us recover classical time series models (ARIMA, ETS), building on powerful deep SSM-based models like S4 / LSSL.
> > > > > >
> > > > > > We agree this is a simple and natural idea in hindsight, motivating our work! But it's not at all obvious how to actually do this. Overcoming this gap is where our theoretical advances + technical contributions come in.
> > > > > >
> > > > > > **Theoretical challenges**
> > > > > > There are many *possible* ways to represent SSMs and SSM-based deep learning models. As you mention, some are well-known baselines in time series. But simply put, these prior SSM models *didn’t work* (vs. arguably less-principled Transformer-based models (Table 1, Table 7)).
> > > > > >
> > > > > > * Thus the choice of how to represent an effective SSM, and even how one represents the $\boldsymbol{A}$ matrix, has been a deeply studied and open question[^1]. How can we make these models both *expressive + accurate*, and *numerically stable + efficient to train*?
> > > > > >
> > > > > > Models like LSSL and S4 proposed one (complicated) framework for effective SSMs (see footnote [^1]), where specific $\boldsymbol{A}$ matrices were needed to also make these SSMs numerically stable and efficient.
> > > > > >
> > > > > > However, despite all this prior work, our theory surprisingly shows that these prior SSM representations *cannot* model the natural + intuitive time series models (ARIMA, ETS, etc.)
> > > > > >
> > > > > > **Review of theoretical advances**
> > > > > > **(1)** Thus first, our theoretical advances with the companion matrix overcame this expressivity issue, as yet another advance on how to parameterize SSMs.
> > > > > >
> > > > > > **(2)** Second, it was not clear that the companion matrix would still be numerically stable or efficient. We provided new algorithms and *proved* that we could make the companion SSM *stable and efficient*.
> > > > > >
> > > > > > **Implications + technical contributions**
> > > > > > **(1)** While simpler than prior approaches, our simpler and well-motivated companion $\boldsymbol{A}$ matrix enabled these classic time series capabilities, and could get state-of-the-art results.
> > > > > > * The prior $\boldsymbol{A}$ matrices again relied on complicated theory and / or were disconnected from time series principles, under the premise that the specific $\boldsymbol{A}$ were necessary for effective SSMs. We showed simpler alternatives are possible.
> > > > > >
> > > > > > **(2)** We also extended the prior SSM frameworks by being the first to introduce the closed-loop SSM.
> > > > > > * We already saw big improvements with the companion matrix, but this wasn’t enough for consistent state-of-the-art performance or new time series capabilities.
> > > > > > * Thus we introduced the closed-loop SSM, which further improves forecasting accuracy (Table 11, App. D.6.2), and enables generalization to different forecasting horizons (Figure 4, Sec. 4.2.1).
> > > > > >
> > > > > > We hope this addresses your concerns. If not, we are happy to answer any remaining questions. We hope you may consider raising your score accordingly.
> > > > > >
> > > > > > ---
> > > > > >
> > > > > > [^1] For example:
> > > > > > * LSSL proposed a complicated class of matrices relying on a complicated theory of “high-order polynomial projection operators” (HiPPO) (NeurIPS 2021) [1]
> > > > > > * S4 proposed an alternative “diagonal plus low-rank” representation of HiPPO matrices (ICLR 2022)
> > > > > > * More recent work (DSS, NeurIPS 2022 [2]; S4D, NeurIPS 2022 [3]) tried to get simpler versions of the LSSL SSM to work, proposing diagonal $\boldsymbol{A}$ matrices and providing additional theory on why such representations were important for effective SSM-based models.
> > > > > > ---
> > > > > > 1. Gu et al. HiPPO: Recurrent Memory with Optimal Polynomial Projections. NeurIPS 2020.
> > > > > > 2. Gupta et al. Diagonal State Spaces are as Effective as Structured State Spaces. NeurIPS 2022.
> > > > > > 3. Gu et al. On the Parameterization and Initialization of Diagonal State Space Models. NeurIPS 2022.
> > > > > >
> > > > > > ---
> > > > > > **Addressing NLinear**
> > > > > > Finally, we think more rigorous understanding of why an FCN can outperform Transformers in time series is an interesting research question.
> > > > > >
> > > > > > In general, we are excited for future work on whether the attention mechanism is necessary for sequence modeling in time series and other domains.
> > > > > >
> > > > > > Our work on SpaceTime suggests models with more efficient operations can outperform Transformers (and FCNs) on time series, but we think this thread is relatively underexplored.

---

### Official Review · Reviewer_61Ty · 2022-10-30

**Confidence:** 3
**Correctness:** 4
**Technical Novelty And Significance:** 4
**Empirical Novelty And Significance:** 3
**Recommendation:** 8

**Clarity, Quality, Novelty And Reproducibility:**

Clarity: extremely clear and easy to read and understand

Quality: high quality description of method with extensive proofs of claims, and substantial and convincing experiments.

Novelty: this seems to be a novel development of SSM-based architectures with improved modeling capabilities and efficiency for training and inference.

Reproducibility: code to replicate results included in supplementary material.

**Strength And Weaknesses:**

### Strengths

1. The paper is very clear, thorough, and theoretically sound. Proofs are provided for all propositions, and they seem correct as far as I can tell.

2. The proposed architecture seems quite effective, and the results show that the approach has impressive empirical performance. The architecture has desirable qualities compared to other similar architectures, and the paper does a good job about discussing these comparisons.

### Weaknesses

1. It would have been nice to see the approach applied to some other datasets, e.g. more speech or audio-based datasets. That said, the existing evaluations are quite extensive, so this is a good direction for future work.

2. I have a few minor comments (see below)

### Minor comments

1. Using m for the number of SSMs per SpaceTime layer is a little confusing (Architecture section of section 3.1.2), because it is also used as the feature dimension earlier.

2. It would be good to provide a few more details about the speech classification task in the "Speech Audio (single-label classification)" section in section 4.1. Also, the section mentions that "SpaceTime outperforms prior SSMs (LSSL), Transformers, and specialized architectures (Table 3)", but LSSL and Transformer don't have results listed in Table 3.

3. "length-wise pooling": maybe say mean-pooling over length?

4. In section 4.2.2, NLinear is mentioned without much detail. Maybe good to add a little more detail?

5. Caption in Figure 4 referes to "DLinear"; should this be "NLinear"?

6. In proofs in Appendix B for Proposition 1, maybe better to use numbers or letters instead of bullets to refer to the different cases (i.e. ARIMA, Exponential smoothing, Controllable LTI systems)

7. A couple additional references that might be good to add to the "Transformers" section in A.2:

    Conformer (https://arxiv.org/abs/2005.08100)

    Perceiver AR (https://arxiv.org/abs/2202.07765)

8. "Moreover, embarrassingly simple alternatives, such as a single linear layer, have shown to outperform these highly specialized transformers": is there a citation for this?

**Summary Of The Paper:**

This paper proposes a new architecture called SpaceTime, which is a state-space model (SSM) architecture that is based on the companion matrix, and is able to learn to model autoregressive processes. This architecture is applied to both time series classification and forecasting. A "closed-loop" variation of the approach is introduced for long-term forecasting where the architecture generates its own layer-wise inputs. The paper also proposes an algorithm for efficient training and inference for the architecture, which reduces computation from the naive O(dl) to O(d + l), where d is state-space size and l is sequence length.

The SpaceTime approach achieves competitive performance on ECG and speech time series classification and on Informer forecasting tasks. Better qualitative forecasting results are shown for long time horizons, and the architecture is shown to achieve 73% and 80% faster training in terms of wall-clock time compared to Transformers and LSTMs, respectively.

**Summary Of The Review:**

A new SSM-based architecture that has improved modeling capability compared to previous SSM-based models. Claims are well-supported with extensive proofs, and evaluation on time series classification and forecasting are substantial and convincing. The paper is very clear, and code is provided for reproducibility. Overall, an excellent paper that contributes a novel improved SSM architecture that improves on state-of-the-art for time series analysis.

---

> ### Author Response · Authors · 2022-11-17
> **Author Response (1/1)**
>
> Thank you for your review! We are glad that you appreciated our contributions for improving SSM-based architectures, and are happy to hear that you found our paper easy to read and understand.
>
> Many thanks as well for your helpful suggestions and feedback. We have updated the revision with your suggestions.
>
> **Extended datasets and modalities**
> First, we agree that extending our work to more datasets and modalities would be an exciting future direction. We believe recent SSM-based models have a lot of potential for learning from new modalities, especially those with long-range dependencies (e.g., raw speech audio, biological sequences, and more?). We’re excited to see how the ideas in this work for improved autoregressive modeling with deep SSMs could benefit these directions, either out-of-the-box or as one thread to motivate follow-up work.
>
> **Minor comments**
> Second, we updated our revision following your suggestions (changes in blue). We believe these address your comments. Thank you again for your careful review! Please see details on each below.
>
> 1. In Section 3.1.2, we changed the notation from “m” to “s”.
> 2. In Section 4.1, we added additional details on the Speech Audio tasks.
>   * To clarify, for classification we use the same mean pooling over length before a softmax to output class logits as in ECG.
>   * We also clarify that the “X” in Table 3 denotes that LSSL and Transformers were computationally infeasible to run on available GPU resources. We use the same GPU for all methods (please see “Hardware details” in Appendix C). In particular, we note that the Speech audio task requires processing inputs with 16,000 time-steps. This cannot fit in a Transformer’s input context (recall memory scales quadratically with length). We also note this is infeasible to run with LSSL’s impractical SSM formulation (S4 makes the SSM framework in LSSL practical to train [1, 2])
> 3. We changed “length-wise pooling” to “mean pooling over length”.
> 4. In Section 4.2.2, We have clarified that NLinear is an FCN that learns a dense linear mapping between every lag input and horizon output.
> 5. Thanks for the catch! We have corrected this.
> 6. We updated the proof formatting for Proposition 1 (Appendix B.1) to include numbered indices for each case.
> 7. Thanks for bringing these to our attention. We have updated the Transformers section in Appendix A.2 with these references.
> 8. Yes, we were referring to Zeng et. al. 2022 [3] who propose NLinear. However we have updated the language to be more objective.
>
> **References**
> 1. A. Gu, K. Goel, and C. R ́e. Efficiently Modeling Long Sequences with Structured State Spaces. ICLR 2022.
> 2. A. Gu, I. Johnson, K. Goel, K. Saab, T. Dao, A. Rudra, and C. R ́e. Combining Recurrent, Convolutional, and Continuous-time Models with Linear State Space Layers. NeurIPS 2021.
> 3. A. Zeng, M. Chen, L. Zhang, and Q. Xu. Are Transformers Effective for Time Series Forecasting?

---

### Author Response · Authors · 2022-11-17
**General Response + Revision Upload (1/3)**

We thank the reviewers for their helpful comments and clarifying feedback.

First, we appreciate that reviewers consistently noted several positives of our work:
* Reviewers noted our impressive empirical performance [hBe1], substantial and convincing experiments [61Ty], and rigorous empirical analysis [BK6G] across a large number of forecasting and classification tasks and stated goals [61Ty, YRVv].

* Reviewers also appreciated our novel SSM formulation [YRVv, BK6G] and new and interesting SSM-based neural net architectures for time series forecasting [61Ty, hBe1].

* Finally, Reviewer 61Ty recognized our high quality description of method, theoretically sound and extensive proofs of claims, and further desirable qualities and modeling capabilities of our approach, including:
  * More accurate long horizon forecasting over prior state-of-the-art.

  * More efficient training and inference, resulting in 73% and 80% faster training in terms of wall-clock time compared to Transformers and LSTMs.


In our responses to individual reviews, please find clarifying details on how our contributions enable the noted impressive empirical results, and why our claims and evaluation baselines are valid. We thank reviewers for their suggestions and feedback; in our revision we have updated the paper to incorporate this feedback. We discuss details in individual responses.

Finally, in response to Reviewers hBe1, YRVv, and BK6G, **please also find shared responses** in the following thread that clarify the grounding of our contributions and their impact. We highlight that to **(1)** improve the modeling quality, **(2)** speed up training + inference, and **(3)** enable new time series modeling capabilities of state-of-the-art time series architectures, our work builds on several connections to recent *and* classical SSM developments.

---

> ### Author Response · Authors · 2022-11-17
> **Grounding + impact on recent SSM developments (2/3)**
>
> On recent developments, our SSM formulation builds on and improves the framework used in modern SSM-based deep learning models such as LSSL (Gu et al. 2021a, NeurIPS 2021) [1] and S4 (Gu et al. 2021b, ICLR 2022 Outstanding Paper Honorable Mention) [2]. Our contributions building on this connection directly lead to improved modeling quality: **SpaceTime outperforms the S4 SSM with 44.6% to 96.0% lower MSE in 42/45 Informer time series forecasting tasks**, while maintaining fast training and inference speed: **SpaceTime trains up to 4x faster than Transformers** (Table 5). We referenced these works in our Methods (Sec. 3), but make this connection more explicit in our Introduction (Sec. 1) and Preliminaries (Sec. 2).
>
> To expand on this connection, we first note that S4 showed that SSM-based models could substantially outperform Transformers and LSTMs on long-range *image and text* sequence modeling tasks, achieving 96.4% accuracy on a task (Path-X) that no prior deep learning architecture could perform better than chance on—see Long Range Arena (LRA) (Tay et al. ICLR 2021) [3]. They did so while also training up to 5x as fast as Transformers (Table 3 in [2]). These exciting results inspired multiple follow-up SSM-based deep architectures (see our citations in Section 3.1.1, top of page 5).
> * However, these LRA evaluations do not consider *time series data*. On popular time series benchmarks, we (and others) found that prior general sequence SSM-based models like S4 are outperformed by the most recent time series-specific Transformer models (see Table 1, Table 7) [^1].
>
> Motivated by this, we are the first to investigate *why* these modern SSM-based models do not perform as well on time series.
>
> From this investigation, **we uncover fundamental limitations** for autoregressive modeling with the prior SSM frameworks (Proposition 2). Our first core contribution thus directly overcomes these limitations with a new SSM parameterization (the companion matrix).
>
> Finally, to see why this not just an ad-hoc addition to an existing architecture, note that:
> * The prior SSM works such as S4 propose to parameterize the SSM *one way*, and we propose an *alternative* (the companion matrix) that improves autoregressive modeling with extensive theoretical and empirical evidence:
>   * For theory, see Propositions 1, 2 in Section 3.1.1 (proofs in Appendix B.1)
>   * For experiments, see Tables 1, 7, 8 (we outperform S4 in 42 / 45 Informer forecasting settings), and Section 4.2.1, Figure 3 (we uniquely learn ground-truth autoregressive processes that S4 cannot).
> * In addition, while this contribution is motivated by time series modeling, **improving these modern SSM’s autoregressive capabilities is a general advance** widely applicable to any causal modality.
> * Furthermore, our parameterization gives these SSM-based models **new time-series modeling capabilities**, e.g., inherent recovery of classic time series analysis models (Section 4.1.2, Figure 3) and data preprocessing techniques (Section 3.1.3).
>   * Prior works showed that deep architectures can be outperformed by classic time series models [4], motivating more recent works to add these time series elements on Transformer backbones in hybrid architectures [5, 6].
>    * In contrast, we show how **our SSM can simply and inherently recover these classic models** in theory (Proposition 1) and practice (Figure 3).
>
> [^1]: We note that other prior SSM-based models such as DeepAR (cited in Section 3.2) also explored SSM + deep learning models before S4. However, S4 significantly outperforms DeepAR on all 25 evaluation settings in the popular Informer benchmark (Table 7). Thus we focused on building on the S4 SSM framework to improve time series forecasting accuracy.

---

> > ### Author Response · Authors · 2022-11-17
> > **Grounding on classical SSM developments (3/3)**
> >
> > On classical developments, we reiterate on two connections. First recall that **SSMs have historical and rooted ties to classical time series analysis** [^1], which we hypothesized could be beneficial when scaled up into a modern deep learning architecture (and validated as above via improved modeling quality).
> >
> > Second, our **closed-loop SSM layer is the first to draw connections to closed-loop control systems** and implement the “RNN view” of SSMs for improved long-range forecasting. This is analogous to the open-loop systems and “CNN view” explored in prior related SSM-based architectures [1].
> >
> > However, the prior related SSMs (e.g., S4) only use the open-loop SSM. While **we maintain the speed** of these prior SSMs for forecasting long time series horizons (SpaceTime and S4 train **up to 10x faster than LSTMs**; see Figure 8), we **improve forecasting accuracy with up to 96% lower MSE** compared to the S4 SSM (Table 1). In our revision, we also show that closed-loop SSM specifically improves forecasting accuracy. In an ablation we remove the closed-loop component (Appendix D.6.1), and show that the closed-loop SSM improves relative MSE by 33.6% (while the SpaceTime without the closed-loop SSM still improves relative MSE by 48.6% over the prior S4 SSM; table reproduced below).
> >
> > * **We are the first to implement this closed-loop connection**, where to efficiently “rollout” predictions in a recurrent manner (i.e., not having to feed final model outputs back through the entire network as first-layer inputs), we explicitly train last-layer SSMs to also output their future layer-specific inputs.
> > * Furthermore, **we also show how the closed-loop SSM naturally computes this rollout in practice much more quickly than a naïve RNN** (Section 3.2). Eq. 14 and Eq. 15 naturally follow from the SSM (Eq. 1, Eq. 2), and let us compute future time-step predictions using a convolution as in Eq. 8 (i.e., without relying on sequential recurrence).
> >   * Combined with our efficient matrix powering algorithm (Algorithm 1), this leads to training times up to 10x faster than LSTMs when scaling up forecasting horizon length (Figure 8).
> >   * Meanwhile, we consistently achieve 3-20% lower relative MSE than the prior state-of-the-art (across *all methods*, not just SSMs) on long horizon forecasting benchmarks (Table 4).
> >
> > |                                   |  ETTh1 (720)  |  ETTh2 (720)  |  ETTm1 (720)  |  ETTm2 (720)  |
> > |-----------------------------------|:-------------:|:-------------:|:-------------:|:-------------:|
> > | SpaceTime (Default)               | 0.076 ± 0.006 | 0.188 ± 0.013	 | 0.074 ± 0.000 | 0.166 ± 0.001 |
> > | SpaceTime (No closed-loop SSM)    | 0.114 ± 0.026 | 0.278 ± 0.027 | 0.156 ± 0.069 | 0.194 ± 0.026 |
> > | S4 |     0.190     |     0.630     |     0.254     |     0.482     |
> >
> > [^1]: e.g., ARIMA models, Kalman Filters, Exponential Smoothing can all be expressed as SSMs - see Proposition 1, Appendix B.1 for proof

---

> > > ### Author Response · Authors · 2022-11-17
> > > **References**
> > >
> > > 1. A. Gu, I. Johnson, K. Goel, K. Saab, T. Dao, A. Rudra, and C. R ́e. Combining Recurrent, Convolutional, and Continuous-time Models with Linear State Space Layers. NeurIPS 2021.
> > > 2. A. Gu, K. Goel, and C. R ́e. Efficiently Modeling Long Sequences with Structured State Spaces. ICLR 2022.
> > > 3. Y. Tay, M. Dehghani, S. Abnar, Y. Shen, D. Bahri, P. Pham, J. Rao, L. Yang, S. Ruder, and D. Metzler. Long Range Arena : A Benchmark for Efficient Transformers. ICLR 2021
> > > 4. H. Zhou, S. Zhang, J. Peng, S. Zhang, J. Li, H. Xiong, and W. Zhang. Informer: Beyond Efficient Transformer for Long Sequence Time-Series Forecasting. AAAI 2021.
> > > 5. H. Wu, J. Xu, J. Wang, and M. Long. Autoformer: Decomposition Transformers with Autocorrelation for Long-term Series Forecasting. NeurIPS 2022.
> > > 6. T. Zhou, Z. Ma, Q. Wen, X. Wang, L. Sun, and R. Jin. Fedformer: Frequency enhanced decomposed transformer for long-term series forecasting. ICML 2022.

---

### Decision · Program_Chairs · 2023-01-20

**Decision:**

Accept: poster

**Justification For Why Not Higher Score:**

Reviewers have concerns about the presentation.

**Justification For Why Not Lower Score:**

The paper is sufficiently novel, the technical development seems correct, and reviewers agree the results are impressive.

**Metareview: Summary, Strengths And Weaknesses:**

The paper presents SpaceTime, a method for modeling sequence data that has efficient training procedure, and can forecast long horizons.

Strength:

While the general framework of the current paper is similar to previous SSM-based methods (e.g., S4), the specific construction is different: instead of using diagonal + low-rank, the current paper uses shift + low-rank (both result in efficient computation for the recurrence over time).  The paper also shows that previous work has the limitation of not able to model AR(p) while the proposed method can. The authors also propose to perform long-horizon forecasting by learning the last-layer input sequence. The theoretical contributions are novel and interesting.

The paper also presents quite impressive experimental results from a variety of domains.

Weakness:

Reviewers had concerns on the presentation of the background. The paper is technical and has a lot of materials. But people not familiar with the SSM methods may be lost during the initial read, as a lot of introductory materials are located in the supplementary materials which are important to understand the claims. There have been helpful discussions on this paper and hopefully the presentation is improved in the next version.

**Note From Pc:**

if the above contains the word "oral" or "spotlight" please see: "oral" presentation means -> notable-top-5% and "spotlight" means -> notable-top-25%. As stated in our emails, we are disassociating presentation type from AC recommendations